# Immune gene expression networks in sepsis: A network biology approach

**Kyung Soo Kim**[ID][1], **Dong Wook Jekarl**[ID][2,3]*, **Jaeeun Yoo**[3,4], **Seungok Lee**[3,4], **Myungshin Kim**[2,3], **Yonggoo Kim**[2,3]

1 Department of Thoracic and Cardiovascular Surgery, Seoul St. Mary's Hospital, College of Medicine, The Catholic University of Korea, Seoul, Korea, 2 Department of Laboratory Medicine, Seoul St. Mary's Hospital, College of Medicine, The Catholic University of Korea, Seoul, Korea, 3 Laboratory for Development and Evaluation Center, Seoul St. Mary's Hospital, College of Medicine, The Catholic University of Korea, Seoul, Korea, 4 Department of Laboratory Medicine, Incheon St. Mary's Hospital, College of Medicine, The Catholic University of Korea, Seoul, Korea

* bonokarl@catholic.ac.kr

**Data Availability Statement:** All data generated or analyzed during this study are included in this published article (and supplementary material files). The GSE57065, GSE64456, GSE72829, and GSE95233 datasets were downloaded from the

## Abstract

To study the dysregulated host immune response to infection in sepsis, gene expression profiles from the Gene Expression Omnibus (GEO) datasets GSE54514, GSE57065, GSE64456, GSE95233, GSE66099 and GSE72829 were selected. From the Kyoto Encyclopedia of Genes and Genomes (KEGG) immune system pathways, 998 unique genes were selected, and genes were classified as follows based on gene annotation from KEGG, Gene Ontology, and Reactome: adaptive immunity, antigen presentation, cytokines and chemokines, complement, hematopoiesis, innate immunity, leukocyte migration, NK cell activity, platelet activity, and signaling. After correlation matrix formation, correlation coefficient of 0.8 was selected for network generation and network analysis. Total transcriptome was analyzed for differentially expressed genes (DEG), followed by gene set enrichment analysis. The network topological structure revealed that adaptive immunity tended to form a prominent and isolated cluster in sepsis. Common genes within the cluster from the 6 datasets included *CD247*, *CD8A*, *ITK*, *LAT*, and *LCK*. The clustering coefficient and modularity parameters were increased in 5/6 and 4/6 datasets in the sepsis group that seemed to be associated with functional aspect of the network. GSE95233 revealed that the nonsurvivor group showed a prominent and isolated adaptive immunity cluster, whereas the survivor group had isolated complement-coagulation and platelet-related clusters. T cell receptor signaling (TCR) pathway and antigen processing and presentation pathway were down-regulated in 5/6 and 4/6 datasets, respectively. Complement and coagulation, Fc gamma, epsilon related signaling pathways were up-regulated in 5/6 datasets. Altogether, network and gene set enrichment analysis showed that adaptive-immunity-related genes along with TCR pathway were down-regulated and isolated from immune the network that seemed to be associated with unfavorable prognosis. Prominence of platelet and complement-coagulation-related genes in the immune network was associated with survival in sepsis. Complement-coagulation pathway was up-regulated in the sepsis group that was associated with favorable prognosis. Network and gene set enrichment analysis supported elucidation of sepsis pathogenesis.

GEO website (https://www.ncbi.nlm.nih.gov/geo/). The GSE66099 and GSE54514 results published here are in whole or in part based on the publication at https://doi.org/10.1101/095489 and the data and results obtained from the synapse portal accessible at https://doi.org/10.7303/syn5612563.

**Funding:** This study was funded by Seoul St. Mary's Hospital, The Catholic University of Korea to promote scientific work. This hosptial is affiliated to The Catholic University of Korea, Seoul, Korea. Six of the authors work at University. As following authors work at the hospital, they received a salary from Incheon St Mary's Hospital, which is affiliated to The Catholic University of Korea, Seoul, Korea.: Kyung Soo Kim, Dong Wook Jekarl, Myungshin Kim, Yonggoo Kim, Jaeeun Yoo and Seungok Lee. The funders had no role in study design, data collection and analysis, decision to publish, or preparation of the manuscript.

**Competing interests:** The authors have declared that no competing interests exist.

## Introduction

Sepsis is defined as a documented infection with immune dysfunction or dysregulated host response with organ dysfunction [1–3]. Sepsis cases are composed of a heterogeneous subset of patients with various biological and clinical characteristics, including age, medication, underlying disease, causative microbes, and microbes with different antibiotic resistances [4, 5]. These features might act as confounding factors or cause in patients various immune responses or immune dysfunction.

Immune dysfunction or dysregulated host response in sepsis is associated with defective antigen presentation, defective adaptive immunity, defective NK cell activity, decreased immunoglobulin levels, neutrophil abnormalities, hypercytokinemia, complement consumption, and defective bacterial removal [6–8]. These features occur in the time course of disease by themselves or as a combination of more than two features. Measurement of immune-related molecules such as procalcitonin, C-reactive protein, interleukins, chemokines, and other immune-related molecules showed that they were associated with these features [9, 10]. Gene expression profiles from sepsis patients provided comprehensive data that could minimize selection bias and exhibit a wide range of biological responses [11–17]. Up- or down-regulated genes and pathways related to sepsis support the understanding of sepsis [11–13].

The immune response in healthy and diseased states is rarely attributed to single molecules, but rather to complex inter-related molecules. As sepsis-related immune response molecules are complex, a network approach might enhance our understanding of the equilibrium of the immune response in sepsis [18, 19]. A network approach analysis of a small group of cytokines revealed that the network constructed from sepsis patients' data had a smaller network diameter and shorter shortest path and characteristic path than did the control network [20]. These topologic features were related to decreased network function or modularity of the network. In addition, compared to that of day 1, the day 4 cytokine network was decreased in the sepsis group, which was in line with previous literature [21]. These studies constructed networks based on measured immune-associated molecules that could be regarded as actual observable characteristics of sepsis. As molecules could be analyzed independently, analysis of multiple molecules as a system was required. Analysis of these molecules via network analysis including topologic parameters and visualization resulted in an informative outcome. However, previous studies constructed networks based on cytokine measurements in small groups, which resulted in a limited number of recruited molecules. Measuring molecules in multiplex methods is relatively unfeasible compared to measuring gene expression.

In this study, gene expression signatures from a public database, Gene Expression Omnibus (GEO), were analyzed. Total transcriptome was analyzed for differentially expressed genes (DEGs) between healthy control and sepsis groups, followed by gene set enrichment analysis for DEGs, which revealed up- or down-regulated pathways. We selected 998 immune-related genes included in immune response pathways using gene expression profiles to construct a network for the healthy control and sepsis groups.

## Methods

### Cases in the cohort

This was a retrospective study, and our protocol was approved by the Institutional Review Board (IRB) of Incheon St. Mary's Hospital and Seoul St. Mary's Hospital in accordance with the Declaration of Helsinki. As public datasets were used, informed consent was waived by the IRB. The public datasets were downloaded from the Gene Express Omnibus (GEO) website (http://www.ncbi.nlm.nih.gov/geo/). The mentioned databases were searched using the

following common terminology: sepsis, human, GPL570 (Affymetrix Human Genome U133 Plus 2.0 Array or hgu133plus2.db), GPL6947 (Illumina HumanHT-12 V3.0 expression bead-chip), and GPL10558 (Illumina HumanHT-12 V4.0 expression beadchip) [22, 23]. These plat-forms were selected because they allow the study of genes related to the immune response. GPL570, GPL6947, and GPL10558 consisted of 58075, 48803, and 47231 probes, respectively. Entering GPL570, GPL6947, and GPL10558 with other terminology, 74 datasets or series were found. Among them, the GSE54514 [11], GSE57065 [12], GSE64456 [13], GSE66099 [14], GSE72829 [15], and GSE95233 [16] datasets were selected because their study designs were suitable to the purpose of this study [19], which included a healthy control group and a sepsis group, with more than 10 samples in each group. Datasets with drug treatment, studied tissue other than whole blood, and known sepsis etiology were excluded (S1 Fig in S1 File). All the datasets used transcriptomes derived from whole blood. Data regarding the time point of gene expression were derived from the first sample after admission. GSE66099 and GSE72829 were composed of several unique datasets, and healthy control and septic shock data from GSE66099 were selected. Healthy control cases and initial sepsis cases were used for each data-set. In the case of GSE72829, a healthy control and a definite bacterial infection group were selected. Randomized selection of sepsis cases without replacement was performed to match the number of healthy control cases and form a balanced correlation matrix. GSE95233 was additionally analyzed for survivor and nonsurvivor sepsis cases (GSE95233-1). Baseline char-acteristics of the datasets are listed in Table 1.

## Sepsis diagnosis

Employed gene expression datasets used sepsis diagnosis based on sepsis criteria [1]. The patients were classified as having systemic inflammatory response syndrome if they met two or more of the following criteria: WBC count $\geq 12.0 \times 10^9$/L or $\leq 4.0 \times 10^9$/L, or presenting more than 10% immature cells; body temperature $\geq 38°$ C or $\leq 36°$ C; respiratory rate $\geq 20$ breaths per minute, or heart rate $\geq 90$ beats per minute. Sepsis was diagnosed with SIRS and documented bacterial infection, and septic shock was diagnosed with sepsis and hypotension not responding to fluid resuscitation [1].

## Bioinformatics

For differentially expressed genes (DEG) analysis between control and patient groups based on total transcriptome, gene set enrichment for pathway analysis and network construction, several appropriate R software packages were used [24], including Bioconductor [25], affy [26], GEOquery [27], lumi [28], and limma [29]. The data were downloaded from the GEO database, and the data matrix of gene expression values was formatted using the GEOquery package [27]. In the case of Affymetrix gene expression data sets, the raw microarray data (CEL files) were normalized using the Robust Multichip Averaging (RMA) method, and log2 transformation was performed [25–27]. In the case of Illumina gene expression data sets, the arrays were normalized using the quantile method [27, 29]. The data were annotated with cor-responding gene names using R packages hgu133plus2.db, illuminaHumanv3.db, and illu-mina-Humanv4.db [30, 31].

## Gene set enrichment for pathway analysis

Analysis of DEG was performed for total transcriptome. After a t-test using basic functions in R software, genes were selected based on false discovery rate of less than 0.01 (FDR < 0.01) [32]. Based on DEG analysis, gene set enrichment for pathway analysis using GAGE package, based on R language, was performed [33]. GAGE package applies input data to the Kyoto

**Table 1. Baseline characteristics of recruited datasets for adult and paediatric patient.**

| | GSE54514 | GSE57065 | GSE95233 | GSE64456 | GSE66099 | GSE72829 |
|---|---|---|---|---|---|---|
| **Patient type** | **Adult** | **Adult** | **Adult** | **Pediatric** | **Pediatric** | **Pediatric** |
| Age, mean (yr) [a,b] | ≥18, 60.3 | ≥18, 62 | ≥18, 65 | ≤2 mo, 29 d | ≤10, 3.7 | ≤17, 1.9 mo |
| Sex (male %)[b] | 40 | 67.9 | 65 | 52 | 58 | 62 |
| Nonsurvivor (%) | 25.7 | 17.9 (28 d) | NA | NA | 16.3 | 19.2 |
| Cohort description | ICU with sepsis or | ICU with | ICU with | ED with sepsis | PICU with sepsis or | Discovery group |
| | septic shock | septic shock | septic shock | (PECARN study) | septic shock | (IRIS study) |
| Severity evaluation | APACHEII | SOFA | SOFA | YOS[b] | PRISM | NA |
| Control (n) | 18 | 25 | 22 | 19 | 49 | 52 |
| Sepsis or septic shock (n) | 35 | 28 | 51 | 89 | 198 | 94 |
| Randomized selection without replacement (septic shock) | 18 | 25 | 22 | - | - | - |
| Randomized selection without replacement (sepsis) | - | - | - | 18 | 49 | 52 |
| Measured frequency, frequency (time points) | 5 (1, 2, 3, 4, 5 d) | 3 (0, 24, 28 hr) | 3 (1–2, 3–4, 7–10 d) | 1 | 1 | 1 |
| Recruited time points[c] | 0–24 hr | 0–24 hr | 1–2 d | NA | 0–24 hr | 5 d |
| Platform | Illumina, GPL6947 | Affymetrix, GPL570 | Affymetrix, GPL570 | Illumina, GPL10558 | Affymetrix, GPL570 | Illumina, GPL10558 |
| Reference | [11] | [12] | [16] | [13] | [14] | [15] |

ED, emergency department; ICU, intensive care unit; NA, data not available; hr, hour; d, day; APACHEII, acute physiology and chronic health evaluation II; SOFA, sequential (sepsis related) organ failure assessment.

[a]GSE57065 and GSE72829 were summarized using median age.

[b] Percent of male and age are summarized data derived from the previous studies, which could be different from the analysis performed in this study. The sepsis patient was randomly selected corresponding to the number of the control cases.

[c]YOS, Yale observation score (6–30).

Encylopedia of Genes and Genomes (KEGG) pathways. Significant pathways were plotted for heatmap that was selected based on global *P*-value of less than 0.05.

## Immune response related genes

To analyze the function of the immune response in the control and sepsis groups, KEGG immune system pathways were used [34]. Twenty-one pathways of the immune system were included, as follows: hematopoietic cell lineage (has04640); complement and coagulation cascade (has04610); platelet activation (hsa04611); Toll-like receptor signaling (hsa04620); Toll and Imd signaling (hsa04624); NOD-like receptor signaling pathway (hsa04621); RIG-I-like receptor signaling (hsa04622); cytosolic DNA-sensing (hsa04623); C-type lectin receptor signaling (hsa04625); natural killer cell-mediated cytotoxicity (hsa04650); antigen processing and presentation (hsa04612); T cell receptor signaling (hsa04660); Th1 and Th2 cell differentiation (04658); Th17 cell differentiation (hsa04659); B cell receptor signaling (hsa04662); Fc epsilon RI signaling (hsa04664); Fc gamma R-mediated phagocytosis (hsa04666); leukocyte transendothelial migration (hsa04670); intestinal immune network for IgA production (hsa04672), and chemokine signaling (hsa04062). The 21 pathways were assigned to adaptive immunity, innate immunity, NK cell activity, leukocyte migration, antigen presentation, complement-coagulation, platelet activity, cytokines-chemokines, signaling, and hematopoiesis. From them, 1908 genes in these pathways and 998 unique genes were extracted (S1 and S2 Tables in S1 File).

As genes were included in multiple pathways, we utilized Gene Ontology (GO) and Reactome pathways for additional information [35, 36]. From the GO pathway, we included activation of the innate immune response, complement activation, and immune response-activating signal transduction. From the immune response pathway, the adaptive immune response and the innate immune response were included. Leukocyte homeostasis, leukocyte activation, and leukocyte migration were also collected. From the Reactome database, we included the hemostasis pathway; platelet homeostasis; platelet adhesion to exposed collagen; platelet activation, signaling, and aggregation; formation of fibrin clots; dissolution of fibrin clots, and cell surface interactions at the vascular wall. In the immune system pathway, adaptive immune system, innate immune system, and cytokine signaling in the immune system were included (S3 Table in S1 File).

Genes included in multiple pathways were assigned to one pathway with their most frequent function among the KEGG, GO, and Reactome pathways. If the frequency of the gene that was included in the pathway was equal, undetermined (U) value was assigned. If the frequency was equal and included chemokines-cytokines, signaling, or hematopoiesis pathways, pathways other than these ones were assigned (S3 Table in S1 File).

## Network topology analysis

The network was constructed using Spearman's correlations equal to or greater than 0.8 and a P-value of less than 0.05 for each set of molecules. Each gene was regarded as a node, and the correlation pairs between the genes were regarded as links [37]. Links were created between the nodes with significant correlation coefficients. Network analysis results were plotted, and the topologic parameters were calculated using Cytoscape and an additional application, NetworkAnalyzer [37–39]. The clustering coefficient is defined as the number of links between associated nodes. The degree of a node is defined as the number of links connected to a node (gene), and a hub is defined as a node with a relatively high degree. Network density is the ratio of links in the network to the maximum possible number of links. The length of a path is the number of unique links between the nodes that are present in the path. Distance is the shortest path length between two nodes. The network diameter is the maximum length of the shortest path. The characteristic path length is the average shortest path length and provides the expected distance between two nodes. The average number of neighbors is the number of identical nodes connected to a node. The normalized version of the average number of neighbors is the network density. Network heterogeneity is the variance in the number of links of a node divided by the number of mean links. The average number of neighbors indicates the average connectivity of the nodes. The topological coefficient is a relative measure of the extent to which a node shares neighbors with other nodes. Network centralization is close to one when the network resembles a star-like figure, while uniformly connected networks have a centralization value close to zero. The stress centrality of a node is the number of shortest paths passing through that node. The betweenness centrality of a node is the number of shortest paths between two other nodes that pass through that node, divided by the total number of shortest paths between the two nodes; betweenness centrality reflects the amount of control that this node exerts over other nodes. Closeness centrality reflects how close nodes are, which implies that nodes with high closeness centrality values have the shortest distance to other nodes on average. Eccentricity reflects the maximum distance of a node from other nodes [38, 39]; a high eccentricity value means that all other nodes are in proximity, a low eccentricity value means that there are at least one node and its neighbors that are far from that node. Degree correlation defines the relationship between the degree of a node and that of connected nodes [40]. Modularity is defined as the tendency of nodes to cluster and to interact,

coordinating biological processes [41, 42]. The parameters were analyzed using comparison of proportions (MedCalc Software version 18.2.1, Mariakerke, Belgium).

## Results

### Gene set enrichment for pathway analysis

Total transcriptome analysis of healthy control and sepsis group revealed that 176 to 8229 genes showed expression differences in datasets. Gene set enrichment for pathway analysis was performed based on KEGG pathways (S4 Table in S1 File). Among up-regulated or down-regulated pathways with statistical significance, we searched for pathways associated with immune-related pathway (S2–S6 Figs in S1 File). We found that complement and coagulation cascade pathway and Fc epsilon RI signaling and Fc gamma R-mediated phagocytosis pathways were up-regulated in 2/3 of datasets from adult patients and in 3/3 of those from pediatric patients. T cell receptor signaling pathway was down-regulated in 2/3 and 3/3 of datasets from adult and pediatric patients, respectively. Antigen presentation and processing were down-regulated in 2/3 of datasets from adult and pediatric patients (Table 2). Altogether, these findings were interpreted along with the network analysis results. In network analysis, adaptive immunity genes were prominent and isolated from network in sepsis patient group (Fig 1). T cell receptor signaling (TCR) pathway was down-regulated in 5/6 of the datasets. These data imply that adaptive immunity related with T cells might be impaired or dysregulated in sepsis. In addition, adaptive immunity was associated with unfavorable prognosis in sepsis group. Complement and coagulation cascade was prominent in sepsis survivors of GSE95233 datasets (Fig 2). Complement and coagulation cascade was increased in sepsis in 5/6 of datasets and these data imply that these pathways may be involved in convalescence process.

### Network structure and immune function

The network topologic structure revealed that, in sepsis, adaptive immunity genes (green colored) formed a prominent cluster, isolated from the rest of the network (Fig 1 and S7–S18 Figs in S1 File). In that isolated cluster, *CD247* (CD3 Zeta chain), *CD8A*, *ITK* (tyrosine protein kinase ITK / TSK), *LAT* (linker for activation of T cells), *LCK* (leukocyte C-terminal Src kinase) were found in all 6 datasets. *CD2*, *FYN* (Src family tyrosine kinase), *GATA3* (GATA-binding protein 3), *IL7R*, *RASGRP1* (RAS guanyl-releasing protein 1) were all found in adult datasets, but 2/3were found in the pediatric group. *CBLB* (E2 ubiquitin protein ligase CBL-B), *CD3D*, *CD3G*, *ZAP70* (Zeta-chain-associated protein kinase 70) were found in 2/3 of adult datasets, but all were found in pediatric datasets. Other genes are listed in S5 Table in S1 File. In addition, NK cell activity (light blue) tended to be included in the network in the sepsis group, whereas it was isolated in the healthy control group. The nonsurvivor network from GSE95233 revealed that adaptive immunity was isolated and prominent from the rest of the network, whereas platelet activity (olive green) and the compliment and coagulation cascade pathway (yellow) were isolated and prominent in the survivor group (Fig 2). Platelet, complement and coagulation-related genes were prominent in survivors; *C6*, *C7*, *F11*, *COL1A1*, *COL3A1*, *ADCY6*, *GP5*, *FGG*, *MYLK4*, *SERPINE1*, *SERPIND1*, *CR2*, etc. were included. *CD8B*, *IL1A*, *IL3*, *IL12B*, *IL21*, *IL23R*, and other genes were also included. The isolated gene cluster from nonsurvivors included *CD2*, *CD3D*, *CD3G*, *CD8A*, *CD247*, *CBLB*, *GATA3*, *ITK*, *LCK*, *CTLA4*, *IL10*, *ZAP70*, and other genes.

### Network parameter analysis

Comparison of topologic parameters revealed that the clustering coefficient in sepsis was increased in 2/3 of adult datasets and in 3/3 of pediatric datasets (Table 3). Network

**Table 2. Differentially expressed gene analysis between control and sepsis followed by gene set pathway analysis[a].**

|  | GSE54514 | GSE57065 | GSE95233 | GSE64456 | GSE66099 | GSE72829 |
|---|---|---|---|---|---|---|
| Patient type | Adult | Adult | Adult | Pediatric | Pediatric | Pediatric |
| Control cases (n) | 18 | 25 | 22 | 19 | 49 | 52 |
| Sequential selection of sepsis cases (n) | 18 | 25 | 22 | 19 | 49 | 52 |
| Differentially expressed Genes (n) | 176 | 7885 | 6784 | 3123 | 8229 | 7463 |
| Up regulated pathway (n) | 0 | 60 | 44 | 13 | 78 | 51 |
| Down regulated pathway (n) | 0 | 33 | 33 | 16 | 33 | 15 |
| Up regulated immune pathway |  | C, Fc, RIG | C, Fc | C, Fc, NK, K, TLR | APC, C, CXC & etc, Fc, IgA, | BCR, C, Fc, CXC & etc, LK, NK, TLR |
| Down regulated immune pathway |  | APC, TCR, H, K | APC, BCR, CXC & etc, IgA, TCR | APC, IgA, TCR | H, TCR | APC, IgA, TCR |

[a]Differentially expressed gene was selected based on FDR (P<0.01).

APC, Antigen processing and presentation; BCR, B cell receptor signaling; C, Complement and coagulation; CXC & etc, chemokine signaling; Fc, Fc gamma R mediated phagocytosis or Fc epsilon RI signaling; H, hematopoietic cell lineage; IgA, intestinal immune network for IgA production; LM, leukocyte endothelial migration; NK, NK cell mediated cytotoxicity; RIG, RIG-I like receptor signaling; TLR, Toll like receptor signaling; TCR, T cell receptor signaling.

heterogeneity was increased in 3/3 of adult datasets, but was decreased in 3/3 of pediatric datasets. Modularity value was increased in 1/3 of adult datasets, whereas modularity value was increased in 3/3 of pediatric datasets in the sepsis group. Shortest path was decreased in 2/3 of adult and pediatric datasets, whereas average degree was increased in 2/3 of adult and pediatric datasets (Table 3). For GSE95233-1 analysis, difference between network parameters was calculated as follows: (case—control / control) x 100. The difference of clustering coefficient between survivor (0.390) and nonsurvivor (0.337) was -13.5%. The difference of modularity between survivor (0.581) and nonsurvivor (0.694) was 19.4%.

We searched for nodes with such a high degree that they could be considered as hub nodes. In the GSE54514 control and sepsis groups, *CD3D* and *ITPR3* (inositol 1,4,5 trisphosphate receptor type 3) had degree 33 and 43, respectively. In the GSE57065 control and sepsis groups, *ATG5* (autophagy-related 5), and *GBP5* (guanylate-binding protein 5) had degree 19 and 25, respectively. In the GSE95233 control and sepsis groups, *IFNGR1* (interferon gamma receptor 1), and *ESAM* (endothelial cell selective adhesion molecule) had degree 49 and 21, respectively.

In the GSE64456 control and sepsis groups, *LYN* (tyrosine protein kinase Lyn), and *IFI16* (interferon gamma-inducible protein 16) had degree 56 and 70, respectively. In the GSE66099 control and sepsis groups, *NCF2* (neutrophil cytosolic factor 2), *NLRP12* (NLR family pyrin domain-containing 12), and *NCR2* (natural cytotoxicity-triggering receptor 2) had degree 26, 26, and 34, respectively. In the GSE72829 control and sepsis groups, *C5AR1* (complement C5a receptor 1) and *CSF3R* (colony-stimulating factor 3 receptor) had degree 50 and 46, respectively. These network parameters are listed in S6–S17 Tables in S1 File.

## Discussion

The features of immune dysfunction or immune response in sepsis include defective antigen presentation and adaptive immunity (T and B cell abnormalities), abnormalities in NK cell activity, diminished immunoglobulin level, alteration in neutrophils, abnormal cytokine levels, unbalanced pro- and anti-inflammatory cytokines, complement consumption, and defective bacterial killing [4, 5]. This complexity hampers understanding of immunologic features in

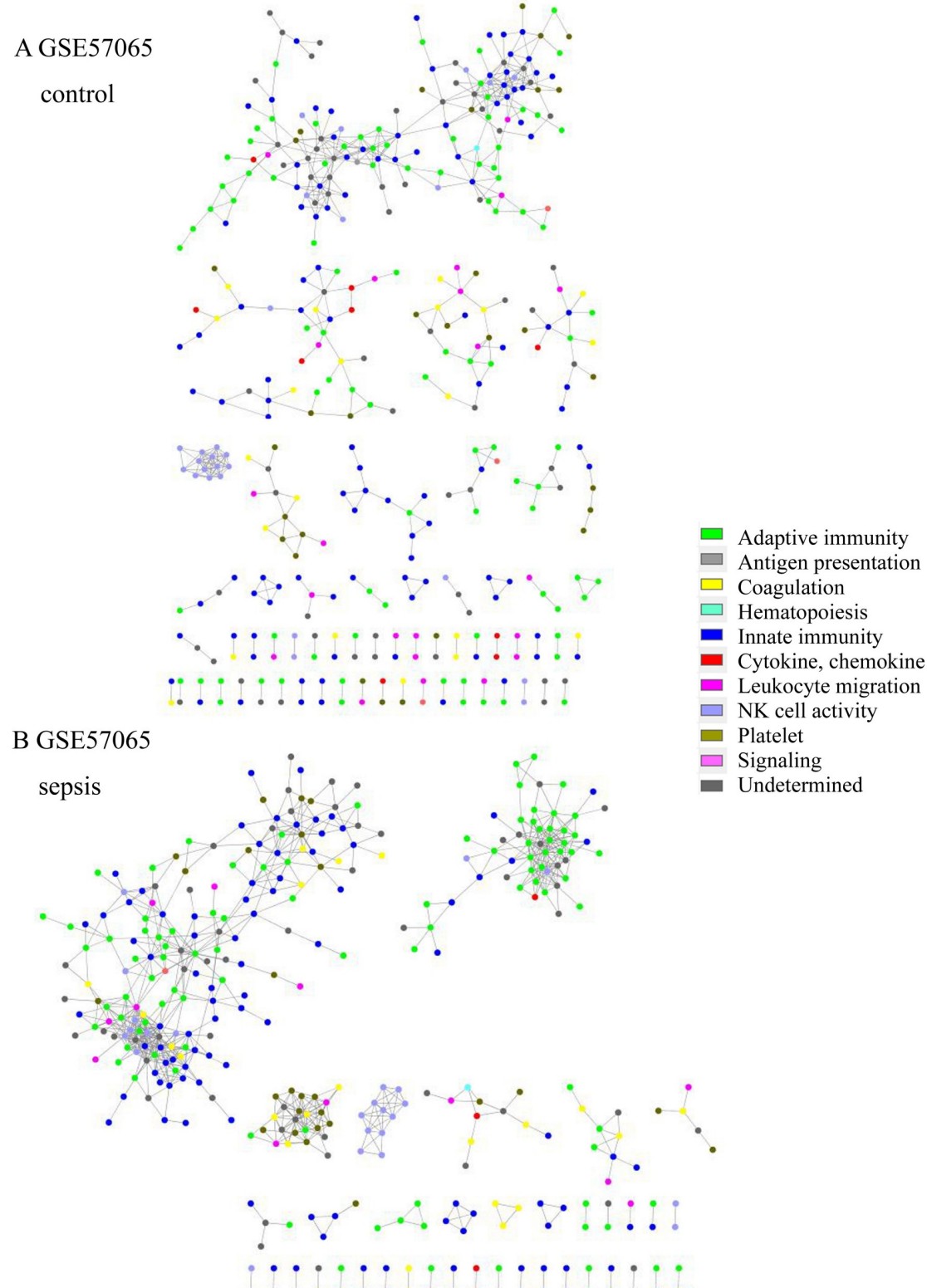

**Fig 1.** Network structure of GSE57065 of the (A) control group and (B) sepsis group. In the sepsis group, the adaptive immunity network was prominent and isolated (green).

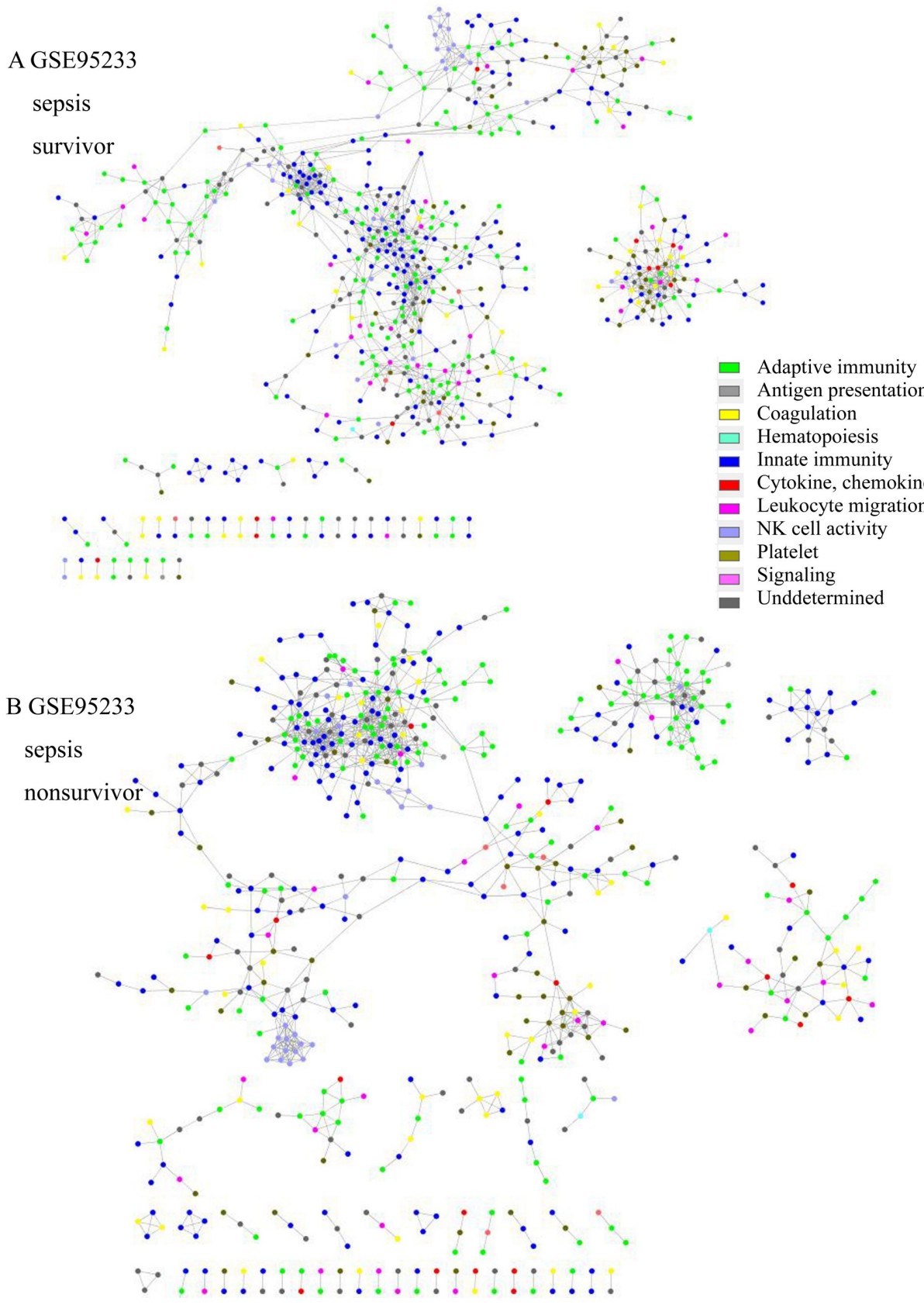

**Fig 2.** Network structure of GSE95233 of (A) sepsis survivors and (B) sepsis nonsurvivors. In the survivor group, the platelet (olive green) and coagulation cascade network (yellow) were isolated and prominent. In the nonsurvivor group, the adaptive immunity (green) and innate immunity networks (blue) were prominent and isolated.

sepsis. Immune suppression, immune paralysis and hypo-inflammatory response are reported in sepsis that is intermingled with hyperinflammatory state [6–8].

This study showed that in the adult sepsis group, there were more down-regulated pathways, whereas in the pediatric group, there were more up-regulated pathways. These data suggest that immune suppression might be more dominant in the adult sepsis group, whereas hyper-inflammatory states might be dominant in the pediatric sepsis group. The mechanism of immune response was indeed different between adult and pediatric patients in sepsis. However, down-regulation of TCR and APC pathway and up-regulation of complement-coagulation cascade pathway was found in most of adult and pediatric sepsis patients. These findings are congruent with previous reports that, in sepsis, CD4+ T cell showed cell exhaustion, increased apoptosis and decreased adhesion molecules, and CD28 and TCR diversity. In sepsis, CD8+ T cell also showed cell exhaustion, increased apoptosis, and decreased cytotoxic function, cytokine secretion, and TCR diversity [8]. Antigen presentation by follicular dendritic cell or dendritic cells is decreased in sepsis [8]. In the present study, isolation and prominent adaptive immune genes from the rest of the immune network were noted, and these data imply that part of the adaptive immune function was not in equilibrium or in a regulated state. As this isolated adaptive immune network was found in nonsurvivors, initial network state along with down-regulated pathways related with adaptive immunity resulted in unfavorable prognosis.

**Table 3. Network topological parameters from gene expression data from GEO dataset for adult and paediatric patient.**

| | GSE54514 | | GSE57065 | | GSE95233 | | GSE95233-1 | | GSE64456 | | GSE66099 | | GSE72829 | |
|---|---|---|---|---|---|---|---|---|---|---|---|---|---|---|
| Patient type | Adult | | Adult | | Adult | | Adult | | Pediatric | | Pediatric | | Pediatric | |
| Disease type | Control | septic shock | control | septic shock | control | septic shock | survivor | non survivor | Control | sepsis | Control | sepsis | Control | sepsis |
| Clustering coefficient | 0.319 | 0.323 | 0.328 | 0.280 | 0.328 | 0.367 | 0.390 | 0.337 | 0.367 | 0.390 | 0.337 | 0.388 | 0.381 | 0.435 |
| Network density | 0.013 | 0.016 | 0.012 | 0.010 | 0.012 | 0.039 | 0.026 | 0.014 | 0.039 | 0.026 | 0.014 | 0.012 | 0.030 | 0.034 |
| Network heterogeneity | 1.039 | 1.142 | 1.151 | 1.172 | 1.151 | 1.504 | 1.358 | 1.193 | 1.504 | 1.358 | 1.193 | 1.091 | 1.249 | 1.129 |
| Connected components | 27 | 30 | 42 | 61 | 42 | 34 | 33 | 47 | 34 | 33 | 47 | 32 | 16 | 25 |
| Network diameter | 19 | 17 | 14 | 15 | 14 | 12 | 19 | 11 | 12 | 19 | 11 | 14 | 13 | 9 |
| Network centralization | 0.059 | 0.079 | 0.103 | 0.048 | 0.103 | 0.196 | 0.113 | 0.075 | 0.196 | 0.113 | 0.075 | 0.078 | 0.148 | 0.121 |
| Shortest path | 160514 | 80598 | 59638 | 17222 | 59638 | 142640 | 121152 | 11730 | 142640 | 121152 | 11730 | 36902 | 38092 | 22532 |
| Characteristic path length | 7.418 | 5.267 | 5.276 | 5.088 | 5.276 | 3.862 | 7.211 | 3.482 | 3.862 | 7.211 | 3.482 | 4.705 | 3.92 | 2.824 |
| Average degree, $<k>$ | 5.93 | 7.42 | 5.30 | 3.64 | 5.30 | 17.87 | 11.71 | 4.04 | 17.87 | 11.71 | 4.04 | 4.50 | 8.43 | 10.27 |
| Number of nodes, $N$ | 464 | 453 | 426 | 367 | 426 | 457 | 448 | 297 | 457 | 448 | 297 | 380 | 284 | 299 |
| Degree correlation, μ | 0.489 | 0.508 | 0.422 | 0.810 | 0.422 | 0.403 | 0.715 | 0.546 | 0.403 | 0.715 | 0.546 | 0.210 | 0.475 | 0.346 |
| Modularity, $Mc$ | 0.743 | 0.662 | 0.644 | 0.806 | 0.644 | 0.198 | 0.581 | 0.694 | 0.198 | 0.581 | 0.694 | 0.745 | 0.434 | 0.603 |

GSE95233-1 indicates survivors (n = 16) and non-survivors (n = 16) among sepsis patients.

Complement and coagulation pathway was up-regulated in sepsis patients. However, network analysis showed that platelet and complement-and-coagulation-related genes were prominent and isolated in survivors, and that this initial state resulted in a favorable prognosis. On the contrary, it is reported that persistently activated complement system, including C3a and C5a, is found in sepsis, which is associated with unfavorable prognosis [6–8]. Excessive activation of complement system potentiated activated innate immune system in sepsis, and this also resulted in an unfavorable prognosis in the sepsis group [6]. These findings imply that initial inactivation of complement-coagulation pathway or persistent, uncontrolled activation of these pathways might be associated with unfavorable prognosis.

Network analysis along with gene set enrichment analysis from gene expression profiles may have elucidated sepsis pathogenesis and host immune response in sepsis [11–17]. In our study, the interrelations among immune genes were analyzed using a network analysis approach that might reflect the comprehensive aspects of the dynamics of gene expression. Analysis of the topologic structure of the network in sepsis revealed that adaptive-immunity-related genes formed a cluster in sepsis patients (S7–S18 Figs in S1 File), which tended to be isolated.

Formation of a cluster or module with adaptive immune genes might be a normal process to respond to sepsis or might be a pathologic process associated with immune dysregulation. To elucidate the nature of the cluster, we analyzed survivors and nonsurvivors from the sepsis group in GSE95233. The survivor group showed a prominent and isolated cluster with complement-coagulation and platelet activity genes, whereas the nonsurvivor group showed a cluster with prominent and isolated adaptive immunity genes (Fig 2). Although this analysis included only one dataset, as a cluster of adaptive immunity was observed among the nonsurvivor group, these isolated and prominent genes seemed to be related to adverse effects. Platelet and complement cascade activation might be related to clearance of microbes, which supports the adaptive immune response [43, 44]. Activation of adaptive immunity is required to resolve sepsis, but the isolated and prominent adaptive immune response in this study seemed to be dysregulated or impaired. Additional data from gene set enrichment for pathway analysis showed that TCR pathway was down regulated in all the datasets. As the isolated cluster in network analysis included genes related to TCR pathway, adaptive immunity seemed to be suppressed or dysregulated, which resulted in an unfavorable prognosis [7, 45].

Clustering coefficient is one of the network topologic parameters that reflects the number of links between neighboring nodes, and its increased values denote an increased probability of similar biological function [45, 46]. In this study, the clustering coefficient was increased in 2/3 of adult datasets and 3/3 of pediatric datasets in the sepsis group, which implies increased function of the immune network. Previous studies revealed that an increased clustering coefficient correlated with bloodstream C-reactive protein concentration or inflammation in hepatocellular carcinoma patients before and after transarterial chemoembolization [47]. Network analysis revealed that inflammatory process seemed to be activated in sepsis group; further studies are required to understand the relation between the clustering coefficient and the inflammatory response in the immune network.

Modularity of a network is related to subgroups or a clusters within that network [48, 49]. High modularity implies dense connections between nodes within the same cluster but sparse connections between nodes in different clusters [50, 51]. Modularity is expected to promote evolvability and multifunctionality, and functions as a driving biological process [52, 53]. In this study, the sepsis group showed increased modularity values in 3/3 of the studied pediatric datasets. On the contrary, only 1/3 of the adult datasets in the sepsis group showed increased modularity. These results are consistent with the finding that, in the sepsis group, the pediatric datasets showed more significant pathways compared to those of the adult datasets (Table 2).

Extreme age groups (patients younger than 1 year or older than 50 years) are related with increases incidence and mortality of sepsis [54, 55]. Although organ development, immunity, immune cell subsets, cytokines response to antigen, and hormonal effects might be different between the 2 extreme age groups, we found a common pattern in this study. In particular, CD8A-related genes were isolated and TCR pathway was down-regulated, and antigen-presenting cells were isolated from rest of the immune network. CD8A molecule is related with CD8+ T cells, which is known to be quantitatively and qualitatively different in extreme age groups. CD8+ T cells are decreased in older age group, and CD8+ T cell in neonates are biased toward an innate immune response, unlike in adults, which functions as a cellular immune response [55, 56]. These findings could be part of prior information or knowledge that could help, directly or indirectly, elucidate pathobiology in sepsis, which might be associated with management of the sepsis. Further studies are required in regard to the functional aspect of the isolated cluster of adaptive immunity from network analysis and function of CD8+ T cell in sepsis. On the other hand, network analysis showed that network heterogeneity was higher in adult sepsis, whereas it was lower in pediatric sepsis. Gene set enrichment analysis showed that the adult sepsis group had more down-regulated pathways compared to the pediatric group, which had more up-regulated pathways. High network heterogeneity values imply that the nodes are connected with other nodes of different types, while lower values denote that the nodes are connected with similar other nodes. Altogether, these data suggest that immune senescence or immune hyperactivation are a driving force of immune response in adult and pediatric sepsis, respectively.

Among prominent clusters from survivors, *IL1A*, *IL12*, *IL21*, and *IL23* genes were also included. Although statistically insignificant, previous studies showed that the mean cytokines levels (pg/mL) for survivors (n = 63) and nonsurvivors (n = 17) were as follows [49]: IL1A, 24.8, 7.4; IL12, 15.7, 0; and IL21, 138.9, 10.1, IL23, 75.4, 16.7, respectively. Mean cytokine level of IL1A, IL12, IL21, IL23 was increased in survivors of sepsis. These data suggest that platelet, complement and coagulation related molecules and IL1A, IL12, IL21, IL23 might be closely related with convalescence process in sepsis.

Altogether, network analysis showed isolated components or isolated clusters of genes related to adaptive immune response. The genes in the cluster were related to adaptive immune response or T cells and the gene set enrichment analysis showed that the T cell signaling pathway was decreased in 2/3 of adult and 3/3 of pediatric sepsis datasets. From these findings, T cell signaling or T cell related functions seemed to be impaired or decreased in sepsis cases. Among survivors of sepsis, complement and coagulation cascade and platelet-related genes were prominent in network analysis. Gene set enrichment analysis showed that complement and coagulation cascade pathway was up-regulated in 2/3 of adult and 3/3 of pediatric sepsis cases. From these findings, complement and coagulation pathway seemed to be associated with convalescence process during sepsis process. Clustering coefficient that was related with inflammatory process was increased in 2/3 of adult and 3/3 in pediatric datasets.

A limitation of this study is that the use of various platforms might have increased variations among datasets. The cases recruited for normal control and sepsis was relatively small because of the small sample size of the normal control group. Survivors and nonsurvivors networks were analyzed in only one dataset; thus further studies are required for more robust results. In addition, only genes in the immune pathways were analyzed; though, other genes related to metabolism and the endocrine system might have an effect on the sepsis network. Multiple gene functional annotations were reduced to a gene function that included the most frequent function from the KEGG, GO and Reactome pathways, which might have caused some bias. As this study was a retrospective study using public datasets, sex and age was not

matched for control and sepsis cases. The results might have been affected by these parameters.

## Conclusion

Immune dysfunction might be caused by prominence and isolation of the adaptive immune response from the rest of the immune network. The isolated cluster included T cell receptor signaling gene and that pathway was down-regulated as resulting from gene set enrichment for pathway analysis. The increased clustering coefficient and modularity implied that an inflammatory response was activated in the sepsis group. Survivors of sepsis showed a prominent cluster of genes that was related to platelet and complement and coagulation cascade pathways. Up-regulated complement and coagulation cascade pathway in sepsis seemed to be related with convalescence process or with favorable prognosis. Network and gene set enrichment analysis supported elucidation of sepsis pathogenesis.

## Supporting information

**S1 File.**
(DOCX)

## Acknowledgments

The authors thank the researchers who provided data in the GEO public database.

## Author Contributions

**Conceptualization:** Kyung Soo Kim, Dong Wook Jekarl, Yonggoo Kim.

**Data curation:** Kyung Soo Kim, Dong Wook Jekarl, Jaeeun Yoo.

**Formal analysis:** Kyung Soo Kim, Dong Wook Jekarl, Jaeeun Yoo.

**Methodology:** Kyung Soo Kim, Dong Wook Jekarl, Jaeeun Yoo.

**Supervision:** Dong Wook Jekarl, Seungok Lee, Myungshin Kim, Yonggoo Kim.

**Writing – original draft:** Kyung Soo Kim, Dong Wook Jekarl.

**Writing – review & editing:** Kyung Soo Kim, Dong Wook Jekarl, Jaeeun Yoo, Seungok Lee, Myungshin Kim, Yonggoo Kim.

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
