## [Decision Letter · Decision Letter 0]

27 Jul 2020

PONE-D-20-17102

Immune gene expression networks in sepsis: A network biology approach

PLOS ONE

Dear Dr. JEKARL,

Thank you for submitting your manuscript to PLOS ONE. After careful consideration, we feel that it has merit but does not fully meet PLOS ONE’s publication criteria as it currently stands. Therefore, we invite you to submit a revised version of the manuscript that addresses the points raised during the review process.

ACADEMIC EDITOR:

There are major concerns regarding the quality of the analyses presented in the manuscript. The authors need to significantly revise the manuscript to address the points raised by both reviewers. Suitably addressing Reviewer 1's comments regarding the lack of technical rigor is critical.

We look forward to receiving your revised manuscript.

Kind regards,

Jishnu Das, Ph.D.

Academic Editor

PLOS ONE

Journal Requirements:

2. Please note that the DOI link provided to the synapse portal (doi:10.7303/syn5612563) in the Data Availability section of your Methods does not appear to be a working link. Please check that you have provided the correct link/DOI for this data.

3.Thank you for stating the following in the Acknowledgments Section of your manuscript:

[This study was supported by the Research Fund of Seoul St. Mary's Hospital, The Catholic University of Korea.]

 [The author(s) received no specific funding for this work.]

Reviewers' comments:

Reviewer's Responses to Questions

**Comments to the Author**

1. Is the manuscript technically sound, and do the data support the conclusions?

Reviewer #1: No

Reviewer #2: Yes

2. Has the statistical analysis been performed appropriately and rigorously? 

Reviewer #1: No

Reviewer #2: Yes

3. Have the authors made all data underlying the findings in their manuscript fully available?

Reviewer #1: Yes

Reviewer #2: Yes

4. Is the manuscript presented in an intelligible fashion and written in standard English?

Reviewer #1: Yes

Reviewer #2: Yes

5. Review Comments to the Author

Reviewer #1: Kim et al present a study focus on the expression of ~1000 genes associated with inflammation and immunity in the context of sepsis. They examine publicly available, retrospective gene expression datasets to define transcriptional networks. There are multiple issues with this study that limit its usefulness and originality:

1. The authors focus on only 998 genes. There have been multiple studies over the past nearly 20 years that have examined the full transcriptome after sepsis (with insights into inflammation and immunity, and including detailed network analyses), and so a focus on only inflammation and immunity is anachronistic at best and very incomplete at worst.

2. The criteria used for defining sepsis patients are from the 1st consensus definition; we are now at the 3rd consensus definition, and thus the applicability to the clinical setting is possibly limited.

3. The datasets used involve a mix of adult and pediatric patients. There is extensive literature documenting differences in pediatric vs. adult sepsis patients with regard to multiple inflammatory and immune responses as well as outcomes.

4. The source biological material for the transcriptomic analyses is not described. These data are could be from peripheral blood mononuclear cells or whole blood (or were they derived from other tissues?), making interpretation of this study and comparison to other studies difficult.

Reviewer #2: Dear Associate Editor,

The manuscript 'Gene expression networks in sepsis: A network biology approach' by Kim et al analyzes various immune system pathways in patients with sepsis compared to healthy controls using network analysis approach and show that adaptive immune cells are isolated and prominent in that network in patients with sepsis. The strength of this manuscript is that they have used diverse datasets for analysis and obtained comparable results across all the datasets. However, there are some concerns in this manuscript, which the authors have failed to address.

Major comment:

1. In the introduction, the authors nicely summarize the importance of network approach analysis in sepsis as shown by other groups, however two key elements are missing:

a. What is unique or novel about their network approach analysis compared to others and what is the rationale behind that?

b. What was their motivation to perform such an analysis in sepsis?

2. There is a confusion in the two main figures:

a. The dataset identifiers mentioned in the results section of the manuscript are contradictory to those in figures 1 and 2. It looks like the labeling in figure 1 and 2 are swapped.

b. The survivors and non-survivors (mentioned under figure 2 in the results section) have not been labelled in any of the main figures.

3. The discussion section is very long and the authors need to strike a balance between the results and the discussion in the way it is written. There are many elements in the discussion section that actually belongs to the results section.

4. The current notion is that IL-10 is an anti-inflammatory cytokine, although IL-10 has been shown to be increased amongst non-survivors (ref 28), there is no direct evidence to state that IL-10 is related to unfavorable prognosis. High IL-10 response in non-survivors could be the due to the result of a compensated anti-inflammatory response in sepsis, following the initial pro-inflammation. Therefore, the discussion on high IL-10 is related to unfavorable prognosis in sepsis without a proper evidence is an overstatement and not correct.

5. The discussion about the main computational finding on the role of adaptive immune cells in sepsis identified using their network analysis and their inference in the pathophysiology of SIRS or sepsis is not clear and needs to be elucidated.

Minor comments:

1. The authors should stick to using the abbreviations instead of full form after using it for the first time and have to be consistent throughout the manuscript.

Example: GEO – line 86, KEGG – line 150, GO – line 166

2. Are there any references for the rationale behind using such platforms? Lines 106 and 107.

3. What does ‘(n)’ in the table 1 indicate?

4. Why are the figure legends not listed separately and written along with the text?

5. Maintain a similar font and size throughout the manuscript, line 298 is different.

6. PLOS authors have the option to publish the peer review history of their article (what does this mean?). If published, this will include your full peer review and any attached files.

Reviewer #1: No

---

## [Author Response · Author response to Decision Letter 0]

12 Aug 2020

PONE-D-20-17102

Immune gene expression networks in sepsis: A network biology approach

Reviewer #1: Kim et al present a study focus on the expression of ~1000 genes associated with inflammation and immunity in the context of sepsis. They examine publicly available, retrospective gene expression datasets to define transcriptional networks. There are multiple issues with this study that limit its usefulness and originality:

1. The authors focus on only 998 genes. There have been multiple studies over the past nearly 20 years that have examined the full transcriptome after sepsis (with insights into inflammation and immunity, and including detailed network analyses), and so a focus on only inflammation and immunity is anachronistic at best and very incomplete at worst.

 Thank you for reading and managing this article and the precise comments. 

The definition of sepsis was revised in 2016 (SEPSIS-3) that includes dysregulated host response to infection. These response includes cellular, metabolic, circulatory, neurologic, and immune dysregulation. Oxidative stress or mitochondrial dysfunction, insulin resistance, hypoxemia or hypotension, metabolic acidosis, encephalopathy or neurologic dysregulation are known dysfunctions in sepsis. Among these dysregulated host response, the authors intended to study immune dysfunction via 998 immune related gene expression in this study. 

Based on the comment, we have performed full transcriptome analysis by differentially expressed genes (DEG) analysis followed by gene set enrichment for pathway analysis (Table 3). Differentially expressed gene (DEG) was performed between control and patient group with equal sample number. After t-test, genes were selected based on false discovery rate of less than 0.01 (FDR<0.01) except GSE54514, which used global P-value. Based on DEG, gene set enrichment for pathway analysis using GAGE package using R software was performed (Supplementary figure S14-S19, Supplementary Table S17). 

Among these pathways, the authors intended to study pathway that was related to immune function. Among up regulated or down regulated pathways with statistical significance, pathways associated with immune related pathway was searched. We have found that complement and coagulation cascade pathway and Fc epsilon RI signaling or Fc gamma R mediated phagocytosis pathway was increased in 66% (2/3) and 100% (3/3) datasets from adult and paediatric patients, respectively. T cell receptor signaling pathway was decreased in 100% (3/3) and 100% (3/3) datasets from adult and paediatric patients, respectively. Antigen presentation and processing was decreased in 66% (2/3) and 66% (2/3) datasets from adult and paediatric patients, respectively. 

Altogether, these findings could be interpreted along with the network analysis results. In network analysis, adaptive immunity associated with CD3 and CD8 genes were prominent among sepsis patients (Fig. 1). T cell receptor signaling pathway was decreased in 100% (6/6) of the datasets, which implies that T cell function seemed to be decreased or impaired. In network analysis, complement and coagulation cascade was prominent in sepsis survivors of GSE95233 datasets (Fig2). Complement and coagulation cascade was increased among sepsis from 83.3% (5/6) of datasets, which seemed to be a convalescence process. 

 1.1 Following Table was added in result section

Table 3. Differentially expressed gene analysis between control and sepsis followed by gene set pathway analysis 

 GSE54514a GSE57065 GSE95233 GSE64456 GSE66099 GSE72829

Patient type Adult Adult Adult Paediatric Paediatric Paediatric

Control cases (n) 18 25 22 19 49 52

Sequential selection 

of sepsis cases (n) 18 25 22 19 49 52

Differentially expressed 

Genes (n) 2180 7885 6784 3123 8229 7463

Up regulated pathway (n) 4 60 44 13 78 51

Down regulated pathway (n) 1 33 33 16 33 15

Up regulated immune

pathway C, Fc, RIG C, Fc Fc, NK, K, Toll, C K, IgA, Fc, APC, C LK, Fc, NK, B cell, Toll, K, C

Down regulated immune

pathway T cell APC, T cell, H, K APC, IgA, T cell, K, B cell IgA, APC, T cell T cell, H IgA, APC, T cell

aDifferentially expressed gene of GSE54514 was selected based on global P-value (0.05) and other datasets are based on FDR (<0.01).

APC, Antigen processing and presentation; B cell, B cell receptor signaling; C, Complement and coagulation; Fc, Fc gamma R mediated phagocytosis, Fc epsilon RI signaling; H, hematopoietic cell lineage; IgA, intestinal immune network for IgA production; K, chemokine signaling; LM, leukocyte endothelial migration; NK, NK cell mediated cytotoxicity; RIG, RIG-I like receptor signaling; Toll, Toll like receptor signaling; T cell, T cell receptor signaling.

 1.2 Following sentence was added in result section

Gene set enrichment for pathway analysis

 After differentially expressed gene (DEG) analysis, gene set enrichment for pathway analysis was performed based on KEGG pathway. Among up regulated or down regulated pathways with statistical significance, pathways associated with immune related pathway was searched. We have found that complement and coagulation cascade pathway and Fc epsilon RI signaling or Fc gamma R mediated phagocytosis pathway was increased in 66% (2/3) and 100% (3/3) datasets from adult and paediatric patients, respectively. T cell receptor signaling pathway was decreased in 100% (3/3) and 100% (3/3) datasets from adult and paediatric patients, respectively. Antigen presentation and processing was decreased in 66% (2/3) and 66% (2/3) datasets from adult and paediatric patients, respectively (Table 3). Altogether, these findings could be interpreted along with the network analysis results. In network analysis, adaptive immunity associated with CD3 and CD8 genes were prominent and isolated in sepsis patient group (Fig. 1). T cell receptor signaling pathway was decreased in 100% (6/6) of the datasets, and these data implies that T cell function seemed to be decreased or impaired. In network analysis, complement and coagulation cascade was prominent in sepsis survivors of GSE95233 datasets (Fig2). Complement and coagulation cascade was increased among sepsis in 83.3% (5/6) of datasets, and these data implies that complement and coagulation cascade seemed to be a involved in convalescence process. 

 1.3 Following supplementary figures were added in supplementary materials

S14 Fig. The gene set analysis result of GSE54514. Red cells indicate up regulated gene sets, whereas gene cells indicate down regulated gene sets.

S15 Fig. The gene set analysis result of GSE57065. Red cells indicate up regulated gene sets, whereas gene cells indicate down regulated gene sets.

S16 Fig. The gene set analysis result of GSE95233. Red cells indicate up regulated gene sets, whereas gene cells indicate down regulated gene sets.

S17 Fig. The gene set analysis result of GSE64456. Red cells indicate up regulated gene sets, whereas gene cells indicate down regulated gene sets.

S18 Fig. The gene set analysis result of GSE66099. Red cells indicate up regulated gene sets, whereas gene cells indicate down regulated gene sets.

S19 Fig. The gene set analysis result of GSE72829 Red cells indicate up regulated gene sets, whereas gene cells indicate down regulated gene sets.

 1.4 Following paragraph was added in method section

Gene set enrichment for pathway analysis

Differentially expressed gene (DEG) was performed for total transcriptome between control and patient group with equal sample number. After t-test using basic functions in R software, genes were selected based on false discovery rate of less than 0.01 (FDR<0.01) 53. Based on DEG, gene set enrichment for pathway analysis using GAGE package using R software was performed 54. GAGE package applies input data to KEGG pathway. Significant pathways were plotted for heatmap that was selected based on global P value of less than 0.05. 

2. The criteria used for defining sepsis patients are from the 1st consensus definition; we are now at the 3rd consensus definition, and thus the applicability to the clinical setting is possibly limited.

Thank you for you kind comment.

Most of the public data are studied based on sepsis-1 criteria.The definition of sepsis-1 was focused on inflammation, whereas sepsis 3 was focused on organ failure. The severe sepsis in sepsis-1 criteria are comparable to sepsis in sepsis-3 criteria. Septic shock in sepsis-1 are comparable to septic shock in sepsis-3. However, this could be limitation of this study and following sentence was added in limitation section.

 Following sentence was added in the limiation section

" Datasets are based on sepsis-1 criteria for sepsis diagnosis, which was revised to sepsis-3 criteria. This could limit the applicability to the clinical settings."

3. The datasets used involve a mix of adult and pediatric patients. There is extensive literature documenting differences in pediatric vs. adult sepsis patients with regard to multiple inflammatory and immune responses as well as outcomes.

 Thank you for your kind comments.

The actual analysis was performed based on respective datasets. Adult or paediatric group are not intermingled or analyzed together. The datasets are analyzed independently by each datasets. However, as there was error in Table1, Table 2, we corrected the error in Table 1 and Table 2 as follows. Analysis of the results were corrected according to adult and paediatric datasets, which was as follows. 

 3.1 Following paragraph was revised in result section

Network parameter analysis

 Comparison of topologic parameters revealed that the clustering coefficient in sepsis was increased 66% (2/3) of adult datasets and 100% (3/3) in paediatric datasets (Table 2). Network heterogeneity was increased in 100% (3/3) of adult datasets, but 100% (3/3) were decreased in paediatric datasets, respectively. Modularity value was decreased in 66% (2/3) of adult datasets, whereas modularity value was increased in 100% (3/3) of paediatric datasets in the sepsis group. Average degree and shortest path were increased in the sepsis group among 66% (2/3) of adult and paediatric datasets, respectively. Other parameters did not showed significant results between the healthy control group and the sepsis group (Table 2).

 3.2 Table 1 was rearranged as follows

Table 1. Baseline characteristics of recruited datasets for adult and paediatric patient

 GSE54514 GSE57065 GSE95233 GSE64456 GSE66099 GSE72829

Patient type Adult Adult Adult Paediatric Paediatric Paediatric

Age, mean (yr) a,b ≥18, 60.3 ≥18, 62 ≥18, 65 ≤2 mo, 29 d ≤10, 3.7 ≤17, 1.9 mo

Sex (male %)b 40 67.9 65 52 58 62

Nonsurvivor (%) 25.7 17.9 (28 d) NA NA 16.3 19.2

Cohort description ICU with sepsis or 

septic shock ICU with 

septic shock ICU with 

septic shock ED with sepsis

(PECARN study) PICU with sepsis or 

septic shock Discovery group 

(IRIS study)

Severity evaluation APACHEII SOFA SOFA YOSb PRISM NA

Control (n) 18 25 22 19 49 52

Sepsis or septic shock (n) 145 28 102 89 198 94

Randomized selection without 

replacement (septic shock) 18 25 - - - -

Randomized selection without 

replacement (sepsis) - - 22 18 49 52

Measured time points 5 (1, 2, 3, 4, 5 d) 3 (0, 24, 28 hr) 3 (1-2, 3-4, 7-10 d) 1 1 1

Recruited time pointsc 0-24 hr 0 - 24 hr 1-2 d NA 0 - 24 hr 5 d

Platform Illumina, GPL6947 Affymetrix, GPL570 Affymetrix, GPL570 Illumina, GPL10558 Affymetrix, GPL570 Illumina, GPL10558

Reference [11] [12] [16] [13] [14] [15]

ED, emergency department; ICU, intensive care unit; NA, data not available; hr, hour; d, day.

 3.3 Table 2 was rearranged as follows

Table 2. Network topological parameters from gene expression data from GEO dataset for adult and paediatric patient

 GSE54514 GSE57065 GSE95233 GSE95233-1 GSE64456 GSE66099 GSE72829

Patient type Adult Adult Adult Adult Paediatirc Paediatirc Paediatirc

Disese type control septic

shock control sepsis control control sepsis control control septic

shock control septic

shock control sepsis

Clustering coefficient 0.319 0.323 0.328 0.280 0.328 0.367 0.390 0.337 0.367 0.390 0.337 0.388 0.381 0.435

Network density 0.013 0.016 0.012 0.010 0.012 0.039 0.026 0.014 0.039 0.026 0.014 0.012 0.030 0.034

Network heterogeneity 1.039 1.142 1.151 1.172 1.151 1.504 1.358 1.193 1.504 1.358 1.193 1.091 1.249 1.129

Connected components 27 30 42 61 42 34 33 47 34 33 47 32 16 25

Network diameter 19 17 14 15 14 12 19 11 12 19 11 14 13 9

Network centralization 0.059 0.079 0.103 0.048 0.103 0.196 0.113 0.075 0.196 0.113 0.075 0.078 0.148 0.121

Shortest path 160514 80598 59638 17222 59638 142640 121152 11730 142640 121152 11730 36902 38092 22532

Characteristic 

path length 7.418 5.267 5.276 5.088 5.276 3.862 7.211 3.482 3.862 7.211 3.482 4.705 3.92 2.824

Average degree, <k> 5.93 7.42 5.30 3.64 5.30 17.87 11.71 4.04 17.87 11.71 4.04 4.50 8.43 10.27

Number of nodes, N 464 453 426 367 426 457 448 297 457 448 297 380 284 299

Degree correlation, μ 0.489 0.508 0.422 0.810 0.422 0.403 0.715 0.546 0.403 0.715 0.546 0.210 0.475 0.346

Modularity, Mc 0.743 0.662 0.644 0.806 0.644 0.198 0.581 0.694 0.198 0.581 0.694 0.745 0.434 0.603

 3.4 Interpretation of network topology was analyzed by adult and paediatric datasets, respectively. The Following section was corrected as follows:

In that isolated cluster, CD247 (CD3 Zeta chain), CD8A, ITK (tyrosine protein kinase ITK / TSK), LAT (linker for activation of T cells), LCK (leukocyte C-terminal Src kinase) were found in all 6 datasets. CD2, FYN (Src family tyrosine kinase), GATA3 (GATA-binding protein 3), IL7R, RASGRP1 (RAS guanyl-releasing protein 1) were all found in adult datasets, but 66% (2/3) were found in paediatric group. CBLB (E2 ubiquitin protein ligase CBL-B), CD3D, CD3G, ZAP70 (Zeta chain-associated protein kinase70) were found in 66% (2/3) of adult datasets, but all were found in paediatric datasets.

3.5 Interpretation of network parameters was analyzed by adult and paediatric datasets, respectively. The Following section was corrected as follows:

Comparison of topologic parameters revealed that the clustering coefficient in sepsis was increased 66% (2/3) of adult datasets and 100% (3/3) in paediatric datasets (Table 2). Network heterogeneity was increased in 100% (3/3) of adult datasets, but 100% (3/3) were decreased in paediatric datasets, respectively. Modularity value was decreased in 66% (2/3) of adult datasets, whereas modularity value was increased in 100% (3/3) of paediatric datasets in the sepsis group. Average degree and shortest path were increased in the sepsis group among 66% (2/3) of adult and paediatric datasets, respectively. Other parameters did not showed significant results between the healthy control group and the sepsis group (Table 2).

4. The source biological material for the transcriptomic analyses is not described. These data are could be from peripheral blood mononuclear cells or whole blood (or were they derived from other tissues?), making interpretation of this study and comparison to other studies difficult.

 Thank you for your kind comments.

We have added following sentence in the method section

 All the datasets used transcriptome derived from whole blood.

Reviewer #2: Dear Associate Editor,

The manuscript 'Gene expression networks in sepsis: A network biology approach' by Kim et al analyzes various immune system pathways in patients with sepsis compared to healthy controls using network analysis approach and show that adaptive immune cells are isolated and prominent in that network in patients with sepsis. The strength of this manuscript is that they have used diverse datasets for analysis and obtained comparable results across all the datasets. However, there are some concerns in this manuscript, which the authors have failed to address.

Major comment:

1. In the introduction, the authors nicely summarize the importance of network approach analysis in sepsis as shown by other groups, however two key elements are missing:

a. What is unique or novel about their network approach analysis compared to others and what is the rationale behind that?

b. What was their motivation to perform such an analysis in sepsis?

 Thank you for your kind comments.

We have added following sentences in introduction.

1-a. These studies constructed network based on measured immune associated molecules that could be regarded as actual observable characteristics of sepsis. Measuring molecules in multiplex methods are relatively unfeasible compared to that of gene expression.

1-b. As molecules could be analyzed independently, analysis of multiple molecules as a system was required. Analysis of these molecules via network analysis including topologic parameters and visualization resulted in informative outcome. 

1.1 a, b We have corrected introduction as follows.

 The immune response between healthy and diseased states is rarely attributed to single molecules but rather to complex inter-related molecules. As sepsis-related immune response molecules are complex, a network approach might enhance our understanding of the equilibrium of the immune response in sepsis18,19. A network approach analysis of a small group of cytokines revealed that the network constructed from sepsis had a lower network diameter and lower shortest path and characteristic path length than the control network20. These topologic features were related to decreased network function or modularity of the network. In addition, compared to that of day 1, the day 4 cytokine network was decreased in sepsis, which was in line with previous literature21. These studies constructed network based on measured immune associated molecules that could be regarded as actual observable characteristics of sepsis. As molecules could be analyzed independently, analysis of multiple molecules as a system was required. Analysis of these molecules via network analysis including topologic parameters and visualization resulted in informative outcome. However, previous studies constructed networks based on small groups of cytokine measurements that resulted in a limited size of recruited molecules. Measuring molecules in multiplex methods are relatively unfeasible compared to that of gene expression.

2. There is a confusion in the two main figures:

a. The dataset identifiers mentioned in the results section of the manuscript are contradictory to those in figures 1 and 2. It looks like the labeling in figure 1 and 2 are swapped.

b. The survivors and non-survivors (mentioned under figure 2 in the results section) have not been labelled in any of the main figures.

 Thank you for your kind comments.

We have corrected the error in the figures.

3. The discussion section is very long and the authors need to strike a balance between the results and the discussion in the way it is written. There are many elements in the discussion section that actually belongs to the results section.

 Thank you for your kind comments. We have revised discussion section as follows.

3.1 Deleted sentences in discussion section are as follows.

CD247, CD8A, ITK, LAT, and LCK were found in all 6 datasets, whereas CD2, CD3D, FYN, GATA3, IL7R, RASGRP1, ZAP70, CBLB and CD3G were found in 5 datasets.

3.2 Following discussion section are revised.

most of the genes were annotated to the T cell receptor signalling pathway, which implies that these functions were dysregulated or uncontrolled with other immune networks compared to those in the healthy normal control group. Altogether, adaptive immunity, especially T cell associated gene seemed to be suppressed or dysregulated7. 

 most of the genes were annotated to the T cell receptor signalling pathway, which implies that these functions were dysregulated or uncontrolled with other immune networks compared to those in the healthy normal control group. In addition, gene set enrichment for pathway analysis showed that T cell receptor signalling pathway was down regulated in studies all the datasets. Altogether, adaptive immunity, especially T cell associated gene seemed to be suppressed or dysregulated7. 

3.3 Following sentences are deleted.

In the healthy control group, the gene expression network was intermingled with that of innate and adaptive immune genes, whereas genes related to adaptive immunity were isolated and prominent in the sepsis group. In addition, isolated NK cell activity in the control group tended to be integrated into the sepsis group. Isolation or clustering of the network was formed by genes correlating within the cluster and not with the genes outside. Together, these genes might be associated with immune dysregulation or an abnormal host immune response4,5.

3.4 Following sentences are revised as follows

Activation of adaptive immunity is required to resolve sepsis, but the isolated and prominent adaptive immune response in this study seemed to be suppressed or dysregulated. activated without control or harmony with other networks. Complement-coagulation along with platelet-related genes seemed to be activated for convalescence. The gene cluster from nonsurvivors were similar from the genes in isolated network in sepsis. The gene cluster from nonsurvivors shared 4 genes among 5 genes from 6 datasets and 6 genes out of 9 genes from 5 data sets. The gene cluster from nonsurvivors included GATA3, which is known to be the master regulator of CD4 Th2 cell differentiation45.

-> Activation of adaptive immunity is required to resolve sepsis, but the isolated and prominent adaptive immune response in this study seemed to be suppressed or dysregulated.

3.5 Following sentences are moved from discussion to result section

The gene cluster from nonsurvivors were similar from the genes in isolated network in sepsis. shared 4 genes among 5 genes from 6 datasets and 6 genes out of 9 genes from 5 data sets. The gene cluster from nonsurvivors included GATA3, which is known to be the master regulator of CD4 Th2 cell differentiation45

3.6 Following sentence was revised as follows:

Further studies are required to understand the relation between the clustering coefficient and the inflammatory response in the immune network.

 Network analysis revealed that inflammatory process seemed to be activated in sepsis group and further studies are required to understand the relation between the clustering coefficient and the inflammatory response in the immune network. 

3.7 Following sentence was revised as follows:

Shortest path length tended to be decreased in the sepsis group, which was in line with previous literature20

 Shortest path length was decreased in 66% (2/3) of adult and paediatric datasets, respectively, which was in line with previous literature20

3.8 Following sentence was revised as follows

Modularity of a network represents a subgroup or a cluster within a network28,29. Modularity implies dense connectivity between the nodes within the same cluster but has sparse connection with other nodes outside of the cluster32,33. Modularity is expected to promote evolvability and multifunctionality and function as a driving biological process29,32. In this study, the sepsis group showed increased modularity values in 83.3% of the studied datasets. This result might be due to the prominence or isolation of the adaptive immune network. Although modularity implies the biological process of a network, the isolation and disconnection of a module with a component of a similar functional node might result in adverse reactions in the whole network.

 Modularity of a network represents a subgroup or a cluster within a network28,29. Modularity implies dense connectivity between the nodes within the same cluster but has sparse connection with other nodes outside of the cluster32,33. Modularity is expected to promote evolvability and multifunctionality and function as a driving biological process29,32.

In this study, the sepsis group in paediatric datasets showed increased modularity values in 100% of the studied datasets. On the contrary, only 33% of sepsis group in adult datasets showed increased modularity. These results are consistent with the data that sepsis group from paediatric datasets showed more significant pathways compared to that of adult group (Table 3). Although modularity implies the biological process of a network, the isolation and disconnection of a module with a component of a similar functional node might result in adverse reactions in the whole network.

3.9 Following sentence was revised as follows

Shortest path length tended to be decreased in the sepsis group, which was in line with previous literature20. This parameter, along with network diameter, is related to the functional and biological processes of a network and increases the probability of the presence of a sub-network or module with biological functions29,30. In this study, these values were decreased in the sepsis group, which might have caused decreased modularity. However, isolated and prominent module formation might have enhanced modularity values in the context of decreasing path length. It has been reported that as a network evolves, complex biological functions are executed through modules, which increases the path length or network diameter34,35. The inconsistency of the decreased shortest path and diameter and increased modularity and clustering coefficient might be an indication of immune dysfunction, which requires further study.

 Shortest path length was decreased in 66% (2/3) of adult and paediatric datasets, respectively, which was in line with previous literature20. This parameter, along with network diameter, is related to the functional and biological processes of a network and increases the probability of the presence of a sub-network or module with biological functions46,47 . It has been reported that as a network evolves, complex biological functions are executed through modules, which increases the path length or network diameter52,53. The inconsistency of the decreased shortest path and increased clustering coefficient might be an indication of immune dysfunction, which requires further study.

3.10 Following sentence was deleted:

Demographic features, especially sex and age of patients greatly affect prevalence and mortality of the sepsis. There is growing evidence that there are differences in immune system between male and female due to genetic and hormonal causes54,55. Type I interferon activity, T cell numbers and antibody responses are greater in females than in males55. The sex difference in sepsis was supported by the global statistics that the females showed higher incidence of sepsis, whereas males showed higher morality56. The sex difference could not be analyzed in this study due to the small sample sizes and further studies are required to elucidate the sex difference. 

3.11 Following sentence was deleted:

CD247, CD8A, ITK, LAT, LCK genes are included in the isolated network in sepsis derived from both age groups. Among them, LCK are intracellular signalling molecule associated with the cytoplasmic tail of CD4 and CD8A molecules.

3.11 Following sentence was deleted:

The sex difference in regard to immune response should be considered, because there is a growing evidence that the genetic and hormonal backgrounds are related with infection54,55. In this study, as the number of samples was limited, the sex difference could not be analyzed.

3.12 Following sentence was revised:

Conclusion

 Immune dysfunction might be caused by isolation and prominence of the adaptive immune response from rest of the immune network. The increased modularity and clustering coefficient imply that a modular immune response might be upregulated in the sepsis group compared to that in the normal control group, which seems to be uncontrolled or dysregulated.

 Conclusion

 Immune dysfunction might be caused by isolation and prominence of the adaptive immune response from rest of the immune network. The isolated gene cluster included T cell receptor signaling gene and that pathway was down regulated by gene set enrichment for pathway analysis. T cell signaling or T cell related functions seemed to be impaired or decreased in sepsis cases. Survivors of sepsis showed a prominent cluster of genes that was related to complement and coagulation cascade. As this pathway was up regulated in most of sepsis datasets, this pathway seemed to be related with convalescence process. 

4. The current notion is that IL-10 is an anti-inflammatory cytokine, although IL-10 has been shown to be increased amongst non-survivors (ref 28), there is no direct evidence to state that IL-10 is related to unfavorable prognosis. High IL-10 response in non-survivors could be the due to the result of a compensated anti-inflammatory response in sepsis, following the initial pro-inflammation. Therefore, the discussion on high IL-10 is related to unfavorable prognosis in sepsis without a proper evidence is an overstatement and not correct.

 Thank you for your kind comment.

We have deleted following sentences.

In addition, IL-10 was also included in that cluster; IL-10 is a cytokine that stimulates Th2 cells, and a high concentration of IL-10 along with IL-17, IL-18 and CXCL10 was related to an unfavourable prognosis in sepsis patients28.

5. The discussion about the main computational finding on the role of adaptive immune cells in sepsis identified using their network analysis and their inference in the pathophysiology of SIRS or sepsis is not clear and needs to be elucidated.

 Thank you for your kind comment.

We have performed additional analysis, which was gene set enrichment for pathway analysis (Supplementary Table S17, Supplmentary Figure S14-S19). Altogether, network analysis showed isolated component or isolated cluster of genes related to adaptive immune response. The genes in the cluster were related to adaptive immune response or T cells and the pathway analysis showed that the T cell signaling pathway was decreased in 100% (3/3) in adult and 100% (3/3) in paedicatric sepsis datasets. From these findings, T cell signaling or T cell related functions seemed to be impaired or decreased in sepsis cases. Among survivor of sepsis, complement and coagulation cascade and platelet related genes was prominent in network analysis. Gene set enrichment analysis showed that complement and coagulation cascade pathway was up regulated in 66% (2/3) in adult and 100% (3/3) in paediatric sepsis cases. From these findings, complement and coagulation pathway seemed to be associated with convalescence process during sepsis process. Clustering coefficient that was related with inflammatory process was increased in 66% (2/3) of adult and 100% (3/3) in paediatric datasets. 

5.1 Following sentence was added in discussion section.

 Altogether, network analysis showed isolated component or isolated cluster of genes related to adaptive immune response. The genes in the cluster were related to adaptive immune response or T cells and the pathway analysis showed that the T cell signaling pathway was decreased in 100% (3/3) in adult and 100% (3/3) in paedicatric sepsis datasets. From these findings, T cell signaling or T cell related functions seemed to be impaired or decreased in sepsis cases. Among survivor of sepsis, complement and coagulation cascade and platelet related genes was prominent in network analysis. Gene set enrichment analysis showed that complement and coagulation cascade pathway was up regulated in 66% (2/3) in adult and 100% (3/3) in paediatric sepsis cases. From these findings, complement and coagulation pathway seemed to be associated with convalescence process during sepsis process. Clustering coefficient that was related with inflammatory process was increased in 66% (2/3) of adult and 100% (3/3) in paediatric datasets. 

5.2 Following table below from gene set enrichment for pathway analysis was added as a Supplementary Table S17

Minor comments:

1. The authors should stick to using the abbreviations instead of full form after using it for the first time and have to be consistent throughout the manuscript.

Example: GEO - line 86, KEGG - line 150, GO - line 166

 Thank you for your kind comment.

We have corrected the errors as commented.

2. Are there any references for the rationale behind using such platforms? Lines 106 and 107.

 Thank you for your kind comment.

Following articles are cited. 

22. Ritchie M, Dunning M, Smith M, Shi W, Lynch A. BeadArray expression analysis using bioconductor. PLoS Comp Biol 2011; 7:e1002276. https://doi.org/10.1371/journal.pcbi.1002276.

23. Jaksik R, Iwanaszko M, Rzeszowska-Wolny R, Kimmel M. Microarray experiments and factors which affect their reliability. Biol Direct 2015; 10:46 10.1186/s13062-015-0077-2.

3. What does '(n)' in the table 1 indicate?

 Thank you for your kind comment. 

We are sorry for confusion . The measured frequency or frequency of blood drawn are described. For example, GSE54514 datasets showed that for eacg patients, as much as 5 time of blood dare drawn at day 1, day 2, day3, day4, and day 5 after admission. Then experimented as much as 5 time per patients. 5 (1, 2, 3, 4, 5 d)

 We have changed the Table 1 as follows.

Measured freqeuncy, frequency (time points) 5 (1, 2, 3, 4, 5 d) 3 (0, 24, 28 hr) 3 (1-2, 3-4, 7-10 d) 1 1 1

4. Why are the figure legends not listed separately and written along with the text?

 Thank you for your kind comment. 

Submission guideline states as follows.

"Figure captions must be inserted in the text of the manuscript, immediately following the paragraph in which the figure is first cited (read order). Do not include captions as part of the figure files themselves or submit them in a separate document"

5. Maintain a similar font and size throughout the manuscript, line 298 is different.

 Thank you for your kind comment. 

We have corrected the error. Font size was changed to 12 and Times New Roman was used for the font.

Additional revision

Following Supplementary Table S17 was added, which was the result of gene set enrichment for pathway analysis. Gene set enrichment for pathway analysis data was summarized in Table 3. 

Supplmentary Table S17. 

GSE54514 p.geomean stat.mean p.val q.val set.size

hsa04142 Lysosome 0.025847966 1.269518545 5.09E-08 8.20E-06 114

hsa00970 Aminoacyl-tRNA biosynthesis 0.088028345 0.881659727 0.000130305 0.008555202 40

hsa04141 Protein processing in endoplasmic reticulum 0.108398782 0.852836674 0.000159414 0.008555202 152

hsa00600 Sphingolipid metabolism 0.183824309 0.745837522 0.000888854 0.035776374 33

hsa00500 Starch and sucrose metabolism 0.174616207 0.592419696 0.00683581 0.22011307 32

hsa04260 Cardiac muscle contraction 0.150475266 0.517270132 0.015534859 0.353170066 49

hsa00670 One carbon pool by folate 0.24989082 0.511164476 0.01766595 0.353170066 15

hsa03030 DNA replication 0.140145009 0.502213408 0.019475018 0.353170066 36

hsa00520 Amino sugar and nucleotide sugar metabolism 0.185600438 0.492848731 0.019742426 0.353170066 42

hsa04810 Regulation of actin cytoskeleton 0.142203717 0.470551959 0.023133818 0.372454467 168

hsa04622 RIG-I-like receptor signaling pathway 0.198576053 0.463690703 0.025639176 0.375264301 58

hsa00310 Lysine degradation 0.258921614 0.443101954 0.031635146 0.424438213 37

hsa00561 Glycerolipid metabolism 0.292506444 0.427968945 0.035771277 0.428628393 39

hsa00983 Drug metabolism - other enzymes 0.25490034 0.419779579 0.039240986 0.428628393 27

hsa00564 Glycerophospholipid metabolism 0.255687907 0.414859748 0.040160748 0.428628393 63

hsa04973 Carbohydrate digestion and absorption 0.269226767 0.399200271 0.046898393 0.428628393 31

hsa00630 Glyoxylate and dicarboxylate metabolism 0.258395943 0.407462628 0.047746415 0.428628393 16

hsa02010 ABC transporters 0.255973642 0.394894213 0.049534371 0.428628393 32

hsa03060 Protein export 0.201584586 0.396801811 0.051659627 0.428628393 23

hsa00510 N-Glycan biosynthesis 0.258174577 0.38240521 0.054661249 0.428628393 44

hsa04114 Oocyte meiosis 0.25415967 0.370225054 0.05878054 0.428628393 97

hsa03410 Base excision repair 0.26846419 0.370962959 0.06076491 0.428628393 31

hsa00512 Mucin type O-Glycan biosynthesis 0.297860294 0.371924262 0.061232628 0.428628393 18

hsa04962 Vasopressin-regulated water reabsorption 0.307990132 0.356276752 0.066901165 0.448795313 37

hsa04666 Fc gamma R-mediated phagocytosis 0.149263639 0.326641851 0.083783981 0.484269587 85

hsa04670 Leukocyte transendothelial migration 0.179800404 0.325133026 0.084901015 0.484269587 91

hsa00250 Alanine, aspartate and glutamate metabolism 0.308043361 0.327973968 0.085817892 0.484269587 21

hsa04020 Calcium signaling pathway 0.30905421 0.320665521 0.087228474 0.484269587 121

hsa00534 Glycosaminoglycan biosynthesis - heparan sulfate 0.327068995 0.323210599 0.089572826 0.484269587 17

hsa04914 Progesterone-mediated oocyte maturation 0.318424759 0.316995636 0.090236569 0.484269587 74

hsa00531 Glycosaminoglycan degradation 0.28104011 0.312304755 0.097423406 0.495162094 17

hsa00052 Galactose metabolism 0.301508758 0.299776925 0.103657516 0.495162094 23

hsa00100 Steroid biosynthesis 0.309712284 0.303208041 0.10394196 0.495162094 17

hsa04722 Neurotrophin signaling pathway 0.209032778 0.296981402 0.104568392 0.495162094 118

hsa00040 Pentose and glucuronate interconversions 0.339423254 0.297522093 0.107723493 0.495528066 17

hsa04614 Renin-angiotensin system 0.299906414 0.2965293 0.116933139 0.522950984 10

hsa00010 Glycolysis / Gluconeogenesis 0.20023476 0.277225272 0.124573091 0.542061287 53

hsa00790 Folate biosynthesis 0.341498338 0.262181962 0.140744805 0.568424515 10

hsa00640 Propanoate metabolism 0.264798564 0.258017879 0.142572288 0.568424515 29

hsa00770 Pantothenate and CoA biosynthesis 0.290883135 0.2552472 0.146246621 0.568424515 14

hsa04110 Cell cycle 0.217646458 0.248708696 0.147876265 0.568424515 113

hsa04972 Pancreatic secretion 0.338662349 0.247249009 0.148284656 0.568424515 62

hsa00982 Drug metabolism - cytochrome P450 0.367722445 0.238589699 0.157562697 0.58994405 35

hsa04971 Gastric acid secretion 0.3592355 0.233274674 0.162455793 0.594440515 53

hsa00980 Metabolism of xenobiotics by cytochrome P450 0.332875037 0.226637555 0.170044422 0.595901983 38

hsa00140 Steroid hormone biosynthesis 0.354305041 0.22741378 0.170257709 0.595901983 27

hsa00900 Terpenoid backbone biosynthesis 0.332436368 0.218545769 0.184698046 0.630739186 12

hsa00190 Oxidative phosphorylation 0.037330963 0.208137994 0.191092518 0.630739186 98

hsa04910 Insulin signaling pathway 0.245798672 0.205116192 0.1919641 0.630739186 122

hsa04070 Phosphatidylinositol signaling system 0.334222679 0.192350212 0.208023871 0.642372746 63

hsa04977 Vitamin digestion and absorption 0.384343117 0.195255211 0.208279084 0.642372746 14

hsa04912 GnRH signaling pathway 0.342673996 0.189741884 0.211326016 0.642372746 77

hsa00260 Glycine, serine and threonine metabolism 0.321173596 0.19479429 0.21146432 0.642372746 22

hsa04975 Fat digestion and absorption 0.369961393 0.179142864 0.22581784 0.673271708 24

hsa04510 Focal adhesion 0.278160495 0.173626264 0.230738055 0.675433216 152

hsa00562 Inositol phosphate metabolism 0.345858678 0.164064696 0.244301366 0.702366426 49

hsa04120 Ubiquitin mediated proteolysis 0.345414226 0.135754803 0.283040175 0.749230808 124

hsa00565 Ether lipid metabolism 0.393313072 0.13629968 0.283349434 0.749230808 26

hsa00511 Other glycan degradation 0.34406099 0.132503787 0.288935834 0.749230808 15

hsa00360 Phenylalanine metabolism 0.391046096 0.133360583 0.291832263 0.749230808 11

hsa04964 Proximal tubule bicarbonate reclamation 0.37667497 0.132046378 0.293578294 0.749230808 13

hsa04974 Protein digestion and absorption 0.423776406 0.127078542 0.295942314 0.749230808 46

hsa04976 Bile secretion 0.378893273 0.126104602 0.297061262 0.749230808 41

hsa00592 alpha-Linolenic acid metabolism 0.41786042 0.123279378 0.30524373 0.749230808 10

hsa00270 Cysteine and methionine metabolism 0.318554397 0.120569815 0.309732703 0.749230808 29

hsa03320 PPAR signaling pathway 0.386684884 0.116911974 0.310656418 0.749230808 49

hsa00240 Pyrimidine metabolism 0.363458028 0.11620759 0.311791703 0.749230808 86

hsa01040 Biosynthesis of unsaturated fatty acids 0.387275259 0.113252693 0.318749406 0.754686093 17

hsa04210 Apoptosis 0.305235308 0.098878697 0.338606349 0.776437433 82

hsa04610 Complement and coagulation cascades 0.353012098 0.098902363 0.340223888 0.776437433 43

hsa04662 B cell receptor signaling pathway 0.260627497 0.091463676 0.348141754 0.776437433 71

hsa04742 Taste transduction 0.419851888 0.092921726 0.348238904 0.776437433 23

hsa04740 Olfactory transduction 0.352684989 0.088588164 0.35389701 0.776437433 91

hsa00380 Tryptophan metabolism 0.391587333 0.08729509 0.356871863 0.776437433 32

hsa04920 Adipocytokine signaling pathway 0.371053644 0.079635611 0.368659921 0.784034854 60

hsa00053 Ascorbate and aldarate metabolism 0.414632821 0.074706156 0.379728253 0.784034854 12

hsa04146 Peroxisome 0.361172032 0.070587225 0.382781845 0.784034854 61

hsa03450 Non-homologous end-joining 0.435855274 0.071922973 0.384160447 0.784034854 10

hsa04115 p53 signaling pathway 0.391925676 0.069158745 0.384712754 0.784034854 58

hsa00620 Pyruvate metabolism 0.321389717 0.064295199 0.394042236 0.786704121 36

hsa04010 MAPK signaling pathway 0.313311625 0.062364703 0.395795241 0.786704121 206

hsa00290 Valine, leucine and isoleucine biosynthesis 0.334268388 0.061382719 0.401453979 0.788220618 10

hsa04520 Adherens junction 0.324247483 0.048347278 0.416929952 0.793369782 57

hsa04310 Wnt signaling pathway 0.435163792 0.048277858 0.419061166 0.793369782 113

hsa00051 Fructose and mannose metabolism 0.341772085 0.043156679 0.425408167 0.793369782 31

hsa00591 Linoleic acid metabolism 0.446939775 0.042645284 0.429284738 0.793369782 13

hsa00280 Valine, leucine and isoleucine degradation 0.317635293 0.041651145 0.431136109 0.793369782 38

hsa03015 mRNA surveillance pathway 0.431686244 0.039670161 0.433643111 0.793369782 71

hsa03018 RNA degradation 0.346366886 0.032491197 0.446251782 0.807264459 66

hsa04144 Endocytosis 0.333055224 0.026846334 0.454833815 0.808758859 166

hsa00030 Pentose phosphate pathway 0.317824725 0.0257338 0.457249015 0.808758859 24

hsa04970 Salivary secretion 0.416543972 0.022445557 0.46214792 0.808758859 53

hsa03440 Homologous recombination 0.425157786 0.012762907 0.478044595 0.827582578 24

hsa04621 NOD-like receptor signaling pathway 0.277751297 0.010179839 0.48881845 0.832685052 53

hsa04960 Aldosterone-regulated sodium reabsorption 0.437266196 0.00515905 0.4913359 0.832685052 28

hsa00603 Glycosphingolipid biosynthesis - globo series 0.449189859 -0.006807241 0.508352064 0.84747946 10

hsa03050 Proteasome 0.238915289 -0.010484047 0.512024664 0.84747946 41

hsa04340 Hedgehog signaling pathway 0.461833673 -0.009327661 0.515857062 0.84747946 32

hsa00480 Glutathione metabolism 0.384508884 -0.017753272 0.530419193 0.857959851 41

hsa00830 Retinol metabolism 0.474820452 -0.020041995 0.532894318 0.857959851 28

hsa04145 Phagosome 0.306430871 -0.025812974 0.543588662 0.862097576 131

hsa00601 Glycosphingolipid biosynthesis - lacto and neolacto series 0.431939299 -0.027274556 0.54617362 0.862097576 17

hsa04270 Vascular smooth muscle contraction 0.418991488 -0.032770335 0.554027095 0.865487504 87

hsa04916 Melanogenesis 0.438601359 -0.035310316 0.559072673 0.865487504 72

hsa00330 Arginine and proline metabolism 0.387194909 -0.038035297 0.565743481 0.867473338 35

hsa00350 Tyrosine metabolism 0.451928409 -0.047513912 0.578971415 0.871664378 25

hsa00740 Riboflavin metabolism 0.461009902 -0.051380624 0.582426007 0.871664378 10

hsa00514 Other types of O-glycan biosynthesis 0.46499724 -0.050843399 0.584718961 0.871664378 29

hsa00910 Nitrogen metabolism 0.389061199 -0.056243514 0.590636298 0.872407743 14

hsa00450 Selenocompound metabolism 0.469366125 -0.059659743 0.598334181 0.875743665 14

hsa04664 Fc epsilon RI signaling pathway 0.331403961 -0.067141326 0.606750669 0.878715972 64

hsa04630 Jak-STAT signaling pathway 0.304332486 -0.066561794 0.611280676 0.878715972 112

hsa04512 ECM-receptor interaction 0.311105659 -0.074953004 0.625090399 0.890615525 51

hsa03430 Mismatch repair 0.410418436 -0.081333155 0.633964455 0.895335765 23

hsa04730 Long-term depression 0.448465686 -0.087380224 0.643065335 0.899010592 48

hsa00410 beta-Alanine metabolism 0.429205235 -0.093451249 0.650230831 0.899010592 17

hsa00071 Fatty acid metabolism 0.427905821 -0.097695646 0.658523906 0.899010592 33

hsa04150 mTOR signaling pathway 0.448876505 -0.097928081 0.658902173 0.899010592 45

hsa04320 Dorso-ventral axis formation 0.450974382 -0.112685352 0.680401978 0.920543853 17

hsa00340 Histidine metabolism 0.420885104 -0.124974453 0.699900186 0.939032749 20

hsa04710 Circadian rhythm - mammal 0.517842981 -0.1309111 0.708518549 0.942739557 20

hsa00920 Sulfur metabolism 0.463099405 -0.146929292 0.725639974 0.949930389 10

hsa00650 Butanoate metabolism 0.462646611 -0.14474415 0.725723216 0.949930389 20

hsa00860 Porphyrin and chlorophyll metabolism 0.352093404 -0.166428259 0.75483994 0.963093999 27

hsa00020 Citrate cycle (TCA cycle) 0.345764883 -0.170516025 0.758009742 0.963093999 27

hsa03420 Nucleotide excision repair 0.438925925 -0.170040315 0.762146338 0.963093999 40

hsa04140 Regulation of autophagy 0.501766316 -0.174275298 0.766671141 0.963093999 24

hsa04380 Osteoclast differentiation 0.148156136 -0.189624873 0.786435035 0.963093999 112

hsa00590 Arachidonic acid metabolism 0.503617728 -0.197975009 0.797417779 0.963093999 35

hsa00533 Glycosaminoglycan biosynthesis - keratan sulfate 0.532138836 -0.200390682 0.797795062 0.963093999 12

hsa03008 Ribosome biogenesis in eukaryotes 0.333180871 -0.200270704 0.798234973 0.963093999 65

hsa00760 Nicotinate and nicotinamide metabolism 0.528910076 -0.206511288 0.806064252 0.963093999 20

hsa04623 Cytosolic DNA-sensing pathway 0.437963673 -0.20638293 0.806666663 0.963093999 41

hsa04540 Gap junction 0.441976755 -0.206664205 0.807835502 0.963093999 71

hsa04530 Tight junction 0.44193829 -0.208545097 0.809906484 0.963093999 99

hsa04720 Long-term potentiation 0.490372452 -0.21137621 0.813545241 0.963093999 57

hsa04744 Phototransduction 0.55914291 -0.236412336 0.838759444 0.966055827 18

hsa03040 Spliceosome 0.304296201 -0.239609263 0.842039058 0.966055827 122

hsa00230 Purine metabolism 0.46796738 -0.238557953 0.843705602 0.966055827 136

hsa04330 Notch signaling pathway 0.46873001 -0.241488559 0.844589685 0.966055827 42

hsa04130 SNARE interactions in vesicular transport 0.509554977 -0.243339648 0.846048892 0.966055827 31

hsa00563 Glycosylphosphatidylinositol(GPI)-anchor biosynthesis 0.56895764 -0.251407671 0.854575085 0.968919639 21

hsa00532 Glycosaminoglycan biosynthesis - chondroitin sulfate 0.502124633 -0.262852465 0.862913279 0.971531734 19

hsa03013 RNA transport 0.295742887 -0.285477866 0.884918199 0.989387709 131

hsa04360 Axon guidance 0.5175373 -0.304991628 0.900993629 0.992123539 89

hsa00604 Glycosphingolipid biosynthesis - ganglio series 0.593477566 -0.31452579 0.904852663 0.992123539 12

hsa04672 Intestinal immune network for IgA production 0.21337552 -0.31147188 0.905851927 0.992123539 39

hsa04370 VEGF signaling pathway 0.533095949 -0.347139807 0.927802579 1 58

hsa04966 Collecting duct acid secretion 0.54307371 -0.35534608 0.930671803 1 19

hsa04350 TGF-beta signaling pathway 0.566615481 -0.389158586 0.949727553 1 64

hsa03022 Basal transcription factors 0.585770861 -0.395426272 0.951033122 1 30

hsa04012 ErbB signaling pathway 0.554808925 -0.400498237 0.95451798 1 76

hsa04640 Hematopoietic cell lineage 0.399662255 -0.467402264 0.975061345 1 76

hsa04620 Toll-like receptor signaling pathway 0.368806166 -0.475967315 0.976865581 1 82

hsa04062 Chemokine signaling pathway 0.344192063 -0.474163521 0.977260148 1 150

hsa04612 Antigen processing and presentation 0.214081952 -0.481110381 0.977825928 1 69

hsa03020 RNA polymerase 0.620754028 -0.494288871 0.980686154 1 24

hsa04514 Cell adhesion molecules (CAMs) 0.255167259 -0.610800137 0.994744854 1 102

hsa04650 Natural killer cell mediated cytotoxicity 0.472831909 -0.695347858 0.998274891 1 112

hsa04660 T cell receptor signaling pathway 0.667202143 -0.862374107 0.999861365 1 96

hsa03010 Ribosome 0.002095243 -1.890710517 1 1 88

hsa00061 Fatty acid biosynthesis NA NA NA NA 6

hsa00072 Synthesis and degradation of ketone bodies NA NA NA NA 7

hsa00120 Primary bile acid biosynthesis NA NA NA NA 9

hsa00130 Ubiquinone and other terpenoid-quinone biosynthesis NA NA NA NA 5

hsa00232 Caffeine metabolism NA NA NA NA 1

hsa00300 Lysine biosynthesis NA NA NA NA 3

hsa00400 Phenylalanine, tyrosine and tryptophan biosynthesis NA NA NA NA 3

hsa00430 Taurine and hypotaurine metabolism NA NA NA NA 6

hsa00460 Cyanoamino acid metabolism NA NA NA NA 4

hsa00471 D-Glutamine and D-glutamate metabolism NA NA NA NA 2

hsa00472 D-Arginine and D-ornithine metabolism NA NA NA NA 0

hsa00730 Thiamine metabolism NA NA NA NA 3

hsa00750 Vitamin B6 metabolism NA NA NA NA 5

hsa00780 Biotin metabolism NA NA NA NA 2

hsa00785 Lipoic acid metabolism NA NA NA NA 3

hsa04122 Sulfur relay system NA NA NA NA 9

hsa04660 T cell receptor signaling pathway 0.136214253 -0.862374107 0.000138635 0.022320274 96

hsa04650 Natural killer cell mediated cytotoxicity 0.116909546 -0.695347858 0.001725109 0.138871256 112

hsa04514 Cell adhesion molecules (CAMs) 0.069023632 -0.610800137 0.005255146 0.282026165 102

hsa03020 RNA polymerase 0.272268632 -0.494288871 0.019313846 0.501890441 24

hsa04612 Antigen processing and presentation 0.081538447 -0.481110381 0.022174072 0.501890441 69

hsa04062 Chemokine signaling pathway 0.135205504 -0.474163521 0.022739852 0.501890441 150

hsa04620 Toll-like receptor signaling pathway 0.13883093 -0.475967315 0.023134419 0.501890441 82

hsa04640 Hematopoietic cell lineage 0.165285289 -0.467402264 0.024938655 0.501890441 76

hsa04012 ErbB signaling pathway 0.277142133 -0.400498237 0.04548202 0.735805808 76

hsa03022 Basal transcription factors 0.298484727 -0.395426272 0.048966878 0.735805808 30

hsa04350 TGF-beta signaling pathway 0.292048704 -0.389158586 0.050272447 0.735805808 64

hsa04966 Collecting duct acid secretion 0.303544751 -0.35534608 0.069328197 0.894137288 19

hsa04370 VEGF signaling pathway 0.286561239 -0.347139807 0.072197421 0.894137288 58

hsa04672 Intestinal immune network for IgA production 0.123902817 -0.31147188 0.094148073 0.99625161 39

hsa00604 Glycosphingolipid biosynthesis - ganglio series 0.358452847 -0.31452579 0.095147337 0.99625161 12

hsa04360 Axon guidance 0.301663998 -0.304991628 0.099006371 0.99625161 89

hsa03013 RNA transport 0.159966018 -0.285477866 0.115081801 1 131

hsa00532 Glycosaminoglycan biosynthesis - chondroitin sulfate 0.323152217 -0.262852465 0.137086721 1 19

hsa00563 Glycosylphosphatidylinositol(GPI)-anchor biosynthesis 0.379562063 -0.251407671 0.145424915 1 21

hsa04130 SNARE interactions in vesicular transport 0.33600731 -0.243339648 0.153951108 1 31

hsa04330 Notch signaling pathway 0.306869511 -0.241488559 0.155410315 1 42

hsa00230 Purine metabolism 0.31234081 -0.238557953 0.156294398 1 136

hsa03040 Spliceosome 0.173578172 -0.239609263 0.157960942 1 122

hsa04744 Phototransduction 0.381424186 -0.236412336 0.161240556 1 18

hsa04720 Long-term potentiation 0.341097583 -0.21137621 0.186454759 1 57

hsa04530 Tight junction 0.294441545 -0.208545097 0.190093516 1 99

hsa04540 Gap junction 0.303891541 -0.206664205 0.192164498 1 71

hsa04623 Cytosolic DNA-sensing pathway 0.304797059 -0.20638293 0.193333337 1 41

hsa00760 Nicotinate and nicotinamide metabolism 0.376359987 -0.206511288 0.193935748 1 20

hsa03008 Ribosome biogenesis in eukaryotes 0.221009065 -0.200270704 0.201765027 1 65

hsa00533 Glycosaminoglycan biosynthesis - keratan sulfate 0.386003068 -0.200390682 0.202204938 1 12

hsa00590 Arachidonic acid metabolism 0.361444745 -0.197975009 0.202582221 1 35

hsa04380 Osteoclast differentiation 0.098870485 -0.189624873 0.213564965 1 112

hsa04140 Regulation of autophagy 0.373873992 -0.174275298 0.233328859 1 24

hsa03420 Nucleotide excision repair 0.326468419 -0.170040315 0.237853662 1 40

hsa00020 Citrate cycle (TCA cycle) 0.2497233 -0.170516025 0.241990258 1 27

hsa00860 Porphyrin and chlorophyll metabolism 0.26448502 -0.166428259 0.24516006 1 27

hsa00650 Butanoate metabolism 0.362321227 -0.14474415 0.274276784 1 20

hsa00920 Sulfur metabolism 0.364393515 -0.146929292 0.274360026 1 10

hsa04710 Circadian rhythm - mammal 0.418319238 -0.1309111 0.291481451 1 20

hsa00340 Histidine metabolism 0.342161218 -0.124974453 0.300099814 1 20

hsa04320 Dorso-ventral axis formation 0.373595592 -0.112685352 0.319598022 1 17

hsa04150 mTOR signaling pathway 0.374019224 -0.097928081 0.341097827 1 45

hsa00071 Fatty acid metabolism 0.360393804 -0.097695646 0.341476094 1 33

hsa00410 beta-Alanine metabolism 0.365870176 -0.093451249 0.349769169 1 17

hsa04730 Long-term depression 0.382943389 -0.087380224 0.356934665 1 48

hsa03430 Mismatch repair 0.361918324 -0.081333155 0.366035545 1 23

hsa04512 ECM-receptor interaction 0.277348567 -0.074953004 0.374909601 1 51

hsa04630 Jak-STAT signaling pathway 0.275463443 -0.066561794 0.388719324 1 112

hsa04664 Fc epsilon RI signaling pathway 0.269755422 -0.067141326 0.393249331 1 64

hsa00450 Selenocompound metabolism 0.425995821 -0.059659743 0.401665819 1 14

hsa00910 Nitrogen metabolism 0.354601714 -0.056243514 0.409363702 1 14

hsa00514 Other types of O-glycan biosynthesis 0.427357067 -0.050843399 0.415281039 1 29

hsa00740 Riboflavin metabolism 0.423313094 -0.051380624 0.417573993 1 10

hsa00350 Tyrosine metabolism 0.4176519 -0.047513912 0.421028585 1 25

hsa00330 Arginine and proline metabolism 0.369158011 -0.038035297 0.434256519 1 35

hsa04916 Melanogenesis 0.411020347 -0.035310316 0.440927327 1 72

hsa04270 Vascular smooth muscle contraction 0.385386751 -0.032770335 0.445972905 1 87

hsa00601 Glycosphingolipid biosynthesis - lacto and neolacto series 0.413805829 -0.027274556 0.45382638 1 17

hsa04145 Phagosome 0.295731261 -0.025812974 0.456411338 1 131

hsa00830 Retinol metabolism 0.457726717 -0.020041995 0.467105682 1 28

hsa00480 Glutathione metabolism 0.375752091 -0.017753272 0.469580807 1 41

hsa04340 Hedgehog signaling pathway 0.455113761 -0.009327661 0.484142938 1 32

hsa03050 Proteasome 0.222295647 -0.010484047 0.487975336 1 41

hsa00603 Glycosphingolipid biosynthesis - globo series 0.441800324 -0.006807241 0.491647936 1 10

hsa04960 Aldosterone-regulated sodium reabsorption 0.440602431 0.00515905 0.5086641 1 28

hsa04621 NOD-like receptor signaling pathway 0.304210931 0.010179839 0.51118155 1 53

hsa03440 Homologous recombination 0.432630488 0.012762907 0.521955405 1 24

hsa04970 Salivary secretion 0.432697161 0.022445557 0.53785208 1 53

hsa00030 Pentose phosphate pathway 0.330594907 0.0257338 0.542750985 1 24

hsa04144 Endocytosis 0.350688271 0.026846334 0.545166185 1 166

hsa03018 RNA degradation 0.371059579 0.032491197 0.553748218 1 66

hsa03015 mRNA surveillance pathway 0.463195069 0.039670161 0.566356889 1 71

hsa00280 Valine, leucine and isoleucine degradation 0.342972261 0.041651145 0.568863891 1 38

hsa00591 Linoleic acid metabolism 0.47793164 0.042645284 0.570715262 1 13

hsa00051 Fructose and mannose metabolism 0.358838557 0.043156679 0.574591833 1 31

hsa04310 Wnt signaling pathway 0.471705527 0.048277858 0.580938834 1 113

hsa04520 Adherens junction 0.339189913 0.048347278 0.583070048 1 57

hsa00290 Valine, leucine and isoleucine biosynthesis 0.372969854 0.061382719 0.598546021 1 10

hsa04010 MAPK signaling pathway 0.34800637 0.062364703 0.604204759 1 206

hsa00620 Pyruvate metabolism 0.359944739 0.064295199 0.605957764 1 36

hsa04115 p53 signaling pathway 0.436600315 0.069158745 0.615287246 1 58

hsa03450 Non-homologous end-joining 0.489929417 0.071922973 0.615839553 1 10

hsa04146 Peroxisome 0.405958937 0.070587225 0.617218155 1 61

hsa00053 Ascorbate and aldarate metabolism 0.468576595 0.074706156 0.620271747 1 12

hsa04920 Adipocytokine signaling pathway 0.426384472 0.079635611 0.631340079 1 60

hsa00380 Tryptophan metabolism 0.452409033 0.08729509 0.643128137 1 32

hsa04740 Olfactory transduction 0.408596481 0.088588164 0.64610299 1 91

hsa04742 Taste transduction 0.486023644 0.092921726 0.651761096 1 23

hsa04662 B cell receptor signaling pathway 0.294646319 0.091463676 0.651858246 1 71

hsa04610 Complement and coagulation cascades 0.424788858 0.098902363 0.659776112 1 43

hsa04210 Apoptosis 0.365273768 0.098878697 0.661393651 1 82

hsa01040 Biosynthesis of unsaturated fatty acids 0.467302308 0.113252693 0.681250594 1 17

hsa00240 Pyrimidine metabolism 0.444599654 0.11620759 0.688208297 1 86

hsa03320 PPAR signaling pathway 0.466931096 0.116911974 0.689343582 1 49

hsa00270 Cysteine and methionine metabolism 0.401737471 0.120569815 0.690267297 1 29

hsa00592 alpha-Linolenic acid metabolism 0.508458368 0.123279378 0.69475627 1 10

hsa04976 Bile secretion 0.464640387 0.126104602 0.702938738 1 41

hsa04974 Protein digestion and absorption 0.522478806 0.127078542 0.704057686 1 46

hsa04964 Proximal tubule bicarbonate reclamation 0.468711064 0.132046378 0.706421706 1 13

hsa00360 Phenylalanine metabolism 0.483639952 0.133360583 0.708167737 1 11

hsa00511 Other glycan degradation 0.42424033 0.132503787 0.711064166 1 15

hsa00565 Ether lipid metabolism 0.490879538 0.13629968 0.716650566 1 26

hsa04120 Ubiquitin mediated proteolysis 0.439426877 0.135754803 0.716959825 1 124

hsa00562 Inositol phosphate metabolism 0.452298465 0.164064696 0.755698634 1 49

hsa04510 Focal adhesion 0.364179241 0.173626264 0.769261945 1 152

hsa04975 Fat digestion and absorption 0.496570773 0.179142864 0.77418216 1 24

hsa00260 Glycine, serine and threonine metabolism 0.45311236 0.19479429 0.78853568 1 22

hsa04912 GnRH signaling pathway 0.47169064 0.189741884 0.788673984 1 77

hsa04977 Vitamin digestion and absorption 0.527591748 0.195255211 0.791720916 1 14

hsa04070 Phosphatidylinositol signaling system 0.457367461 0.192350212 0.791976129 1 63

hsa04910 Insulin signaling pathway 0.334478863 0.205116192 0.8080359 1 122

hsa00190 Oxidative phosphorylation 0.060528215 0.208137994 0.808907445 1 98

hsa00900 Terpenoid backbone biosynthesis 0.478282933 0.218545769 0.815301954 1 12

hsa00140 Steroid hormone biosynthesis 0.518148387 0.22741378 0.829742291 1 27

hsa00980 Metabolism of xenobiotics by cytochrome P450 0.485277778 0.226637555 0.829955578 1 38

hsa04971 Gastric acid secretion 0.530311028 0.233274674 0.837544207 1 53

hsa00982 Drug metabolism - cytochrome P450 0.545327016 0.238589699 0.842437303 1 35

hsa04972 Pancreatic secretion 0.513530861 0.247249009 0.851715344 1 62

hsa04110 Cell cycle 0.361281654 0.248708696 0.852123735 1 113

hsa00770 Pantothenate and CoA biosynthesis 0.444532334 0.2552472 0.853753379 1 14

hsa00640 Propanoate metabolism 0.425501602 0.258017879 0.857427712 1 29

hsa00790 Folate biosynthesis 0.524051286 0.262181962 0.859255195 1 10

hsa00010 Glycolysis / Gluconeogenesis 0.346032518 0.277225272 0.875426909 1 53

hsa04614 Renin-angiotensin system 0.487961985 0.2965293 0.883066861 1 10

hsa00040 Pentose and glucuronate interconversions 0.553628545 0.297522093 0.892276507 1 17

hsa04722 Neurotrophin signaling pathway 0.349423356 0.296981402 0.895431608 1 118

hsa00100 Steroid biosynthesis 0.513321879 0.303208041 0.89605804 1 17

hsa00052 Galactose metabolism 0.488136204 0.299776925 0.896342484 1 23

hsa00531 Glycosaminoglycan degradation 0.474554399 0.312304755 0.902576594 1 17

hsa04914 Progesterone-mediated oocyte maturation 0.542899538 0.316995636 0.909763431 1 74

hsa00534 Glycosaminoglycan biosynthesis - heparan sulfate 0.558033864 0.323210599 0.910427174 1 17

hsa04020 Calcium signaling pathway 0.523004003 0.320665521 0.912771526 1 121

hsa00250 Alanine, aspartate and glutamate metabolism 0.534890255 0.327973968 0.914182108 1 21

hsa04670 Leukocyte transendothelial migration 0.317449549 0.325133026 0.915098985 1 91

hsa04666 Fc gamma R-mediated phagocytosis 0.26512417 0.326641851 0.916216019 1 85

hsa04962 Vasopressin-regulated water reabsorption 0.558749396 0.356276752 0.933098835 1 37

hsa00512 Mucin type O-Glycan biosynthesis 0.551808658 0.371924262 0.938767372 1 18

hsa03410 Base excision repair 0.515470076 0.370962959 0.93923509 1 31

hsa04114 Oocyte meiosis 0.479264611 0.370225054 0.94121946 1 97

hsa00510 N-Glycan biosynthesis 0.514702769 0.38240521 0.945338751 1 44

hsa03060 Protein export 0.417899291 0.396801811 0.948340373 1 23

hsa02010 ABC transporters 0.513031406 0.394894213 0.950465629 1 32

hsa00630 Glyoxylate and dicarboxylate metabolism 0.52364952 0.407462628 0.952253585 1 16

hsa04973 Carbohydrate digestion and absorption 0.524658775 0.399200271 0.953101607 1 31

hsa00564 Glycerophospholipid metabolism 0.525247324 0.414859748 0.959839252 1 63

hsa00983 Drug metabolism - other enzymes 0.514204782 0.419779579 0.960759014 1 27

hsa00561 Glycerolipid metabolism 0.596324792 0.427968945 0.964228723 1 39

hsa00310 Lysine degradation 0.558348525 0.443101954 0.968364854 1 37

hsa04622 RIG-I-like receptor signaling pathway 0.456901567 0.463690703 0.974360824 1 58

hsa04810 Regulation of actin cytoskeleton 0.319966449 0.470551959 0.976866182 1 168

hsa00520 Amino sugar and nucleotide sugar metabolism 0.458192385 0.492848731 0.980257574 1 42

hsa03030 DNA replication 0.38849312 0.502213408 0.980524982 1 36

hsa00670 One carbon pool by folate 0.59007797 0.511164476 0.98233405 1 15

hsa04260 Cardiac muscle contraction 0.410814559 0.517270132 0.984465141 1 49

hsa00500 Starch and sucrose metabolism 0.507331557 0.592419696 0.99316419 1 32

hsa00600 Sphingolipid metabolism 0.662377407 0.745837522 0.999111146 1 33

hsa04141 Protein processing in endoplasmic reticulum 0.559852183 0.852836674 0.999840586 1 152

hsa00970 Aminoacyl-tRNA biosynthesis 0.493250441 0.881659727 0.999869695 1 40

hsa04142 Lysosome 0.402794454 1.269518545 0.999999949 1 114

hsa03010 Ribosome 1.40E-06 -1.890710517 1 1 88

hsa00061 Fatty acid biosynthesis NA NA NA NA 6

hsa00072 Synthesis and degradation of ketone bodies NA NA NA NA 7

hsa00120 Primary bile acid biosynthesis NA NA NA NA 9

hsa00130 Ubiquinone and other terpenoid-quinone biosynthesis NA NA NA NA 5

hsa00232 Caffeine metabolism NA NA NA NA 1

hsa00300 Lysine biosynthesis NA NA NA NA 3

hsa00400 Phenylalanine, tyrosine and tryptophan biosynthesis NA NA NA NA 3

hsa00430 Taurine and hypotaurine metabolism NA NA NA NA 6

hsa00460 Cyanoamino acid metabolism NA NA NA NA 4

hsa00471 D-Glutamine and D-glutamate metabolism NA NA NA NA 2

hsa00472 D-Arginine and D-ornithine metabolism NA NA NA NA 0

hsa00730 Thiamine metabolism NA NA NA NA 3

hsa00750 Vitamin B6 metabolism NA NA NA NA 5

hsa00780 Biotin metabolism NA NA NA NA 2

hsa00785 Lipoic acid metabolism NA NA NA NA 3

hsa04122 Sulfur relay system NA NA NA NA 9

GSE57065 p.geomean stat.mean p.val q.val set.size

hsa04610 Complement and coagulation cascades 0.006323695 2.557470086 2.86E-34 3.61E-32 22

hsa03320 PPAR signaling pathway 0.009484718 2.423348559 1.51E-31 9.49E-30 23

hsa04512 ECM-receptor interaction 0.033066077 1.752947614 5.78E-18 2.43E-16 32

hsa00190 Oxidative phosphorylation 0.033995879 1.648845137 2.61E-16 8.22E-15 55

hsa04810 Regulation of actin cytoskeleton 0.04412825 1.619092984 4.58E-16 1.16E-14 87

hsa00480 Glutathione metabolism 0.073983063 1.440816773 9.07E-13 1.90E-11 22

hsa00980 Metabolism of xenobiotics by cytochrome P450 0.09832949 1.261096653 4.55E-10 8.19E-09 17

hsa04510 Focal adhesion 0.094310615 1.219373069 6.82E-10 1.07E-08 77

hsa04115 p53 signaling pathway 0.105763393 1.185104354 2.45E-09 3.43E-08 35

hsa00564 Glycerophospholipid metabolism 0.110800687 1.178702969 2.82E-09 3.55E-08 36

hsa04270 Vascular smooth muscle contraction 0.112105143 1.158940797 4.64E-09 5.31E-08 46

hsa04110 Cell cycle 0.084875981 1.156511344 5.05E-09 5.31E-08 73

hsa04114 Oocyte meiosis 0.106659615 1.148173946 6.18E-09 5.99E-08 57

hsa04621 NOD-like receptor signaling pathway 0.117444921 1.131738888 1.34E-08 1.21E-07 24

hsa04666 Fc gamma R-mediated phagocytosis 0.120512095 1.079777444 4.43E-08 3.72E-07 48

hsa00760 Nicotinate and nicotinamide metabolism 0.135047197 1.093036465 6.12E-08 4.82E-07 12

hsa04142 Lysosome 0.122132829 1.062334826 7.07E-08 5.24E-07 51

hsa03050 Proteasome 0.104879832 1.077954357 9.06E-08 6.34E-07 20

hsa04975 Fat digestion and absorption 0.139709561 1.076401023 1.01E-07 6.67E-07 11

hsa00982 Drug metabolism - cytochrome P450 0.132741497 1.047449477 1.63E-07 1.03E-06 18

hsa00512 Mucin type O-Glycan biosynthesis 0.145104296 1.03887001 1.86E-07 1.12E-06 16

hsa00983 Drug metabolism - other enzymes 0.153295604 1.017635843 3.37E-07 1.93E-06 14

hsa00600 Sphingolipid metabolism 0.145321237 0.996496531 5.46E-07 2.99E-06 18

hsa04914 Progesterone-mediated oocyte maturation 0.155146284 0.963241312 8.63E-07 4.53E-06 48

hsa04130 SNARE interactions in vesicular transport 0.157969674 0.927354124 3.10E-06 1.56E-05 19

hsa04620 Toll-like receptor signaling pathway 0.154301705 0.906496796 3.63E-06 1.76E-05 40

hsa00010 Glycolysis / Gluconeogenesis 0.176676377 0.882224723 6.30E-06 2.94E-05 32

hsa04380 Osteoclast differentiation 0.174794935 0.837186175 1.57E-05 7.08E-05 63

hsa04540 Gap junction 0.190927257 0.822060927 2.33E-05 0.000101035 33

hsa04910 Insulin signaling pathway 0.190240471 0.814558066 2.54E-05 0.000106635 61

hsa04740 Olfactory transduction 0.181676931 0.795203478 4.80E-05 0.000194961 21

hsa04920 Adipocytokine signaling pathway 0.208744286 0.77741916 5.86E-05 0.000230558 30

hsa00260 Glycine, serine and threonine metabolism 0.186618432 0.793688228 6.11E-05 0.000233218 13

hsa00565 Ether lipid metabolism 0.210741096 0.772445822 7.57E-05 0.000280648 15

hsa04730 Long-term depression 0.229508887 0.71919779 0.000189212 0.000681163 22

hsa04145 Phagosome 0.201873607 0.700067041 0.000250791 0.00087777 66

hsa04912 GnRH signaling pathway 0.235485716 0.692392594 0.000290031 0.000987672 38

hsa04020 Calcium signaling pathway 0.226863057 0.665419209 0.00046548 0.001543434 60

hsa04622 RIG-I-like receptor signaling pathway 0.226783165 0.66978891 0.000496501 0.00160408 19

hsa04260 Cardiac muscle contraction 0.236662851 0.664122781 0.0005113 0.001610595 26

hsa00052 Galactose metabolism 0.241525952 0.658613474 0.000603231 0.001853831 16

hsa04141 Protein processing in endoplasmic reticulum 0.21008418 0.64753844 0.000641139 0.001923416 73

hsa04976 Bile secretion 0.22367835 0.613242698 0.001270444 0.003722698 24

hsa00520 Amino sugar and nucleotide sugar metabolism 0.238195543 0.606883483 0.001355407 0.003881393 26

hsa04966 Collecting duct acid secretion 0.264173444 0.607748261 0.001447386 0.004052679 12

hsa00051 Fructose and mannose metabolism 0.275016633 0.556020365 0.003126938 0.008565092 15

hsa04370 VEGF signaling pathway 0.276809143 0.535029288 0.003947382 0.010582342 35

hsa04144 Endocytosis 0.269651939 0.495980721 0.006748619 0.017715124 83

hsa04972 Pancreatic secretion 0.297339047 0.483228191 0.008221347 0.021140605 32

hsa00240 Pyrimidine metabolism 0.284675211 0.466165507 0.010305233 0.025969188 41

hsa00380 Tryptophan metabolism 0.298913997 0.450467062 0.013536764 0.033443769 14

hsa04120 Ubiquitin mediated proteolysis 0.266374455 0.430095275 0.016161947 0.039116059 67

hsa00500 Starch and sucrose metabolism 0.242862861 0.438474035 0.016453581 0.039116059 22

hsa00330 Arginine and proline metabolism 0.307542608 0.413545829 0.020406246 0.047614574 25

hsa04623 Cytosolic DNA-sensing pathway 0.306670012 0.411779465 0.021530656 0.049324777 16

hsa04670 Leukocyte transendothelial migration 0.316107805 0.397233084 0.024100342 0.05422577 44

hsa00030 Pentose phosphate pathway 0.333170004 0.372203862 0.034033496 0.075231938 13

hsa00590 Arachidonic acid metabolism 0.323238716 0.351848932 0.040796998 0.08862796 23

hsa03440 Homologous recombination 0.335745663 0.353219158 0.041536054 0.088704116 14

hsa04664 Fc epsilon RI signaling pathway 0.337001288 0.346755923 0.04244902 0.089142941 37

hsa04146 Peroxisome 0.334309761 0.331725449 0.049778762 0.102821704 38

hsa04010 MAPK signaling pathway 0.349075014 0.325789019 0.051986252 0.10564948 115

hsa00534 Glycosaminoglycan biosynthesis - heparan sulfate 0.350372309 0.317377344 0.059747911 0.119495823 13

hsa04630 Jak-STAT signaling pathway 0.337957504 0.307136844 0.063172461 0.124370783 58

hsa00670 One carbon pool by folate 0.355395341 0.287184159 0.079626267 0.154352456 12

hsa04210 Apoptosis 0.449418067 0.056735492 0.388846053 0.739940075 46

hsa04722 Neurotrophin signaling pathway 0.431430029 0.054288695 0.393460199 0.739940075 57

hsa03022 Basal transcription factors 0.451147091 0.033928731 0.432168589 0.800782974 18

hsa03060 Protein export 0.425263488 0.024222386 0.449584292 0.820980012 10

hsa04012 ErbB signaling pathway 0.462441166 0.017319915 0.465410076 0.837738138 35

hsa02010 ABC transporters 0.459138946 0.002992497 0.495453384 0.879255301 14

hsa00071 Fatty acid metabolism 0.482006474 -0.012001475 0.523663412 0.916410972 17

hsa04530 Tight junction 0.489833858 -0.019501103 0.538595435 0.929630477 44

hsa04360 Axon guidance 0.460183028 -0.041949749 0.582766049 0.992277327 47

hsa04974 Protein digestion and absorption 0.443277041 -0.055794664 0.607741176 1 22

hsa00350 Tyrosine metabolism 0.47591606 -0.09133216 0.673147094 1 18

hsa00230 Purine metabolism 0.508506455 -0.122534621 0.729318147 1 68

hsa04962 Vasopressin-regulated water reabsorption 0.538837605 -0.160162354 0.785621423 1 17

hsa04720 Long-term potentiation 0.551346292 -0.180977424 0.815553104 1 27

hsa00514 Other types of O-glycan biosynthesis 0.555359733 -0.213419928 0.852991089 1 14

hsa00561 Glycerolipid metabolism 0.554455185 -0.21241531 0.853152495 1 17

hsa04971 Gastric acid secretion 0.560430398 -0.226702517 0.869182106 1 22

hsa04150 mTOR signaling pathway 0.56730353 -0.234823538 0.877524828 1 21

hsa04742 Taste transduction 0.553295103 -0.256586374 0.893726374 1 10

hsa04970 Salivary secretion 0.557646711 -0.253868279 0.89585237 1 31

hsa00830 Retinol metabolism 0.576146507 -0.281975685 0.917646471 1 16

hsa00640 Propanoate metabolism 0.588731209 -0.284453623 0.920097185 1 18

hsa00340 Histidine metabolism 0.556999019 -0.310346216 0.934645239 1 13

hsa00620 Pyruvate metabolism 0.569929301 -0.310559692 0.937656104 1 23

hsa03420 Nucleotide excision repair 0.564507152 -0.323576625 0.944667413 1 25

hsa04350 TGF-beta signaling pathway 0.603247234 -0.336026743 0.95255432 1 35

hsa04520 Adherens junction 0.606326925 -0.373886164 0.967849527 1 24

hsa04330 Notch signaling pathway 0.59429091 -0.379344691 0.969324297 1 24

hsa04960 Aldosterone-regulated sodium reabsorption 0.632741574 -0.416445033 0.978932102 1 10

hsa00310 Lysine degradation 0.630090024 -0.462072457 0.988689747 1 22

hsa04916 Melanogenesis 0.646485588 -0.460283933 0.988879279 1 34

hsa04973 Carbohydrate digestion and absorption 0.563684556 -0.470555615 0.989102108 1 18

hsa00532 Glycosaminoglycan biosynthesis - chondroitin sulfate 0.658066183 -0.503942878 0.993264607 1 13

hsa00270 Cysteine and methionine metabolism 0.679659665 -0.544775497 0.996385101 1 17

hsa04340 Hedgehog signaling pathway 0.678342919 -0.600543156 0.998269544 1 11

hsa03430 Mismatch repair 0.698681355 -0.630980814 0.998908488 1 11

hsa00562 Inositol phosphate metabolism 0.719329022 -0.639779258 0.999231949 1 26

hsa04070 Phosphatidylinositol signaling system 0.752229232 -0.78148155 0.999946204 1 34

hsa00510 N-Glycan biosynthesis 0.742642409 -0.804923087 0.999964123 1 24

hsa00563 Glycosylphosphatidylinositol(GPI)-anchor biosynthesis 0.754347074 -0.827410274 0.99996613 1 10

hsa04062 Chemokine signaling pathway 0.725827791 -0.831323797 0.999982373 1 76

hsa03030 DNA replication 0.72731904 -0.855610501 0.999984305 1 17

hsa04640 Hematopoietic cell lineage 0.732854786 -0.849686685 0.999987777 1 58

hsa04662 B cell receptor signaling pathway 0.760958496 -0.898786315 0.999995839 1 44

hsa00020 Citrate cycle (TCA cycle) 0.742066421 -0.929314515 0.999996686 1 17

hsa03018 RNA degradation 0.746401894 -0.916010176 0.999997028 1 40

hsa00280 Valine, leucine and isoleucine degradation 0.760691731 -0.957904813 0.999998633 1 21

hsa00410 beta-Alanine metabolism 0.769873659 -0.98251582 0.999998816 1 11

hsa00970 Aminoacyl-tRNA biosynthesis 0.830592233 -1.07549933 0.999999915 1 15

hsa03410 Base excision repair 0.809762962 -1.079736532 0.999999921 1 15

hsa04310 Wnt signaling pathway 0.829752605 -1.075144023 0.999999955 1 62

hsa03010 Ribosome 0.600564096 -1.103220314 0.999999967 1 50

hsa03015 mRNA surveillance pathway 0.854753573 -1.17004162 0.999999997 1 43

hsa03040 Spliceosome 0.882692491 -1.608389136 1 1 65

hsa03008 Ribosome biogenesis in eukaryotes 0.933492962 -2.697995688 1 1 44

hsa03013 RNA transport 0.902985199 -2.195454485 1 1 83

hsa04514 Cell adhesion molecules (CAMs) 0.949170919 -2.153334239 1 1 57

hsa04612 Antigen processing and presentation 0.952883458 -3.234270584 1 1 52

hsa04650 Natural killer cell mediated cytotoxicity 0.782493364 -1.926989937 1 1 62

hsa04660 T cell receptor signaling pathway 0.928005785 -1.862981781 1 1 60

hsa04672 Intestinal immune network for IgA production 0.982274511 -2.737004834 1 1 29

hsa00040 Pentose and glucuronate interconversions NA NA NA NA 8

hsa00053 Ascorbate and aldarate metabolism NA NA NA NA 6

hsa00061 Fatty acid biosynthesis NA NA NA NA 2

hsa00072 Synthesis and degradation of ketone bodies NA NA NA NA 2

hsa00100 Steroid biosynthesis NA NA NA NA 6

hsa00120 Primary bile acid biosynthesis NA NA NA NA 4

hsa00130 Ubiquinone and other terpenoid-quinone biosynthesis NA NA NA NA 5

hsa00140 Steroid hormone biosynthesis NA NA NA NA 8

hsa00232 Caffeine metabolism NA NA NA NA 2

hsa00250 Alanine, aspartate and glutamate metabolism NA NA NA NA 8

hsa00290 Valine, leucine and isoleucine biosynthesis NA NA NA NA 5

hsa00300 Lysine biosynthesis NA NA NA NA 1

hsa00360 Phenylalanine metabolism NA NA NA NA 9

hsa00400 Phenylalanine, tyrosine and tryptophan biosynthesis NA NA NA NA 2

hsa00430 Taurine and hypotaurine metabolism NA NA NA NA 6

hsa00450 Selenocompound metabolism NA NA NA NA 6

hsa00460 Cyanoamino acid metabolism NA NA NA NA 4

hsa00471 D-Glutamine and D-glutamate metabolism NA NA NA NA 2

hsa00472 D-Arginine and D-ornithine metabolism NA NA NA NA 0

hsa00511 Other glycan degradation NA NA NA NA 6

hsa00531 Glycosaminoglycan degradation NA NA NA NA 5

hsa00533 Glycosaminoglycan biosynthesis - keratan sulfate NA NA NA NA 6

hsa00591 Linoleic acid metabolism NA NA NA NA 8

hsa00592 alpha-Linolenic acid metabolism NA NA NA NA 6

hsa00601 Glycosphingolipid biosynthesis - lacto and neolacto series NA NA NA NA 9

hsa00603 Glycosphingolipid biosynthesis - globo series NA NA NA NA 6

hsa00604 Glycosphingolipid biosynthesis - ganglio series NA NA NA NA 8

hsa00630 Glyoxylate and dicarboxylate metabolism NA NA NA NA 7

hsa00650 Butanoate metabolism NA NA NA NA 8

hsa00730 Thiamine metabolism NA NA NA NA 1

hsa00740 Riboflavin metabolism NA NA NA NA 4

hsa00750 Vitamin B6 metabolism NA NA NA NA 4

hsa00770 Pantothenate and CoA biosynthesis NA NA NA NA 9

hsa00780 Biotin metabolism NA NA NA NA 1

hsa00785 Lipoic acid metabolism NA NA NA NA 1

hsa00790 Folate biosynthesis NA NA NA NA 6

hsa00860 Porphyrin and chlorophyll metabolism NA NA NA NA 8

hsa00900 Terpenoid backbone biosynthesis NA NA NA NA 6

hsa00910 Nitrogen metabolism NA NA NA NA 7

hsa00920 Sulfur metabolism NA NA NA NA 5

hsa01040 Biosynthesis of unsaturated fatty acids NA NA NA NA 5

hsa03020 RNA polymerase NA NA NA NA 7

hsa03450 Non-homologous end-joining NA NA NA NA 7

hsa04122 Sulfur relay system NA NA NA NA 4

hsa04140 Regulation of autophagy NA NA NA NA 9

hsa04320 Dorso-ventral axis formation NA NA NA NA 9

hsa04614 Renin-angiotensin system NA NA NA NA 7

hsa04710 Circadian rhythm - mammal NA NA NA NA 8

hsa04744 Phototransduction NA NA NA NA 7

hsa04964 Proximal tubule bicarbonate reclamation NA NA NA NA 8

hsa04977 Vitamin digestion and absorption NA NA NA NA 6

hsa04612 Antigen processing and presentation 0.000523855 -3.234270584 2.90E-55 3.66E-53 52

hsa04672 Intestinal immune network for IgA production 0.003608133 -2.737004834 1.45E-39 9.14E-38 29

hsa03008 Ribosome biogenesis in eukaryotes 0.002848282 -2.697995688 7.18E-39 3.01E-37 44

hsa03013 RNA transport 0.008861591 -2.195454485 1.67E-27 5.26E-26 83

hsa04514 Cell adhesion molecules (CAMs) 0.013269218 -2.153334239 1.64E-26 4.13E-25 57

hsa04650 Natural killer cell mediated cytotoxicity 0.016188012 -1.926989937 1.29E-21 2.71E-20 62

hsa04660 T cell receptor signaling pathway 0.027671058 -1.862981781 1.71E-20 3.08E-19 60

hsa03040 Spliceosome 0.043947735 -1.608389136 1.03E-15 1.62E-14 65

hsa03015 mRNA surveillance pathway 0.115837243 -1.17004162 3.47E-09 4.86E-08 43

hsa03010 Ribosome 0.07335328 -1.103220314 3.33E-08 4.19E-07 50

hsa04310 Wnt signaling pathway 0.1326302 -1.075144023 4.54E-08 5.20E-07 62

hsa03410 Base excision repair 0.131498851 -1.079736532 7.90E-08 8.26E-07 15

hsa00970 Aminoacyl-tRNA biosynthesis 0.139709149 -1.07549933 8.52E-08 8.26E-07 15

hsa00410 beta-Alanine metabolism 0.14997983 -0.98251582 1.18E-06 1.07E-05 11

hsa00280 Valine, leucine and isoleucine degradation 0.149152517 -0.957904813 1.37E-06 1.15E-05 21

hsa03018 RNA degradation 0.153556353 -0.916010176 2.97E-06 2.34E-05 40

hsa00020 Citrate cycle (TCA cycle) 0.152560813 -0.929314515 3.31E-06 2.46E-05 17

hsa04662 B cell receptor signaling pathway 0.164107074 -0.898786315 4.16E-06 2.91E-05 44

hsa04640 Hematopoietic cell lineage 0.170597846 -0.849686685 1.22E-05 8.11E-05 58

hsa03030 DNA replication 0.171875001 -0.855610501 1.57E-05 9.89E-05 17

hsa04062 Chemokine signaling pathway 0.173611772 -0.831323797 1.76E-05 0.000105764 76

hsa00563 Glycosylphosphatidylinositol(GPI)-anchor biosynthesis 0.197394403 -0.827410274 3.39E-05 0.000193981 10

hsa00510 N-Glycan biosynthesis 0.194260047 -0.804923087 3.59E-05 0.000196544 24

hsa04070 Phosphatidylinositol signaling system 0.205451724 -0.78148155 5.38E-05 0.00028243 34

hsa00562 Inositol phosphate metabolism 0.253258602 -0.639779258 0.000768051 0.003870975 26

hsa03430 Mismatch repair 0.252554997 -0.630980814 0.001091512 0.005289635 11

hsa04340 Hedgehog signaling pathway 0.257368442 -0.600543156 0.001730456 0.008075459 11

hsa00270 Cysteine and methionine metabolism 0.280645536 -0.544775497 0.003614899 0.016267044 17

hsa00532 Glycosaminoglycan biosynthesis - chondroitin sulfate 0.291398593 -0.503942878 0.006735393 0.02926412 13

hsa04973 Carbohydrate digestion and absorption 0.254854847 -0.470555615 0.010897892 0.044534123 18

hsa04916 Melanogenesis 0.304475636 -0.460283933 0.011120721 0.044534123 34

hsa00310 Lysine degradation 0.295033806 -0.462072457 0.011310253 0.044534123 22

hsa04960 Aldosterone-regulated sodium reabsorption 0.324334045 -0.416445033 0.021067898 0.080441067 10

hsa04330 Notch signaling pathway 0.317887841 -0.379344691 0.030675703 0.113680548 24

hsa04520 Adherens junction 0.328442497 -0.373886164 0.032150473 0.115741704 24

hsa04350 TGF-beta signaling pathway 0.349250592 -0.336026743 0.04744568 0.166059879 35

hsa03420 Nucleotide excision repair 0.329832375 -0.323576625 0.055332587 0.18842989 25

hsa00620 Pyruvate metabolism 0.342572971 -0.310559692 0.062343896 0.206719234 23

hsa00340 Histidine metabolism 0.333928116 -0.310346216 0.065354761 0.211146149 13

hsa00640 Propanoate metabolism 0.372682622 -0.284453623 0.079902815 0.251693867 18

hsa00830 Retinol metabolism 0.364943768 -0.281975685 0.082353529 0.253086455 16

hsa04970 Salivary secretion 0.367057709 -0.253868279 0.10414763 0.311406438 31

hsa04742 Taste transduction 0.3658902 -0.256586374 0.106273626 0.311406438 10

hsa04150 mTOR signaling pathway 0.389306737 -0.234823538 0.122475172 0.350724356 21

hsa04971 Gastric acid secretion 0.388325221 -0.226702517 0.130817894 0.366290102 22

hsa00561 Glycerolipid metabolism 0.394173247 -0.21241531 0.146847505 0.394108996 17

hsa00514 Other types of O-glycan biosynthesis 0.393926675 -0.213419928 0.147008911 0.394108996 14

hsa04720 Long-term potentiation 0.411840939 -0.180977424 0.184446896 0.484173102 27

hsa04962 Vasopressin-regulated water reabsorption 0.416474537 -0.160162354 0.214378577 0.551259198 17

hsa00230 Purine metabolism 0.416052114 -0.122534621 0.270681853 0.68211827 68

hsa00350 Tyrosine metabolism 0.409470461 -0.09133216 0.326852906 0.807518943 18

hsa04974 Protein digestion and absorption 0.403771361 -0.055794664 0.392258824 0.950473304 22

hsa04360 Axon guidance 0.430196156 -0.041949749 0.417233951 0.991914676 47

hsa04530 Tight junction 0.474244569 -0.019501103 0.461404565 1 44

hsa00071 Fatty acid metabolism 0.472677313 -0.012001475 0.476336588 1 17

hsa02010 ABC transporters 0.46215154 0.002992497 0.504546616 1 14

hsa04012 ErbB signaling pathway 0.474935018 0.017319915 0.534589924 1 35

hsa03060 Protein export 0.443777006 0.024222386 0.550415708 1 10

hsa03022 Basal transcription factors 0.474863816 0.033928731 0.567831411 1 18

hsa04722 Neurotrophin signaling pathway 0.472238096 0.054288695 0.606539801 1 57

hsa04210 Apoptosis 0.493081991 0.056735492 0.611153947 1 46

hsa00670 One carbon pool by folate 0.564890872 0.287184159 0.920373733 1 12

hsa04630 Jak-STAT signaling pathway 0.563463798 0.307136844 0.936827539 1 58

hsa00534 Glycosaminoglycan biosynthesis - heparan sulfate 0.585617582 0.317377344 0.940252089 1 13

hsa04010 MAPK signaling pathway 0.596063 0.325789019 0.948013748 1 115

hsa04146 Peroxisome 0.579722145 0.331725449 0.950221238 1 38

hsa04664 Fc epsilon RI signaling pathway 0.595183688 0.346755923 0.95755098 1 37

hsa03440 Homologous recombination 0.595740168 0.353219158 0.958463946 1 14

hsa00590 Arachidonic acid metabolism 0.573712725 0.351848932 0.959203002 1 23

hsa00030 Pentose phosphate pathway 0.608276192 0.372203862 0.965966504 1 13

hsa04670 Leukocyte transendothelial migration 0.607902507 0.397233084 0.975899658 1 44

hsa04623 Cytosolic DNA-sensing pathway 0.602867187 0.411779465 0.978469344 1 16

hsa00330 Arginine and proline metabolism 0.60728957 0.413545829 0.979593754 1 25

hsa00500 Starch and sucrose metabolism 0.531845906 0.438474035 0.983546419 1 22

hsa04120 Ubiquitin mediated proteolysis 0.550911734 0.430095275 0.983838053 1 67

hsa00380 Tryptophan metabolism 0.62017177 0.450467062 0.986463236 1 14

hsa00240 Pyrimidine metabolism 0.619162397 0.466165507 0.989694767 1 41

hsa04972 Pancreatic secretion 0.655654595 0.483228191 0.991778653 1 32

hsa04144 Endocytosis 0.623815554 0.495980721 0.993251381 1 83

hsa04370 VEGF signaling pathway 0.666834121 0.535029288 0.996052618 1 35

hsa00051 Fructose and mannose metabolism 0.678419297 0.556020365 0.996873062 1 15

hsa04966 Collecting duct acid secretion 0.704166055 0.607748261 0.998552614 1 12

hsa00520 Amino sugar and nucleotide sugar metabolism 0.656216144 0.606883483 0.998644593 1 26

hsa04976 Bile secretion 0.628433114 0.613242698 0.998729556 1 24

hsa04141 Protein processing in endoplasmic reticulum 0.644324968 0.64753844 0.999358861 1 73

hsa00052 Galactose metabolism 0.707882858 0.658613474 0.999396769 1 16

hsa04260 Cardiac muscle contraction 0.71065169 0.664122781 0.9994887 1 26

hsa04622 RIG-I-like receptor signaling pathway 0.692420987 0.66978891 0.999503499 1 19

hsa04020 Calcium signaling pathway 0.694595944 0.665419209 0.99953452 1 60

hsa04912 GnRH signaling pathway 0.734398781 0.692392594 0.999709969 1 38

hsa04145 Phagosome 0.675665804 0.700067041 0.999749209 1 66

hsa04730 Long-term depression 0.743032726 0.71919779 0.999810788 1 22

hsa00565 Ether lipid metabolism 0.747723017 0.772445822 0.999924269 1 15

hsa00260 Glycine, serine and threonine metabolism 0.704776342 0.793688228 0.999938919 1 13

hsa04920 Adipocytokine signaling pathway 0.755692937 0.77741916 0.999941446 1 30

hsa04740 Olfactory transduction 0.703626954 0.795203478 0.999952033 1 21

hsa04910 Insulin signaling pathway 0.750087893 0.814558066 0.999974611 1 61

hsa04540 Gap junction 0.756074517 0.822060927 0.999976746 1 33

hsa04380 Osteoclast differentiation 0.732672682 0.837186175 0.99998426 1 63

hsa00010 Glycolysis / Gluconeogenesis 0.776599182 0.882224723 0.999993698 1 32

hsa04620 Toll-like receptor signaling pathway 0.741048659 0.906496796 0.999996367 1 40

hsa04130 SNARE interactions in vesicular transport 0.756803545 0.927354124 0.999996903 1 19

hsa04914 Progesterone-mediated oocyte maturation 0.794016914 0.963241312 0.999999137 1 48

hsa00600 Sphingolipid metabolism 0.784293464 0.996496531 0.999999454 1 18

hsa00983 Drug metabolism - other enzymes 0.822685832 1.017635843 0.999999663 1 14

hsa00512 Mucin type O-Glycan biosynthesis 0.820575761 1.03887001 0.999999814 1 16

hsa00982 Drug metabolism - cytochrome P450 0.789039557 1.047449477 0.999999837 1 18

hsa04975 Fat digestion and absorption 0.826605566 1.076401023 0.999999899 1 11

hsa03050 Proteasome 0.716775376 1.077954357 0.999999909 1 20

hsa04142 Lysosome 0.78760498 1.062334826 0.999999929 1 51

hsa00760 Nicotinate and nicotinamide metabolism 0.827350269 1.093036465 0.999999939 1 12

hsa04666 Fc gamma R-mediated phagocytosis 0.794899718 1.079777444 0.999999956 1 48

hsa04621 NOD-like receptor signaling pathway 0.821443303 1.131738888 0.999999987 1 24

hsa04114 Oocyte meiosis 0.805424104 1.148173946 0.999999994 1 57

hsa04110 Cell cycle 0.737795564 1.156511344 0.999999995 1 73

hsa04270 Vascular smooth muscle contraction 0.834560817 1.158940797 0.999999995 1 46

hsa00564 Glycerophospholipid metabolism 0.845579225 1.178702969 0.999999997 1 36

hsa04115 p53 signaling pathway 0.82909321 1.185104354 0.999999998 1 35

hsa04510 Focal adhesion 0.821324739 1.219373069 0.999999999 1 77

hsa00980 Metabolism of xenobiotics by cytochrome P450 0.849546822 1.261096653 1 1 17

hsa00480 Glutathione metabolism 0.901997925 1.440816773 1 1 22

hsa04810 Regulation of actin cytoskeleton 0.888693247 1.619092984 1 1 87

hsa00190 Oxidative phosphorylation 0.839757082 1.648845137 1 1 55

hsa03320 PPAR signaling pathway 0.985572277 2.423348559 1 1 23

hsa04512 ECM-receptor interaction 0.889855434 1.752947614 1 1 32

hsa04610 Complement and coagulation cascades 0.975002192 2.557470086 1 1 22

hsa00040 Pentose and glucuronate interconversions NA NA NA NA 8

hsa00053 Ascorbate and aldarate metabolism NA NA NA NA 6

hsa00061 Fatty acid biosynthesis NA NA NA NA 2

hsa00072 Synthesis and degradation of ketone bodies NA NA NA NA 2

hsa00100 Steroid biosynthesis NA NA NA NA 6

hsa00120 Primary bile acid biosynthesis NA NA NA NA 4

hsa00130 Ubiquinone and other terpenoid-quinone biosynthesis NA NA NA NA 5

hsa00140 Steroid hormone biosynthesis NA NA NA NA 8

hsa00232 Caffeine metabolism NA NA NA NA 2

hsa00250 Alanine, aspartate and glutamate metabolism NA NA NA NA 8

hsa00290 Valine, leucine and isoleucine biosynthesis NA NA NA NA 5

hsa00300 Lysine biosynthesis NA NA NA NA 1

hsa00360 Phenylalanine metabolism NA NA NA NA 9

hsa00400 Phenylalanine, tyrosine and tryptophan biosynthesis NA NA NA NA 2

hsa00430 Taurine and hypotaurine metabolism NA NA NA NA 6

hsa00450 Selenocompound metabolism NA NA NA NA 6

hsa00460 Cyanoamino acid metabolism NA NA NA NA 4

hsa00471 D-Glutamine and D-glutamate metabolism NA NA NA NA 2

hsa00472 D-Arginine and D-ornithine metabolism NA NA NA NA 0

hsa00511 Other glycan degradation NA NA NA NA 6

hsa00531 Glycosaminoglycan degradation NA NA NA NA 5

hsa00533 Glycosaminoglycan biosynthesis - keratan sulfate NA NA NA NA 6

hsa00591 Linoleic acid metabolism NA NA NA NA 8

hsa00592 alpha-Linolenic acid metabolism NA NA NA NA 6

hsa00601 Glycosphingolipid biosynthesis - lacto and neolacto series NA NA NA NA 9

hsa00603 Glycosphingolipid biosynthesis - globo series NA NA NA NA 6

hsa00604 Glycosphingolipid biosynthesis - ganglio series NA NA NA NA 8

hsa00630 Glyoxylate and dicarboxylate metabolism NA NA NA NA 7

hsa00650 Butanoate metabolism NA NA NA NA 8

hsa00730 Thiamine metabolism NA NA NA NA 1

hsa00740 Riboflavin metabolism NA NA NA NA 4

hsa00750 Vitamin B6 metabolism NA NA NA NA 4

hsa00770 Pantothenate and CoA biosynthesis NA NA NA NA 9

hsa00780 Biotin metabolism NA NA NA NA 1

hsa00785 Lipoic acid metabolism NA NA NA NA 1

hsa00790 Folate biosynthesis NA NA NA NA 6

hsa00860 Porphyrin and chlorophyll metabolism NA NA NA NA 8

hsa00900 Terpenoid backbone biosynthesis NA NA NA NA 6

hsa00910 Nitrogen metabolism NA NA NA NA 7

hsa00920 Sulfur metabolism NA NA NA NA 5

hsa01040 Biosynthesis of unsaturated fatty acids NA NA NA NA 5

hsa03020 RNA polymerase NA NA NA NA 7

hsa03450 Non-homologous end-joining NA NA NA NA 7

hsa04122 Sulfur relay system NA NA NA NA 4

hsa04140 Regulation of autophagy NA NA NA NA 9

hsa04320 Dorso-ventral axis formation NA NA NA NA 9

hsa04614 Renin-angiotensin system NA NA NA NA 7

hsa04710 Circadian rhythm - mammal NA NA NA NA 8

hsa04744 Phototransduction NA NA NA NA 7

hsa04964 Proximal tubule bicarbonate reclamation NA NA NA NA 8

hsa04977 Vitamin digestion and absorption NA NA NA NA 6

GSE95233 p.geomean stat.mean p.val q.val set.size

hsa00190 Oxidative phosphorylation 0.002405527 2.799196786 5.20E-38 6.45E-36 63

hsa03320 PPAR signaling pathway 0.023874351 2.041525896 3.13E-20 1.94E-18 17

hsa04610 Complement and coagulation cascades 0.027170072 1.925687416 1.70E-18 7.01E-17 20

hsa00600 Sphingolipid metabolism 0.043531155 1.739475885 1.97E-15 6.09E-14 16

hsa04115 p53 signaling pathway 0.059317356 1.532608268 8.32E-13 2.06E-11 30

hsa04260 Cardiac muscle contraction 0.064059077 1.478909164 4.89E-12 1.01E-10 30

hsa00480 Glutathione metabolism 0.074180722 1.419136235 4.36E-11 7.72E-10 20

hsa04621 NOD-like receptor signaling pathway 0.082022662 1.347229837 2.47E-10 3.71E-09 31

hsa04114 Oocyte meiosis 0.0802321 1.336928384 2.69E-10 3.71E-09 49

hsa00770 Pantothenate and CoA biosynthesis 0.09051072 1.367442579 4.36E-10 5.40E-09 10

hsa04110 Cell cycle 0.077352396 1.293514089 9.25E-10 1.04E-08 64

hsa04130 SNARE interactions in vesicular transport 0.120251197 1.166149092 4.17E-08 4.31E-07 23

hsa04966 Collecting duct acid secretion 0.121037232 1.181083 5.34E-08 5.09E-07 10

hsa00564 Glycerophospholipid metabolism 0.11804511 1.137647858 6.87E-08 6.08E-07 31

hsa00512 Mucin type O-Glycan biosynthesis 0.12296791 1.128623959 1.37E-07 1.13E-06 15

hsa04142 Lysosome 0.147791752 0.96042121 3.92E-06 3.04E-05 52

hsa03010 Ribosome 0.101761727 0.952292575 5.58E-06 4.07E-05 57

hsa03050 Proteasome 0.155425607 0.916588863 1.18E-05 8.13E-05 28

hsa04810 Regulation of actin cytoskeleton 0.153795464 0.902471843 1.28E-05 8.34E-05 74

hsa00760 Nicotinate and nicotinamide metabolism 0.172569009 0.918897089 1.65E-05 0.000102243 11

hsa00500 Starch and sucrose metabolism 0.176279294 0.896219269 2.13E-05 0.00012579 13

hsa04622 RIG-I-like receptor signaling pathway 0.18326514 0.832021411 5.99E-05 0.000337554 25

hsa04914 Progesterone-mediated oocyte maturation 0.187167838 0.820907802 6.61E-05 0.000355288 44

hsa04540 Gap junction 0.190249119 0.822086341 6.88E-05 0.000355288 27

hsa04512 ECM-receptor interaction 0.198250317 0.776777836 0.000184832 0.000916769 16

hsa04620 Toll-like receptor signaling pathway 0.16949466 0.757073757 0.000220714 0.001052638 43

hsa00983 Drug metabolism - other enzymes 0.225893716 0.733123885 0.000357773 0.001643104 16

hsa00010 Glycolysis / Gluconeogenesis 0.223996747 0.71951314 0.000418399 0.001852911 28

hsa00565 Ether lipid metabolism 0.229568047 0.699884077 0.000680638 0.002910313 12

hsa04141 Protein processing in endoplasmic reticulum 0.226173816 0.681900788 0.000719969 0.002975871 77

hsa04666 Fc gamma R-mediated phagocytosis 0.229237667 0.65880283 0.001072398 0.004289592 45

hsa03060 Protein export 0.249549271 0.640082208 0.00175819 0.006812988 12

hsa04510 Focal adhesion 0.260932042 0.562041878 0.004324374 0.015891748 65

hsa04910 Insulin signaling pathway 0.24335982 0.561929824 0.004416662 0.015891748 53

hsa04730 Long-term depression 0.271049583 0.564432997 0.004485574 0.015891748 19

hsa00240 Pyrimidine metabolism 0.267918513 0.549812329 0.005175318 0.017826094 46

hsa04912 GnRH signaling pathway 0.280525461 0.546837186 0.005397301 0.018088251 35

hsa00052 Galactose metabolism 0.290773336 0.536632657 0.006730104 0.021509737 12

hsa04120 Ubiquitin mediated proteolysis 0.268014629 0.528831177 0.006765159 0.021509737 62

hsa03022 Basal transcription factors 0.30664421 0.476814175 0.01358191 0.042103921 19

hsa00520 Amino sugar and nucleotide sugar metabolism 0.304550057 0.469809265 0.014674645 0.044381854 20

hsa04270 Vascular smooth muscle contraction 0.292864477 0.459014684 0.016213707 0.04786904 40

hsa04140 Regulation of autophagy 0.304982413 0.463150401 0.01724683 0.049735045 10

hsa04920 Adipocytokine signaling pathway 0.31954882 0.419574248 0.025854521 0.07286274 19

hsa00982 Drug metabolism - cytochrome P450 0.32054652 0.375061263 0.042630916 0.117471858 15

hsa04145 Phagosome 0.305152604 0.355111418 0.048713175 0.131313776 63

hsa04976 Bile secretion 0.361439108 0.292586915 0.087524971 0.230916944 19

hsa00410 beta-Alanine metabolism 0.371970908 0.285064325 0.094677734 0.244584147 10

hsa00670 One carbon pool by folate 0.364183609 0.262694129 0.114236267 0.289087696 10

hsa04623 Cytosolic DNA-sensing pathway 0.366978383 0.254134941 0.119304662 0.293481185 23

hsa00590 Arachidonic acid metabolism 0.378843844 0.25495875 0.120705971 0.293481185 11

hsa00051 Fructose and mannose metabolism 0.408307276 0.181185009 0.200480688 0.46459696 17

hsa00140 Steroid hormone biosynthesis 0.392742352 0.182230976 0.201691477 0.46459696 11

hsa04144 Endocytosis 0.385298326 0.178315479 0.202324483 0.46459696 80

hsa00330 Arginine and proline metabolism 0.400007749 0.174296707 0.209125745 0.471483497 23

hsa04380 Osteoclast differentiation 0.370299683 0.164313504 0.221069318 0.489510633 68

hsa00030 Pentose phosphate pathway 0.432383801 0.144500958 0.252945098 0.550266529 10

hsa04146 Peroxisome 0.429049467 0.130951373 0.270787726 0.578048747 31

hsa04010 MAPK signaling pathway 0.399691392 0.12767921 0.275039323 0.578048747 100

hsa03420 Nucleotide excision repair 0.421770752 0.111723024 0.301650743 0.623411536 26

hsa03430 Mismatch repair 0.425176367 0.105504754 0.314511617 0.639335091 11

hsa04974 Protein digestion and absorption 0.409096433 0.086254184 0.345501396 0.680383033 17

hsa04664 Fc epsilon RI signaling pathway 0.430407946 0.085270771 0.345678477 0.680383033 30

hsa04972 Pancreatic secretion 0.451248965 0.060689464 0.389054944 0.753793954 22

hsa04962 Vasopressin-regulated water reabsorption 0.471824615 0.03268503 0.439642548 0.838702707 19

hsa04740 Olfactory transduction 0.461171799 0.00897602 0.483279626 0.907979902 13

hsa04960 Aldosterone-regulated sodium reabsorption 0.476469895 0.001433239 0.496890405 0.919618063 14

hsa03015 mRNA surveillance pathway 0.483397082 -0.01291453 0.523870469 0.949097087 28

hsa00980 Metabolism of xenobiotics by cytochrome P450 0.46091798 -0.017370575 0.532495006 0.949097087 15

hsa04150 mTOR signaling pathway 0.47292885 -0.019289802 0.535780613 0.949097087 23

hsa04340 Hedgehog signaling pathway 0.462662989 -0.031613012 0.557810197 0.970953798 14

hsa00620 Pyruvate metabolism 0.500093423 -0.037140756 0.568413811 0.970953798 16

hsa00640 Propanoate metabolism 0.493372052 -0.039010701 0.571609897 0.970953798 14

hsa04330 Notch signaling pathway 0.486978011 -0.045831183 0.583663284 0.971335763 16

hsa03018 RNA degradation 0.491614107 -0.047445635 0.587501469 0.971335763 31

hsa00020 Citrate cycle (TCA cycle) 0.505988796 -0.05475885 0.600121557 0.977247166 17

hsa00071 Fatty acid metabolism 0.504199231 -0.058851455 0.606838966 0.977247166 16

hsa04020 Calcium signaling pathway 0.516302166 -0.153175556 0.762847733 1 54

hsa00380 Tryptophan metabolism 0.518422045 -0.175853273 0.788310377 1 12

hsa04360 Axon guidance 0.509780843 -0.18935995 0.811219373 1 41

hsa04210 Apoptosis 0.536704845 -0.190111157 0.812507383 1 49

hsa04973 Carbohydrate digestion and absorption 0.546063843 -0.196097416 0.817571972 1 14

hsa04971 Gastric acid secretion 0.54938446 -0.211472955 0.836938788 1 24

hsa04370 VEGF signaling pathway 0.550486729 -0.212760948 0.839042436 1 31

hsa04720 Long-term potentiation 0.557502193 -0.255018734 0.881460693 1 29

hsa03030 DNA replication 0.539718677 -0.280035079 0.902772456 1 19

hsa03040 Spliceosome 0.565471284 -0.314034859 0.928633923 1 62

hsa00561 Glycerolipid metabolism 0.604228446 -0.34748943 0.945763894 1 16

hsa04670 Leukocyte transendothelial migration 0.594623185 -0.3498243 0.948501202 1 41

hsa00230 Purine metabolism 0.60064389 -0.398751409 0.968850921 1 67

hsa00280 Valine, leucine and isoleucine degradation 0.61161551 -0.405330006 0.969706615 1 22

hsa03020 RNA polymerase 0.627935963 -0.420896642 0.973794645 1 15

hsa00563 Glycosylphosphatidylinositol(GPI)-anchor biosynthesis 0.629197644 -0.425756424 0.973819205 1 10

hsa04916 Melanogenesis 0.62085355 -0.424410724 0.97578663 1 31

hsa00270 Cysteine and methionine metabolism 0.64261013 -0.435500686 0.978174685 1 18

hsa04970 Salivary secretion 0.655957267 -0.481615545 0.987497686 1 29

hsa00514 Other types of O-glycan biosynthesis 0.666225585 -0.517012131 0.991443934 1 14

hsa04662 B cell receptor signaling pathway 0.640081314 -0.517485781 0.991968065 1 41

hsa03410 Base excision repair 0.653636769 -0.526089022 0.992113123 1 14

hsa00310 Lysine degradation 0.66470473 -0.528728928 0.992689874 1 17

hsa04350 TGF-beta signaling pathway 0.695985624 -0.566944806 0.99582273 1 26

hsa00510 N-Glycan biosynthesis 0.668029074 -0.573956449 0.996087483 1 22

hsa00830 Retinol metabolism 0.70030628 -0.612940663 0.997594785 1 13

hsa04710 Circadian rhythm - mammal 0.707529521 -0.62694447 0.997934098 1 10

hsa04520 Adherens junction 0.70653717 -0.627616024 0.998206007 1 22

hsa04722 Neurotrophin signaling pathway 0.677203681 -0.643469671 0.998638926 1 50

hsa00532 Glycosaminoglycan biosynthesis - chondroitin sulfate 0.724090055 -0.693820692 0.999216523 1 10

hsa04012 ErbB signaling pathway 0.713986543 -0.715104655 0.999546754 1 30

hsa00562 Inositol phosphate metabolism 0.757191001 -0.786642798 0.999872327 1 31

hsa04742 Taste transduction 0.802124012 -0.968677766 0.999994491 1 11

hsa04630 Jak-STAT signaling pathway 0.788789658 -0.94752164 0.999995032 1 60

hsa04070 Phosphatidylinositol signaling system 0.811839185 -0.962674659 0.999996316 1 42

hsa04530 Tight junction 0.802053756 -1.008323496 0.99999856 1 34

hsa04310 Wnt signaling pathway 0.816982134 -1.034466898 0.999999277 1 49

hsa00970 Aminoacyl-tRNA biosynthesis 0.825350021 -1.099066804 0.999999584 1 10

hsa04062 Chemokine signaling pathway 0.808304824 -1.110360608 0.999999888 1 76

hsa04640 Hematopoietic cell lineage 0.889669695 -1.499443554 1 1 46

hsa03013 RNA transport 0.899836378 -1.557272524 1 1 61

hsa03008 Ribosome biogenesis in eukaryotes 0.889365877 -1.642285908 1 1 37

hsa04514 Cell adhesion molecules (CAMs) 0.99642481 -3.492975858 1 1 56

hsa04612 Antigen processing and presentation 0.996742098 -4.19632946 1 1 48

hsa04650 Natural killer cell mediated cytotoxicity 0.956887957 -2.589426258 1 1 69

hsa04660 T cell receptor signaling pathway 0.984081578 -2.447563982 1 1 57

hsa04672 Intestinal immune network for IgA production 0.986454584 -2.868405959 1 1 25

hsa00040 Pentose and glucuronate interconversions NA NA NA NA 5

hsa00053 Ascorbate and aldarate metabolism NA NA NA NA 4

hsa00061 Fatty acid biosynthesis NA NA NA NA 1

hsa00072 Synthesis and degradation of ketone bodies NA NA NA NA 3

hsa00100 Steroid biosynthesis NA NA NA NA 8

hsa00120 Primary bile acid biosynthesis NA NA NA NA 2

hsa00130 Ubiquinone and other terpenoid-quinone biosynthesis NA NA NA NA 5

hsa00232 Caffeine metabolism NA NA NA NA 3

hsa00250 Alanine, aspartate and glutamate metabolism NA NA NA NA 8

hsa00260 Glycine, serine and threonine metabolism NA NA NA NA 9

hsa00290 Valine, leucine and isoleucine biosynthesis NA NA NA NA 3

hsa00300 Lysine biosynthesis NA NA NA NA 1

hsa00340 Histidine metabolism NA NA NA NA 7

hsa00350 Tyrosine metabolism NA NA NA NA 8

hsa00360 Phenylalanine metabolism NA NA NA NA 5

hsa00400 Phenylalanine, tyrosine and tryptophan biosynthesis NA NA NA NA 2

hsa00430 Taurine and hypotaurine metabolism NA NA NA NA 1

hsa00450 Selenocompound metabolism NA NA NA NA 4

hsa00460 Cyanoamino acid metabolism NA NA NA NA 3

hsa00471 D-Glutamine and D-glutamate metabolism NA NA NA NA 2

hsa00472 D-Arginine and D-ornithine metabolism NA NA NA NA 0

hsa00511 Other glycan degradation NA NA NA NA 7

hsa00531 Glycosaminoglycan degradation NA NA NA NA 7

hsa00533 Glycosaminoglycan biosynthesis - keratan sulfate NA NA NA NA 6

hsa00534 Glycosaminoglycan biosynthesis - heparan sulfate NA NA NA NA 8

hsa00591 Linoleic acid metabolism NA NA NA NA 6

hsa00592 alpha-Linolenic acid metabolism NA NA NA NA 4

hsa00601 Glycosphingolipid biosynthesis - lacto and neolacto series NA NA NA NA 9

hsa00603 Glycosphingolipid biosynthesis - globo series NA NA NA NA 6

hsa00604 Glycosphingolipid biosynthesis - ganglio series NA NA NA NA 8

hsa00630 Glyoxylate and dicarboxylate metabolism NA NA NA NA 9

hsa00650 Butanoate metabolism NA NA NA NA 5

hsa00730 Thiamine metabolism NA NA NA NA 1

hsa00740 Riboflavin metabolism NA NA NA NA 3

hsa00750 Vitamin B6 metabolism NA NA NA NA 2

hsa00780 Biotin metabolism NA NA NA NA 0

hsa00785 Lipoic acid metabolism NA NA NA NA 1

hsa00790 Folate biosynthesis NA NA NA NA 4

hsa00860 Porphyrin and chlorophyll metabolism NA NA NA NA 6

hsa00900 Terpenoid backbone biosynthesis NA NA NA NA 7

hsa00910 Nitrogen metabolism NA NA NA NA 7

hsa00920 Sulfur metabolism NA NA NA NA 4

hsa01040 Biosynthesis of unsaturated fatty acids NA NA NA NA 7

hsa02010 ABC transporters NA NA NA NA 9

hsa03440 Homologous recombination NA NA NA NA 8

hsa03450 Non-homologous end-joining NA NA NA NA 7

hsa04122 Sulfur relay system NA NA NA NA 2

hsa04320 Dorso-ventral axis formation NA NA NA NA 7

hsa04614 Renin-angiotensin system NA NA NA NA 5

hsa04744 Phototransduction NA NA NA NA 8

hsa04964 Proximal tubule bicarbonate reclamation NA NA NA NA 7

hsa04975 Fat digestion and absorption NA NA NA NA 9

hsa04977 Vitamin digestion and absorption NA NA NA NA 7

hsa04612 Antigen processing and presentation 2.27E-05 -4.19632946 7.01E-78 8.69E-76 48

hsa04514 Cell adhesion molecules (CAMs) 0.000275167 -3.492975858 7.12E-57 4.41E-55 56

hsa04672 Intestinal immune network for IgA production 0.002645237 -2.868405959 1.68E-37 6.93E-36 25

hsa04650 Natural killer cell mediated cytotoxicity 0.003747811 -2.589426258 4.69E-33 1.45E-31 69

hsa04660 T cell receptor signaling pathway 0.007193874 -2.447563982 8.09E-30 2.01E-28 57

hsa03008 Ribosome biogenesis in eukaryotes 0.043118563 -1.642285908 2.12E-14 4.38E-13 37

hsa03013 RNA transport 0.053971972 -1.557272524 2.42E-13 4.30E-12 61

hsa04640 Hematopoietic cell lineage 0.058596705 -1.499443554 1.99E-12 3.08E-11 46

hsa04062 Chemokine signaling pathway 0.114471919 -1.110360608 1.12E-07 1.55E-06 76

hsa00970 Aminoacyl-tRNA biosynthesis 0.136777386 -1.099066804 4.16E-07 5.16E-06 10

hsa04310 Wnt signaling pathway 0.140968001 -1.034466898 7.23E-07 8.15E-06 49

hsa04530 Tight junction 0.144967977 -1.008323496 1.44E-06 1.49E-05 34

hsa04070 Phosphatidylinositol signaling system 0.161567494 -0.962674659 3.68E-06 3.51E-05 42

hsa04630 Jak-STAT signaling pathway 0.156512192 -0.94752164 4.97E-06 4.40E-05 60

hsa04742 Taste transduction 0.164815459 -0.968677766 5.51E-06 4.55E-05 11

hsa00562 Inositol phosphate metabolism 0.20563815 -0.786642798 0.000127673 0.000989466 31

hsa04012 ErbB signaling pathway 0.215333087 -0.715104655 0.000453246 0.003306029 30

hsa00532 Glycosaminoglycan biosynthesis - chondroitin sulfate 0.236119334 -0.693820692 0.000783477 0.005397286 10

hsa04722 Neurotrophin signaling pathway 0.227714622 -0.643469671 0.001361074 0.008882797 50

hsa04520 Adherens junction 0.253445218 -0.627616024 0.001793993 0.011122755 22

hsa04710 Circadian rhythm - mammal 0.258587199 -0.62694447 0.002065902 0.012198657 10

hsa00830 Retinol metabolism 0.259222479 -0.612940663 0.002405215 0.013556667 13

hsa00510 N-Glycan biosynthesis 0.260816553 -0.573956449 0.003912517 0.021093567 22

hsa04350 TGF-beta signaling pathway 0.276808272 -0.566944806 0.00417727 0.021582563 26

hsa00310 Lysine degradation 0.280948854 -0.528728928 0.007310126 0.036258226 17

hsa03410 Base excision repair 0.277480592 -0.526089022 0.007886877 0.036887404 14

hsa04662 B cell receptor signaling pathway 0.267297743 -0.517485781 0.008031935 0.036887404 41

hsa00514 Other types of O-glycan biosynthesis 0.288745698 -0.517012131 0.008556066 0.037891148 14

hsa04970 Salivary secretion 0.298667629 -0.481615545 0.012502314 0.053458172 29

hsa00270 Cysteine and methionine metabolism 0.31779388 -0.435500686 0.021825315 0.090211301 18

hsa04916 Melanogenesis 0.307809028 -0.424410724 0.02421337 0.096853478 31

hsa00563 Glycosylphosphatidylinositol(GPI)-anchor biosynthesis 0.318205481 -0.425756424 0.026180795 0.098468607 10

hsa03020 RNA polymerase 0.318128709 -0.420896642 0.026205355 0.098468607 15

hsa00280 Valine, leucine and isoleucine degradation 0.313740567 -0.405330006 0.030293385 0.110356737 22

hsa00230 Purine metabolism 0.312231752 -0.398751409 0.031149079 0.110356737 67

hsa04670 Leukocyte transendothelial migration 0.333252451 -0.3498243 0.051498798 0.17738475 41

hsa00561 Glycerolipid metabolism 0.344661037 -0.34748943 0.054236106 0.181764247 16

hsa03040 Spliceosome 0.33695989 -0.314034859 0.071366077 0.232878777 62

hsa03030 DNA replication 0.342349169 -0.280035079 0.097227544 0.30913373 19

hsa04720 Long-term potentiation 0.364436535 -0.255018734 0.118539307 0.367471851 29

hsa04370 VEGF signaling pathway 0.388621546 -0.212760948 0.160957564 0.481418815 31

hsa04971 Gastric acid secretion 0.388819729 -0.211472955 0.163061212 0.481418815 24

hsa04973 Carbohydrate digestion and absorption 0.398617215 -0.196097416 0.182428028 0.520195506 14

hsa04210 Apoptosis 0.392313712 -0.190111157 0.187492617 0.520195506 49

hsa04360 Axon guidance 0.372183603 -0.18935995 0.188780627 0.520195506 41

hsa00380 Tryptophan metabolism 0.388325157 -0.175853273 0.211689623 0.570641592 12

hsa04020 Calcium signaling pathway 0.401436606 -0.153175556 0.237152267 0.625678321 54

hsa00071 Fatty acid metabolism 0.458066187 -0.058851455 0.393161034 1 16

hsa00020 Citrate cycle (TCA cycle) 0.463937459 -0.05475885 0.399878443 1 17

hsa03018 RNA degradation 0.455178294 -0.047445635 0.412498531 1 31

hsa04330 Notch signaling pathway 0.452373772 -0.045831183 0.416336716 1 16

hsa00640 Propanoate metabolism 0.46349701 -0.039010701 0.428390103 1 14

hsa00620 Pyruvate metabolism 0.471219028 -0.037140756 0.431586189 1 16

hsa04340 Hedgehog signaling pathway 0.440340955 -0.031613012 0.442189803 1 14

hsa04150 mTOR signaling pathway 0.458495402 -0.019289802 0.464219387 1 23

hsa00980 Metabolism of xenobiotics by cytochrome P450 0.449064886 -0.017370575 0.467504994 1 15

hsa03015 mRNA surveillance pathway 0.473359166 -0.01291453 0.476129531 1 28

hsa04960 Aldosterone-regulated sodium reabsorption 0.477092799 0.001433239 0.503109595 1 14

hsa04740 Olfactory transduction 0.468390194 0.00897602 0.516720374 1 13

hsa04962 Vasopressin-regulated water reabsorption 0.497051442 0.03268503 0.560357452 1 19

hsa04972 Pancreatic secretion 0.498085967 0.060689464 0.610945056 1 22

hsa04664 Fc epsilon RI signaling pathway 0.494742064 0.085270771 0.654321523 1 30

hsa04974 Protein digestion and absorption 0.47183504 0.086254184 0.654498604 1 17

hsa03430 Mismatch repair 0.504461898 0.105504754 0.685488383 1 11

hsa03420 Nucleotide excision repair 0.505818768 0.111723024 0.698349257 1 26

hsa04010 MAPK signaling pathway 0.492553192 0.12767921 0.724960677 1 100

hsa04146 Peroxisome 0.530199964 0.130951373 0.729212274 1 31

hsa00030 Pentose phosphate pathway 0.543688226 0.144500958 0.747054902 1 10

hsa04380 Osteoclast differentiation 0.483918119 0.164313504 0.778930682 1 68

hsa00330 Arginine and proline metabolism 0.530166547 0.174296707 0.790874255 1 23

hsa04144 Endocytosis 0.519600384 0.178315479 0.797675517 1 80

hsa00140 Steroid hormone biosynthesis 0.526475503 0.182230976 0.798308523 1 11

hsa00051 Fructose and mannose metabolism 0.546679988 0.181185009 0.799519312 1 17

hsa00590 Arachidonic acid metabolism 0.570427709 0.25495875 0.879294029 1 11

hsa04623 Cytosolic DNA-sensing pathway 0.557675072 0.254134941 0.880695338 1 23

hsa00670 One carbon pool by folate 0.554795924 0.262694129 0.885763733 1 10

hsa00410 beta-Alanine metabolism 0.585168381 0.285064325 0.905322266 1 10

hsa04976 Bile secretion 0.580590188 0.292586915 0.912475029 1 19

hsa04145 Phagosome 0.555574469 0.355111418 0.951286825 1 63

hsa00982 Drug metabolism - cytochrome P450 0.592457984 0.375061263 0.957369084 1 15

hsa04920 Adipocytokine signaling pathway 0.630891889 0.419574248 0.974145479 1 19

hsa04140 Regulation of autophagy 0.642203171 0.463150401 0.98275317 1 10

hsa04270 Vascular smooth muscle contraction 0.626300305 0.459014684 0.983786293 1 40

hsa00520 Amino sugar and nucleotide sugar metabolism 0.653491632 0.469809265 0.985325355 1 20

hsa03022 Basal transcription factors 0.661041852 0.476814175 0.98641809 1 19

hsa04120 Ubiquitin mediated proteolysis 0.646291348 0.528831177 0.993234841 1 62

hsa00052 Galactose metabolism 0.687462737 0.536632657 0.993269896 1 12

hsa04912 GnRH signaling pathway 0.684718997 0.546837186 0.994602699 1 35

hsa00240 Pyrimidine metabolism 0.668038916 0.549812329 0.994824682 1 46

hsa04730 Long-term depression 0.680593855 0.564432997 0.995514426 1 19

hsa04910 Insulin signaling pathway 0.633543422 0.561929824 0.995583338 1 53

hsa04510 Focal adhesion 0.664757309 0.562041878 0.995675626 1 65

hsa03060 Protein export 0.701516917 0.640082208 0.99824181 1 12

hsa04666 Fc gamma R-mediated phagocytosis 0.691801397 0.65880283 0.998927602 1 45

hsa04141 Protein processing in endoplasmic reticulum 0.707083743 0.681900788 0.999280031 1 77

hsa00565 Ether lipid metabolism 0.718036268 0.699884077 0.999319362 1 12

hsa00010 Glycolysis / Gluconeogenesis 0.735564369 0.71951314 0.999581601 1 28

hsa00983 Drug metabolism - other enzymes 0.747265109 0.733123885 0.999642227 1 16

hsa04620 Toll-like receptor signaling pathway 0.638668924 0.757073757 0.999779286 1 43

hsa04512 ECM-receptor interaction 0.719223739 0.776777836 0.999815168 1 16

hsa04540 Gap junction 0.75049247 0.822086341 0.999931235 1 27

hsa04914 Progesterone-mediated oocyte maturation 0.746126178 0.820907802 0.999933856 1 44

hsa04622 RIG-I-like receptor signaling pathway 0.742904404 0.832021411 0.999940111 1 25

hsa00500 Starch and sucrose metabolism 0.775115118 0.896219269 0.999978697 1 13

hsa00760 Nicotinate and nicotinamide metabolism 0.778282497 0.918897089 0.999983509 1 11

hsa04810 Regulation of actin cytoskeleton 0.739967183 0.902471843 0.999987218 1 74

hsa03050 Proteasome 0.750495362 0.916588863 0.9999882 1 28

hsa03010 Ribosome 0.646488497 0.952292575 0.999994417 1 57

hsa04142 Lysosome 0.776112597 0.96042121 0.999996077 1 52

hsa00512 Mucin type O-Glycan biosynthesis 0.827482899 1.128623959 0.999999863 1 15

hsa00564 Glycerophospholipid metabolism 0.829897708 1.137647858 0.999999931 1 31

hsa04966 Collecting duct acid secretion 0.850884406 1.181083 0.999999947 1 10

hsa04130 SNARE interactions in vesicular transport 0.856609981 1.166149092 0.999999958 1 23

hsa04110 Cell cycle 0.816463858 1.293514089 0.999999999 1 64

hsa00770 Pantothenate and CoA biosynthesis 0.882714031 1.367442579 1 1 10

hsa04114 Oocyte meiosis 0.861120216 1.336928384 1 1 49

hsa04621 NOD-like receptor signaling pathway 0.870466033 1.347229837 1 1 31

hsa00480 Glutathione metabolism 0.883171355 1.419136235 1 1 20

hsa04260 Cardiac muscle contraction 0.890214928 1.478909164 1 1 30

hsa04115 p53 signaling pathway 0.902428216 1.532608268 1 1 30

hsa00600 Sphingolipid metabolism 0.936396717 1.739475885 1 1 16

hsa00190 Oxidative phosphorylation 0.987908268 2.799196786 1 1 63

hsa03320 PPAR signaling pathway 0.962964986 2.041525896 1 1 17

hsa04610 Complement and coagulation cascades 0.937569232 1.925687416 1 1 20

hsa00040 Pentose and glucuronate interconversions NA NA NA NA 5

hsa00053 Ascorbate and aldarate metabolism NA NA NA NA 4

hsa00061 Fatty acid biosynthesis NA NA NA NA 1

hsa00072 Synthesis and degradation of ketone bodies NA NA NA NA 3

hsa00100 Steroid biosynthesis NA NA NA NA 8

hsa00120 Primary bile acid biosynthesis NA NA NA NA 2

hsa00130 Ubiquinone and other terpenoid-quinone biosynthesis NA NA NA NA 5

hsa00232 Caffeine metabolism NA NA NA NA 3

hsa00250 Alanine, aspartate and glutamate metabolism NA NA NA NA 8

hsa00260 Glycine, serine and threonine metabolism NA NA NA NA 9

hsa00290 Valine, leucine and isoleucine biosynthesis NA NA NA NA 3

hsa00300 Lysine biosynthesis NA NA NA NA 1

hsa00340 Histidine metabolism NA NA NA NA 7

hsa00350 Tyrosine metabolism NA NA NA NA 8

hsa00360 Phenylalanine metabolism NA NA NA NA 5

hsa00400 Phenylalanine, tyrosine and tryptophan biosynthesis NA NA NA NA 2

hsa00430 Taurine and hypotaurine metabolism NA NA NA NA 1

hsa00450 Selenocompound metabolism NA NA NA NA 4

hsa00460 Cyanoamino acid metabolism NA NA NA NA 3

hsa00471 D-Glutamine and D-glutamate metabolism NA NA NA NA 2

hsa00472 D-Arginine and D-ornithine metabolism NA NA NA NA 0

hsa00511 Other glycan degradation NA NA NA NA 7

hsa00531 Glycosaminoglycan degradation NA NA NA NA 7

hsa00533 Glycosaminoglycan biosynthesis - keratan sulfate NA NA NA NA 6

hsa00534 Glycosaminoglycan biosynthesis - heparan sulfate NA NA NA NA 8

hsa00591 Linoleic acid metabolism NA NA NA NA 6

hsa00592 alpha-Linolenic acid metabolism NA NA NA NA 4

hsa00601 Glycosphingolipid biosynthesis - lacto and neolacto series NA NA NA NA 9

hsa00603 Glycosphingolipid biosynthesis - globo series NA NA NA NA 6

hsa00604 Glycosphingolipid biosynthesis - ganglio series NA NA NA NA 8

hsa00630 Glyoxylate and dicarboxylate metabolism NA NA NA NA 9

hsa00650 Butanoate metabolism NA NA NA NA 5

hsa00730 Thiamine metabolism NA NA NA NA 1

hsa00740 Riboflavin metabolism NA NA NA NA 3

hsa00750 Vitamin B6 metabolism NA NA NA NA 2

hsa00780 Biotin metabolism NA NA NA NA 0

hsa00785 Lipoic acid metabolism NA NA NA NA 1

hsa00790 Folate biosynthesis NA NA NA NA 4

hsa00860 Porphyrin and chlorophyll metabolism NA NA NA NA 6

hsa00900 Terpenoid backbone biosynthesis NA NA NA NA 7

hsa00910 Nitrogen metabolism NA NA NA NA 7

hsa00920 Sulfur metabolism NA NA NA NA 4

hsa01040 Biosynthesis of unsaturated fatty acids NA NA NA NA 7

hsa02010 ABC transporters NA NA NA NA 9

hsa03440 Homologous recombination NA NA NA NA 8

hsa03450 Non-homologous end-joining NA NA NA NA 7

hsa04122 Sulfur relay system NA NA NA NA 2

hsa04320 Dorso-ventral axis formation NA NA NA NA 7

hsa04614 Renin-angiotensin system NA NA NA NA 5

hsa04744 Phototransduction NA NA NA NA 8

hsa04964 Proximal tubule bicarbonate reclamation NA NA NA NA 7

hsa04975 Fat digestion and absorption NA NA NA NA 9

hsa04977 Vitamin digestion and absorption NA NA NA NA 7

GSE64456 p.geomean stat.mean p.val q.val set.size

hsa04810 Regulation of actin cytoskeleton 0.024980698 1.900312857 1.38E-16 1.58E-14 71

hsa04380 Osteoclast differentiation 0.023553361 1.784762018 9.44E-15 5.38E-13 65

hsa04670 Leukocyte transendothelial migration 0.035758481 1.760501027 3.10E-14 1.18E-12 40

hsa04666 Fc gamma R-mediated phagocytosis 0.045133485 1.649307533 6.41E-13 1.83E-11 47

hsa04650 Natural killer cell mediated cytotoxicity 0.072225444 1.368939285 2.04E-09 4.64E-08 44

hsa04664 Fc epsilon RI signaling pathway 0.105157532 1.199401081 1.43E-07 2.50E-06 26

hsa04142 Lysosome 0.097778324 1.186075718 1.53E-07 2.50E-06 51

hsa04620 Toll-like receptor signaling pathway 0.111964363 1.141764999 4.56E-07 6.50E-06 36

hsa04510 Focal adhesion 0.125720461 1.109655708 7.59E-07 9.61E-06 64

hsa04010 MAPK signaling pathway 0.12788067 1.078335 1.42E-06 1.62E-05 97

hsa00500 Starch and sucrose metabolism 0.147212645 1.060429956 3.46E-06 3.59E-05 12

hsa04062 Chemokine signaling pathway 0.124300391 1.031280151 4.04E-06 3.84E-05 72

hsa04520 Adherens junction 0.156783987 0.993747894 9.14E-06 7.85E-05 31

hsa00051 Fructose and mannose metabolism 0.154399738 1.004180356 9.63E-06 7.85E-05 15

hsa04920 Adipocytokine signaling pathway 0.153578219 0.980627912 1.23E-05 9.33E-05 27

hsa00030 Pentose phosphate pathway 0.134763581 1.000753343 1.46E-05 0.000104057 13

hsa04910 Insulin signaling pathway 0.17037218 0.922536153 3.23E-05 0.000216303 50

hsa04530 Tight junction 0.177743253 0.889621278 5.93E-05 0.000375845 41

hsa00010 Glycolysis / Gluconeogenesis 0.165722382 0.881968186 7.36E-05 0.000441622 30

hsa04210 Apoptosis 0.180736565 0.849188966 0.000121182 0.00069074 42

hsa04145 Phagosome 0.143549486 0.840558448 0.000139443 0.000756978 72

hsa04722 Neurotrophin signaling pathway 0.200912653 0.798404883 0.000272888 0.001414054 49

hsa04971 Gastric acid secretion 0.217670806 0.761693351 0.000529168 0.002519363 21

hsa04976 Bile secretion 0.215490917 0.762650629 0.000530392 0.002519363 19

hsa03320 PPAR signaling pathway 0.198346671 0.750264896 0.000669556 0.002987829 19

hsa04144 Endocytosis 0.207710515 0.738460837 0.000681435 0.002987829 67

hsa00052 Galactose metabolism 0.230275804 0.731257184 0.000936234 0.003952986 11

hsa04973 Carbohydrate digestion and absorption 0.24191186 0.701405749 0.001407961 0.005732411 11

hsa04610 Complement and coagulation cascades 0.245060211 0.659420134 0.002422761 0.009523956 16

hsa04360 Axon guidance 0.248083663 0.634750607 0.002988767 0.011357316 39

hsa04621 NOD-like receptor signaling pathway 0.227490129 0.624613615 0.003626699 0.013336891 30

hsa04370 VEGF signaling pathway 0.273274253 0.582740232 0.005923901 0.021103896 25

hsa04730 Long-term depression 0.27623823 0.56216917 0.00769666 0.026588463 20

hsa00564 Glycerophospholipid metabolism 0.290202996 0.531137092 0.01098721 0.036839469 27

hsa00520 Amino sugar and nucleotide sugar metabolism 0.294049878 0.518430093 0.012883231 0.041962525 18

hsa00590 Arachidonic acid metabolism 0.283143841 0.510081145 0.014354733 0.045456654 15

hsa04630 Jak-STAT signaling pathway 0.235728391 0.500166708 0.01536124 0.047329226 50

hsa00983 Drug metabolism - other enzymes 0.318271059 0.458430004 0.02518885 0.075333421 11

hsa04115 p53 signaling pathway 0.311111894 0.450529732 0.02577196 0.075333421 28

hsa00600 Sphingolipid metabolism 0.319834909 0.447575713 0.027971255 0.079718078 15

hsa04540 Gap junction 0.305646155 0.433600064 0.030430748 0.084612325 31

hsa00860 Porphyrin and chlorophyll metabolism 0.286189662 0.421130695 0.037565435 0.101963323 11

hsa04150 mTOR signaling pathway 0.340950455 0.397427429 0.04350669 0.113117877 17

hsa04310 Wnt signaling pathway 0.331469051 0.394569796 0.043659531 0.113117877 33

hsa04662 B cell receptor signaling pathway 0.303354669 0.340558377 0.070081134 0.177538873 36

hsa00532 Glycosaminoglycan biosynthesis - chondroitin sulfate 0.368164857 0.324222579 0.083395241 0.206631215 10

hsa04141 Protein processing in endoplasmic reticulum 0.331189279 0.316661248 0.085190062 0.206631215 55

hsa03050 Proteasome 0.359031213 0.268075669 0.127058766 0.30176457 15

hsa04960 Aldosterone-regulated sodium reabsorption 0.390996471 0.265854694 0.130219016 0.302958528 10

hsa04146 Peroxisome 0.389150368 0.23346118 0.156056346 0.355808468 27

hsa04622 RIG-I-like receptor signaling pathway 0.363319151 0.230038908 0.160749313 0.359321993 19

hsa00562 Inositol phosphate metabolism 0.411370149 0.184126116 0.215220712 0.471830023 16

hsa04270 Vascular smooth muscle contraction 0.401505332 0.175943223 0.223055223 0.479779158 32

hsa04974 Protein digestion and absorption 0.389797521 0.148973549 0.261125277 0.551264473 20

hsa04623 Cytosolic DNA-sensing pathway 0.36800736 0.124878059 0.300262436 0.622362141 17

hsa04914 Progesterone-mediated oocyte maturation 0.44240046 0.103737445 0.326500689 0.664662117 37

hsa04512 ECM-receptor interaction 0.436009472 0.061837572 0.392897886 0.777391009 17

hsa04130 SNARE interactions in vesicular transport 0.459979913 0.059400836 0.399168389 0.777391009 17

hsa04320 Dorso-ventral axis formation 0.460532672 0.056790719 0.406367605 0.777391009 10

hsa00480 Glutathione metabolism 0.431579726 0.053231095 0.409153163 0.777391009 18

hsa04912 GnRH signaling pathway 0.479846099 0.033999205 0.441562404 0.824213327 28

hsa00330 Arginine and proline metabolism 0.452189972 0.027380649 0.452115639 0.824213327 17

hsa04640 Hematopoietic cell lineage 0.45550092 0.025942863 0.455486312 0.824213327 49

hsa04975 Fat digestion and absorption 0.483864467 0.013198276 0.477428233 0.850419041 11

hsa04962 Vasopressin-regulated water reabsorption 0.490358214 0.002697565 0.495347886 0.868763984 15

hsa00565 Ether lipid metabolism 0.49525664 -0.006552329 0.510954746 0.872449515 11

hsa04916 Melanogenesis 0.49429824 -0.007438543 0.512755417 0.872449515 25

hsa04972 Pancreatic secretion 0.504725422 -0.025481841 0.543806231 0.908570002 20

hsa00071 Fatty acid metabolism 0.481760323 -0.03007917 0.549923948 0.908570002 14

hsa04012 ErbB signaling pathway 0.509151509 -0.041425859 0.571024586 0.929954325 28

hsa04340 Hedgehog signaling pathway 0.515430457 -0.051318852 0.586236862 0.941281722 11

hsa04970 Salivary secretion 0.52262667 -0.071986586 0.621631808 0.984250362 16

hsa04740 Olfactory transduction 0.507316273 -0.078865103 0.632770843 0.988162687 26

hsa00534 Glycosaminoglycan biosynthesis - heparan sulfate 0.528288504 -0.093792889 0.655177429 1 13

hsa00561 Glycerolipid metabolism 0.533062754 -0.099841366 0.667028534 1 21

hsa00640 Propanoate metabolism 0.533575311 -0.125014024 0.704416472 1 12

hsa04070 Phosphatidylinositol signaling system 0.540308736 -0.132137598 0.715115518 1 21

hsa04720 Long-term potentiation 0.546122583 -0.133416835 0.717423883 1 23

hsa03022 Basal transcription factors 0.545041553 -0.159747559 0.751115205 1 13

hsa00310 Lysine degradation 0.557897978 -0.199252793 0.804005476 1 14

hsa04350 TGF-beta signaling pathway 0.56592326 -0.209497793 0.816282057 1 23

hsa00350 Tyrosine metabolism 0.574378348 -0.222558116 0.827279636 1 11

hsa00982 Drug metabolism - cytochrome P450 0.577643382 -0.258736332 0.864660474 1 10

hsa04110 Cell cycle 0.546692757 -0.261309618 0.870768802 1 41

hsa04114 Oocyte meiosis 0.592320415 -0.265090918 0.8745427 1 41

hsa03015 mRNA surveillance pathway 0.595717721 -0.297152628 0.89864673 1 20

hsa04020 Calcium signaling pathway 0.604055818 -0.295711026 0.899774828 1 38

hsa00380 Tryptophan metabolism 0.602509326 -0.314256078 0.909773844 1 11

hsa04330 Notch signaling pathway 0.609621025 -0.385076473 0.950341119 1 13

hsa03440 Homologous recombination 0.642460278 -0.418375221 0.961313798 1 10

hsa00240 Pyrimidine metabolism 0.646417304 -0.421613293 0.966005857 1 35

hsa00340 Histidine metabolism 0.649452139 -0.437050794 0.968744737 1 11

hsa00620 Pyruvate metabolism 0.626413943 -0.437595398 0.970432029 1 22

hsa00280 Valine, leucine and isoleucine degradation 0.651704934 -0.44308936 0.971465917 1 18

hsa00510 N-Glycan biosynthesis 0.655610754 -0.464390398 0.975679095 1 11

hsa04120 Ubiquitin mediated proteolysis 0.662805312 -0.508989698 0.985995059 1 38

hsa04660 T cell receptor signaling pathway 0.618600677 -0.531364514 0.989352139 1 52

hsa03018 RNA degradation 0.663623155 -0.558974754 0.99204445 1 31

hsa00514 Other types of O-glycan biosynthesis 0.699843767 -0.598736591 0.994492248 1 12

hsa03420 Nucleotide excision repair 0.698970853 -0.626920967 0.996132789 1 14

hsa04260 Cardiac muscle contraction 0.630022903 -0.635745402 0.996721052 1 19

hsa00020 Citrate cycle (TCA cycle) 0.703246737 -0.681577417 0.998044929 1 12

hsa03410 Base excision repair 0.738252551 -0.710759384 0.998729203 1 14

hsa03030 DNA replication 0.715694883 -0.722475813 0.998874325 1 13

hsa00970 Aminoacyl-tRNA biosynthesis 0.694849798 -0.765389945 0.999459538 1 16

hsa03008 Ribosome biogenesis in eukaryotes 0.718654093 -0.82675526 0.999797242 1 25

hsa00230 Purine metabolism 0.766696857 -0.926353324 0.99996939 1 53

hsa04514 Cell adhesion molecules (CAMs) 0.783995938 -1.153409151 0.999999634 1 43

hsa03040 Spliceosome 0.818204218 -1.211272899 0.999999911 1 55

hsa00190 Oxidative phosphorylation 0.518124003 -1.286456727 0.99999998 1 50

hsa03013 RNA transport 0.81606492 -1.4303563 1 1 52

hsa04612 Antigen processing and presentation 0.846477013 -1.471357121 1 1 35

hsa04672 Intestinal immune network for IgA production 0.920134709 -1.923000744 1 1 28

hsa03010 Ribosome 0.67474983 -4.82383644 1 1 44

hsa00040 Pentose and glucuronate interconversions NA NA NA NA 6

hsa00053 Ascorbate and aldarate metabolism NA NA NA NA 4

hsa00061 Fatty acid biosynthesis NA NA NA NA 2

hsa00072 Synthesis and degradation of ketone bodies NA NA NA NA 1

hsa00100 Steroid biosynthesis NA NA NA NA 6

hsa00120 Primary bile acid biosynthesis NA NA NA NA 2

hsa00130 Ubiquinone and other terpenoid-quinone biosynthesis NA NA NA NA 4

hsa00140 Steroid hormone biosynthesis NA NA NA NA 5

hsa00232 Caffeine metabolism NA NA NA NA 0

hsa00250 Alanine, aspartate and glutamate metabolism NA NA NA NA 7

hsa00260 Glycine, serine and threonine metabolism NA NA NA NA 9

hsa00270 Cysteine and methionine metabolism NA NA NA NA 9

hsa00290 Valine, leucine and isoleucine biosynthesis NA NA NA NA 7

hsa00300 Lysine biosynthesis NA NA NA NA 0

hsa00360 Phenylalanine metabolism NA NA NA NA 5

hsa00400 Phenylalanine, tyrosine and tryptophan biosynthesis NA NA NA NA 1

hsa00410 beta-Alanine metabolism NA NA NA NA 7

hsa00430 Taurine and hypotaurine metabolism NA NA NA NA 1

hsa00450 Selenocompound metabolism NA NA NA NA 6

hsa00460 Cyanoamino acid metabolism NA NA NA NA 3

hsa00471 D-Glutamine and D-glutamate metabolism NA NA NA NA 0

hsa00472 D-Arginine and D-ornithine metabolism NA NA NA NA 0

hsa00511 Other glycan degradation NA NA NA NA 2

hsa00512 Mucin type O-Glycan biosynthesis NA NA NA NA 8

hsa00531 Glycosaminoglycan degradation NA NA NA NA 7

hsa00533 Glycosaminoglycan biosynthesis - keratan sulfate NA NA NA NA 4

hsa00563 Glycosylphosphatidylinositol(GPI)-anchor biosynthesis NA NA NA NA 5

hsa00591 Linoleic acid metabolism NA NA NA NA 4

hsa00592 alpha-Linolenic acid metabolism NA NA NA NA 3

hsa00601 Glycosphingolipid biosynthesis - lacto and neolacto series NA NA NA NA 9

hsa00603 Glycosphingolipid biosynthesis - globo series NA NA NA NA 4

hsa00604 Glycosphingolipid biosynthesis - ganglio series NA NA NA NA 6

hsa00630 Glyoxylate and dicarboxylate metabolism NA NA NA NA 8

hsa00650 Butanoate metabolism NA NA NA NA 7

hsa00670 One carbon pool by folate NA NA NA NA 5

hsa00730 Thiamine metabolism NA NA NA NA 1

hsa00740 Riboflavin metabolism NA NA NA NA 5

hsa00750 Vitamin B6 metabolism NA NA NA NA 2

hsa00760 Nicotinate and nicotinamide metabolism NA NA NA NA 5

hsa00770 Pantothenate and CoA biosynthesis NA NA NA NA 8

hsa00780 Biotin metabolism NA NA NA NA 0

hsa00785 Lipoic acid metabolism NA NA NA NA 2

hsa00790 Folate biosynthesis NA NA NA NA 3

hsa00830 Retinol metabolism NA NA NA NA 5

hsa00900 Terpenoid backbone biosynthesis NA NA NA NA 4

hsa00910 Nitrogen metabolism NA NA NA NA 5

hsa00920 Sulfur metabolism NA NA NA NA 6

hsa00980 Metabolism of xenobiotics by cytochrome P450 NA NA NA NA 9

hsa01040 Biosynthesis of unsaturated fatty acids NA NA NA NA 9

hsa02010 ABC transporters NA NA NA NA 7

hsa03020 RNA polymerase NA NA NA NA 8

hsa03060 Protein export NA NA NA NA 7

hsa03430 Mismatch repair NA NA NA NA 7

hsa03450 Non-homologous end-joining NA NA NA NA 3

hsa04122 Sulfur relay system NA NA NA NA 1

hsa04140 Regulation of autophagy NA NA NA NA 6

hsa04614 Renin-angiotensin system NA NA NA NA 5

hsa04710 Circadian rhythm - mammal NA NA NA NA 5

hsa04742 Taste transduction NA NA NA NA 8

hsa04744 Phototransduction NA NA NA NA 4

hsa04964 Proximal tubule bicarbonate reclamation NA NA NA NA 4

hsa04966 Collecting duct acid secretion NA NA NA NA 9

hsa04977 Vitamin digestion and absorption NA NA NA NA 4

hsa03010 Ribosome 7.77E-07 -4.82383644 3.43E-78 3.91E-76 44

hsa04672 Intestinal immune network for IgA production 0.025613824 -1.923000744 2.27E-16 1.29E-14 28

hsa04612 Antigen processing and presentation 0.059175179 -1.471357121 1.65E-10 6.27E-09 35

hsa03013 RNA transport 0.062799083 -1.4303563 3.80E-10 1.08E-08 52

hsa00190 Oxidative phosphorylation 0.039003681 -1.286456727 2.00E-08 4.56E-07 50

hsa03040 Spliceosome 0.097913217 -1.211272899 8.89E-08 1.69E-06 55

hsa04514 Cell adhesion molecules (CAMs) 0.101366727 -1.153409151 3.66E-07 5.95E-06 43

hsa00230 Purine metabolism 0.159594793 -0.926353324 3.06E-05 0.000436199 53

hsa03008 Ribosome biogenesis in eukaryotes 0.182118354 -0.82675526 0.000202758 0.002568272 25

hsa00970 Aminoacyl-tRNA biosynthesis 0.199499524 -0.765389945 0.000540462 0.006161265 16

hsa03030 DNA replication 0.22562444 -0.722475813 0.001125675 0.011666083 13

hsa03410 Base excision repair 0.236561864 -0.710759384 0.001270797 0.012072575 14

hsa00020 Citrate cycle (TCA cycle) 0.234045694 -0.681577417 0.001955071 0.017144473 12

hsa04260 Cardiac muscle contraction 0.218658227 -0.635745402 0.003278948 0.026700001 19

hsa03420 Nucleotide excision repair 0.257102696 -0.626920967 0.003867211 0.029390805 14

hsa00514 Other types of O-glycan biosynthesis 0.268746322 -0.598736591 0.005507752 0.039242736 12

hsa03018 RNA degradation 0.264004808 -0.558974754 0.00795555 0.053348984 31

hsa04660 T cell receptor signaling pathway 0.256566763 -0.531364514 0.010647861 0.067436456 52

hsa04120 Ubiquitin mediated proteolysis 0.289484996 -0.508989698 0.014004941 0.084029648 38

hsa00510 N-Glycan biosynthesis 0.314209112 -0.464390398 0.024320905 0.13862916 11

hsa00280 Valine, leucine and isoleucine degradation 0.320869632 -0.44308936 0.028534083 0.153215852 18

hsa00620 Pyruvate metabolism 0.308438214 -0.437595398 0.029567971 0.153215852 22

hsa00340 Histidine metabolism 0.324736621 -0.437050794 0.031255263 0.154917388 11

hsa00240 Pyrimidine metabolism 0.327540915 -0.421613293 0.033994143 0.161472178 35

hsa03440 Homologous recombination 0.333293763 -0.418375221 0.038686202 0.176409082 10

hsa04330 Notch signaling pathway 0.32867722 -0.385076473 0.049658881 0.217735092 13

hsa00380 Tryptophan metabolism 0.366118955 -0.314256078 0.090226156 0.380954879 11

hsa04020 Calcium signaling pathway 0.376159732 -0.295711026 0.100225172 0.398423199 38

hsa03015 mRNA surveillance pathway 0.370820501 -0.297152628 0.10135327 0.398423199 20

hsa04114 Oocyte meiosis 0.387222526 -0.265090918 0.1254573 0.475237309 41

hsa04110 Cell cycle 0.355021303 -0.261309618 0.129231198 0.475237309 41

hsa00982 Drug metabolism - cytochrome P450 0.383153567 -0.258736332 0.135339526 0.48214706 10

hsa00350 Tyrosine metabolism 0.40522162 -0.222558116 0.172720364 0.596670348 11

hsa04350 TGF-beta signaling pathway 0.404982156 -0.209497793 0.183717943 0.615995455 23

hsa00310 Lysine degradation 0.406758356 -0.199252793 0.195994524 0.638382163 14

hsa03022 Basal transcription factors 0.423949328 -0.159747559 0.248884795 0.788135185 13

hsa04720 Long-term potentiation 0.442021952 -0.133416835 0.282576117 0.854653447 23

hsa04070 Phosphatidylinositol signaling system 0.437884934 -0.132137598 0.284884482 0.854653447 21

hsa00640 Propanoate metabolism 0.437630008 -0.125014024 0.295583528 0.864013388 12

hsa00561 Glycerolipid metabolism 0.454933697 -0.099841366 0.332971466 0.948968679 21

hsa00534 Glycosaminoglycan biosynthesis - heparan sulfate 0.455986632 -0.093792889 0.344822571 0.958774953 13

hsa04740 Olfactory transduction 0.447156016 -0.078865103 0.367229157 0.996764854 26

hsa04970 Salivary secretion 0.466445299 -0.071986586 0.378368192 1 16

hsa04340 Hedgehog signaling pathway 0.475748551 -0.051318852 0.413763138 1 11

hsa04012 ErbB signaling pathway 0.476830067 -0.041425859 0.428975414 1 28

hsa00071 Fatty acid metabolism 0.459233451 -0.03007917 0.450076052 1 14

hsa04972 Pancreatic secretion 0.484830862 -0.025481841 0.456193769 1 20

hsa04916 Melanogenesis 0.488260643 -0.007438543 0.487244583 1 25

hsa00565 Ether lipid metabolism 0.490277284 -0.006552329 0.489045254 1 11

hsa04962 Vasopressin-regulated water reabsorption 0.492493948 0.002697565 0.504652114 1 15

hsa04975 Fat digestion and absorption 0.494143544 0.013198276 0.522571767 1 11

hsa04640 Hematopoietic cell lineage 0.475761832 0.025942863 0.544513688 1 49

hsa00330 Arginine and proline metabolism 0.472275028 0.027380649 0.547884361 1 17

hsa04912 GnRH signaling pathway 0.506558087 0.033999205 0.558437596 1 28

hsa00480 Glutathione metabolism 0.469997712 0.053231095 0.590846837 1 18

hsa04320 Dorso-ventral axis formation 0.503600092 0.056790719 0.593632395 1 10

hsa04130 SNARE interactions in vesicular transport 0.506161127 0.059400836 0.600831611 1 17

hsa04512 ECM-receptor interaction 0.481232165 0.061837572 0.607102114 1 17

hsa04914 Progesterone-mediated oocyte maturation 0.523878029 0.103737445 0.673499311 1 37

hsa04623 Cytosolic DNA-sensing pathway 0.455529536 0.124878059 0.699737564 1 17

hsa04974 Protein digestion and absorption 0.497807315 0.148973549 0.738874723 1 20

hsa04270 Vascular smooth muscle contraction 0.5332949 0.175943223 0.776944777 1 32

hsa00562 Inositol phosphate metabolism 0.550981155 0.184126116 0.784779288 1 16

hsa04622 RIG-I-like receptor signaling pathway 0.528116453 0.230038908 0.839250687 1 19

hsa04146 Peroxisome 0.567072048 0.23346118 0.843943654 1 27

hsa04960 Aldosterone-regulated sodium reabsorption 0.592768032 0.265854694 0.869780984 1 10

hsa03050 Proteasome 0.557960616 0.268075669 0.872941234 1 15

hsa04141 Protein processing in endoplasmic reticulum 0.564622289 0.316661248 0.914809938 1 55

hsa00532 Glycosaminoglycan biosynthesis - chondroitin sulfate 0.613941996 0.324222579 0.916604759 1 10

hsa04662 B cell receptor signaling pathway 0.524227981 0.340558377 0.929918866 1 36

hsa04310 Wnt signaling pathway 0.627708378 0.394569796 0.956340469 1 33

hsa04150 mTOR signaling pathway 0.643424277 0.397427429 0.95649331 1 17

hsa00860 Porphyrin and chlorophyll metabolism 0.565545027 0.421130695 0.962434565 1 11

hsa04540 Gap junction 0.62150259 0.433600064 0.969569252 1 31

hsa00600 Sphingolipid metabolism 0.652074913 0.447575713 0.972028745 1 15

hsa04115 p53 signaling pathway 0.646496094 0.450529732 0.97422804 1 28

hsa00983 Drug metabolism - other enzymes 0.659874293 0.458430004 0.97481115 1 11

hsa04630 Jak-STAT signaling pathway 0.558502179 0.500166708 0.98463876 1 50

hsa00590 Arachidonic acid metabolism 0.646474592 0.510081145 0.985645267 1 15

hsa00520 Amino sugar and nucleotide sugar metabolism 0.679184835 0.518430093 0.987116769 1 18

hsa00564 Glycerophospholipid metabolism 0.683459054 0.531137092 0.98901279 1 27

hsa04730 Long-term depression 0.686237904 0.56216917 0.99230334 1 20

hsa04370 VEGF signaling pathway 0.702699545 0.582740232 0.994076099 1 25

hsa04621 NOD-like receptor signaling pathway 0.65854571 0.624613615 0.996373301 1 30

hsa04360 Axon guidance 0.704632114 0.634750607 0.997011233 1 39

hsa04610 Complement and coagulation cascades 0.711659505 0.659420134 0.997577239 1 16

hsa04973 Carbohydrate digestion and absorption 0.744492304 0.701405749 0.998592039 1 11

hsa00052 Galactose metabolism 0.74752235 0.731257184 0.999063766 1 11

hsa04144 Endocytosis 0.719766219 0.738460837 0.999318565 1 67

hsa03320 PPAR signaling pathway 0.694428157 0.750264896 0.999330444 1 19

hsa04976 Bile secretion 0.744542264 0.762650629 0.999469608 1 19

hsa04971 Gastric acid secretion 0.752953653 0.761693351 0.999470832 1 21

hsa04722 Neurotrophin signaling pathway 0.75834553 0.798404883 0.999727112 1 49

hsa04145 Phagosome 0.626506377 0.840558448 0.999860557 1 72

hsa04210 Apoptosis 0.752340516 0.849188966 0.999878818 1 42

hsa00010 Glycolysis / Gluconeogenesis 0.742930964 0.881968186 0.999926396 1 30

hsa04530 Tight junction 0.782808536 0.889621278 0.999940656 1 41

hsa04910 Insulin signaling pathway 0.796855098 0.922536153 0.999967744 1 50

hsa00030 Pentose phosphate pathway 0.727141546 1.000753343 0.999985395 1 13

hsa04920 Adipocytokine signaling pathway 0.794788087 0.980627912 0.99998773 1 27

hsa00051 Fructose and mannose metabolism 0.813247994 1.004180356 0.999990366 1 15

hsa04520 Adherens junction 0.819767155 0.993747894 0.999990863 1 31

hsa04062 Chemokine signaling pathway 0.767920201 1.031280151 0.99999596 1 72

hsa00500 Starch and sucrose metabolism 0.837331324 1.060429956 0.999996538 1 12

hsa04010 MAPK signaling pathway 0.81590566 1.078335 0.999998581 1 97

hsa04510 Focal adhesion 0.835612855 1.109655708 0.999999241 1 64

hsa04620 Toll-like receptor signaling pathway 0.81025081 1.141764999 0.999999544 1 36

hsa04142 Lysosome 0.79660643 1.186075718 0.999999847 1 51

hsa04664 Fc epsilon RI signaling pathway 0.814987923 1.199401081 0.999999857 1 26

hsa04650 Natural killer cell mediated cytotoxicity 0.815759204 1.368939285 0.999999998 1 44

hsa04666 Fc gamma R-mediated phagocytosis 0.907783818 1.649307533 1 1 47

hsa04670 Leukocyte transendothelial migration 0.898114304 1.760501027 1 1 40

hsa04380 Osteoclast differentiation 0.777852778 1.784762018 1 1 65

hsa04810 Regulation of actin cytoskeleton 0.916930682 1.900312857 1 1 71

hsa00040 Pentose and glucuronate interconversions NA NA NA NA 6

hsa00053 Ascorbate and aldarate metabolism NA NA NA NA 4

hsa00061 Fatty acid biosynthesis NA NA NA NA 2

hsa00072 Synthesis and degradation of ketone bodies NA NA NA NA 1

hsa00100 Steroid biosynthesis NA NA NA NA 6

hsa00120 Primary bile acid biosynthesis NA NA NA NA 2

hsa00130 Ubiquinone and other terpenoid-quinone biosynthesis NA NA NA NA 4

hsa00140 Steroid hormone biosynthesis NA NA NA NA 5

hsa00232 Caffeine metabolism NA NA NA NA 0

hsa00250 Alanine, aspartate and glutamate metabolism NA NA NA NA 7

hsa00260 Glycine, serine and threonine metabolism NA NA NA NA 9

hsa00270 Cysteine and methionine metabolism NA NA NA NA 9

hsa00290 Valine, leucine and isoleucine biosynthesis NA NA NA NA 7

hsa00300 Lysine biosynthesis NA NA NA NA 0

hsa00360 Phenylalanine metabolism NA NA NA NA 5

hsa00400 Phenylalanine, tyrosine and tryptophan biosynthesis NA NA NA NA 1

hsa00410 beta-Alanine metabolism NA NA NA NA 7

hsa00430 Taurine and hypotaurine metabolism NA NA NA NA 1

hsa00450 Selenocompound metabolism NA NA NA NA 6

hsa00460 Cyanoamino acid metabolism NA NA NA NA 3

hsa00471 D-Glutamine and D-glutamate metabolism NA NA NA NA 0

hsa00472 D-Arginine and D-ornithine metabolism NA NA NA NA 0

hsa00511 Other glycan degradation NA NA NA NA 2

hsa00512 Mucin type O-Glycan biosynthesis NA NA NA NA 8

hsa00531 Glycosaminoglycan degradation NA NA NA NA 7

hsa00533 Glycosaminoglycan biosynthesis - keratan sulfate NA NA NA NA 4

hsa00563 Glycosylphosphatidylinositol(GPI)-anchor biosynthesis NA NA NA NA 5

hsa00591 Linoleic acid metabolism NA NA NA NA 4

hsa00592 alpha-Linolenic acid metabolism NA NA NA NA 3

hsa00601 Glycosphingolipid biosynthesis - lacto and neolacto series NA NA NA NA 9

hsa00603 Glycosphingolipid biosynthesis - globo series NA NA NA NA 4

hsa00604 Glycosphingolipid biosynthesis - ganglio series NA NA NA NA 6

hsa00630 Glyoxylate and dicarboxylate metabolism NA NA NA NA 8

hsa00650 Butanoate metabolism NA NA NA NA 7

hsa00670 One carbon pool by folate NA NA NA NA 5

hsa00730 Thiamine metabolism NA NA NA NA 1

hsa00740 Riboflavin metabolism NA NA NA NA 5

hsa00750 Vitamin B6 metabolism NA NA NA NA 2

hsa00760 Nicotinate and nicotinamide metabolism NA NA NA NA 5

hsa00770 Pantothenate and CoA biosynthesis NA NA NA NA 8

hsa00780 Biotin metabolism NA NA NA NA 0

hsa00785 Lipoic acid metabolism NA NA NA NA 2

hsa00790 Folate biosynthesis NA NA NA NA 3

hsa00830 Retinol metabolism NA NA NA NA 5

hsa00900 Terpenoid backbone biosynthesis NA NA NA NA 4

hsa00910 Nitrogen metabolism NA NA NA NA 5

hsa00920 Sulfur metabolism NA NA NA NA 6

hsa00980 Metabolism of xenobiotics by cytochrome P450 NA NA NA NA 9

hsa01040 Biosynthesis of unsaturated fatty acids NA NA NA NA 9

hsa02010 ABC transporters NA NA NA NA 7

hsa03020 RNA polymerase NA NA NA NA 8

hsa03060 Protein export NA NA NA NA 7

hsa03430 Mismatch repair NA NA NA NA 7

hsa03450 Non-homologous end-joining NA NA NA NA 3

hsa04122 Sulfur relay system NA NA NA NA 1

hsa04140 Regulation of autophagy NA NA NA NA 6

hsa04614 Renin-angiotensin system NA NA NA NA 5

hsa04710 Circadian rhythm - mammal NA NA NA NA 5

hsa04742 Taste transduction NA NA NA NA 8

hsa04744 Phototransduction NA NA NA NA 4

hsa04964 Proximal tubule bicarbonate reclamation NA NA NA NA 4

hsa04966 Collecting duct acid secretion NA NA NA NA 9

hsa04977 Vitamin digestion and absorption NA NA NA NA 4

GSE66099 p.geomean stat.mean p.val q.val set.size

hsa04610 Complement and coagulation cascades 0.003705049 2.627115289 3.01E-68 3.85E-66 27

hsa04142 Lysosome 0.01266978 2.027149078 2.33E-44 1.49E-42 59

hsa04380 Osteoclast differentiation 0.012269396 1.948024403 3.73E-41 1.59E-39 65

hsa04810 Regulation of actin cytoskeleton 0.017974084 1.886943364 3.68E-39 1.18E-37 77

hsa04666 Fc gamma R-mediated phagocytosis 0.029131543 1.692698376 1.66E-31 4.26E-30 45

hsa03320 PPAR signaling pathway 0.050250178 1.597233745 6.17E-28 1.32E-26 24

hsa04621 NOD-like receptor signaling pathway 0.049312688 1.507315834 5.97E-25 1.07E-23 25

hsa04510 Focal adhesion 0.042514625 1.478517677 6.67E-25 1.07E-23 76

hsa00010 Glycolysis / Gluconeogenesis 0.065143465 1.438619389 2.31E-23 3.28E-22 31

hsa00983 Drug metabolism - other enzymes 0.050625477 1.452537004 8.92E-23 1.14E-21 23

hsa04620 Toll-like receptor signaling pathway 0.054148099 1.404108552 3.23E-22 3.76E-21 35

hsa04910 Insulin signaling pathway 0.060899682 1.37837677 6.64E-22 7.08E-21 61

hsa04145 Phagosome 0.052262049 1.302658365 1.04E-19 1.02E-18 73

hsa04512 ECM-receptor interaction 0.060938568 1.290258349 6.09E-19 5.57E-18 33

hsa04740 Olfactory transduction 0.043112696 1.286221584 2.75E-18 2.29E-17 29

hsa00052 Galactose metabolism 0.093603146 1.272071517 2.87E-18 2.29E-17 16

hsa00531 Glycosaminoglycan degradation 0.098164887 1.257329942 7.38E-17 5.55E-16 10

hsa00051 Fructose and mannose metabolism 0.11236772 1.172167665 6.24E-16 4.44E-15 17

hsa00480 Glutathione metabolism 0.115759676 1.110178902 1.18E-14 7.92E-14 24

hsa04920 Adipocytokine signaling pathway 0.121086301 1.089798802 2.54E-14 1.58E-13 33

hsa00190 Oxidative phosphorylation 0.105760971 1.089125289 2.60E-14 1.58E-13 44

hsa04020 Calcium signaling pathway 0.098336386 1.071985624 5.23E-14 3.04E-13 64

hsa00982 Drug metabolism - cytochrome P450 0.081431459 1.087563285 1.44E-13 7.93E-13 23

hsa04010 MAPK signaling pathway 0.10579043 1.04808766 1.49E-13 7.93E-13 98

hsa04670 Leukocyte transendothelial migration 0.107371889 1.039236046 3.31E-13 1.70E-12 46

hsa04144 Endocytosis 0.115257715 0.971773613 6.94E-12 3.42E-11 81

hsa00564 Glycerophospholipid metabolism 0.129797271 0.976771438 7.98E-12 3.78E-11 31

hsa00140 Steroid hormone biosynthesis 0.108621819 0.999920789 8.37E-12 3.83E-11 18

hsa00860 Porphyrin and chlorophyll metabolism 0.134304384 0.982959373 1.26E-11 5.54E-11 19

hsa00980 Metabolism of xenobiotics by cytochrome P450 0.121269955 0.973182501 1.51E-11 6.46E-11 23

hsa00600 Sphingolipid metabolism 0.143705695 0.958750688 3.00E-11 1.24E-10 17

hsa00030 Pentose phosphate pathway 0.150939894 0.952746064 3.82E-11 1.53E-10 17

hsa04320 Dorso-ventral axis formation 0.160274512 0.935747774 1.42E-10 5.52E-10 11

hsa04540 Gap junction 0.140440959 0.894186191 3.45E-10 1.30E-09 36

hsa04114 Oocyte meiosis 0.133707655 0.888103923 3.92E-10 1.43E-09 50

hsa04130 SNARE interactions in vesicular transport 0.150579419 0.883380558 1.01E-09 3.61E-09 16

hsa04270 Vascular smooth muscle contraction 0.176069445 0.8137631 8.35E-09 2.89E-08 41

hsa04912 GnRH signaling pathway 0.170971979 0.808486958 1.05E-08 3.55E-08 39

hsa04914 Progesterone-mediated oocyte maturation 0.172100632 0.804150142 1.26E-08 4.13E-08 42

hsa00512 Mucin type O-Glycan biosynthesis 0.19059102 0.810585824 1.52E-08 4.85E-08 14

hsa04722 Neurotrophin signaling pathway 0.165526478 0.795151256 1.72E-08 5.37E-08 49

hsa00500 Starch and sucrose metabolism 0.129188063 0.76812641 8.44E-08 2.57E-07 29

hsa00760 Nicotinate and nicotinamide metabolism 0.211669604 0.753077767 1.68E-07 5.02E-07 10

hsa04966 Collecting duct acid secretion 0.207210052 0.740427687 2.33E-07 6.77E-07 12

hsa00330 Arginine and proline metabolism 0.208270356 0.729924788 2.39E-07 6.79E-07 23

hsa04141 Protein processing in endoplasmic reticulum 0.189196825 0.715726473 3.23E-07 8.99E-07 60

hsa04115 p53 signaling pathway 0.212291319 0.695100274 7.24E-07 1.97E-06 35

hsa04370 VEGF signaling pathway 0.200775123 0.695719348 7.84E-07 2.09E-06 29

hsa04622 RIG-I-like receptor signaling pathway 0.224190694 0.683065578 1.29E-06 3.32E-06 18

hsa04976 Bile secretion 0.196657044 0.683424679 1.30E-06 3.32E-06 24

hsa04630 Jak-STAT signaling pathway 0.200846678 0.648508842 3.25E-06 8.15E-06 59

hsa00520 Amino sugar and nucleotide sugar metabolism 0.222656696 0.622511145 8.59E-06 2.11E-05 29

hsa04360 Axon guidance 0.21054589 0.591536051 2.00E-05 4.82E-05 52

hsa04260 Cardiac muscle contraction 0.235501332 0.590618725 2.36E-05 5.59E-05 24

hsa04975 Fat digestion and absorption 0.25565959 0.578036426 4.33E-05 0.000100817 10

hsa04664 Fc epsilon RI signaling pathway 0.234623663 0.560978645 5.14E-05 0.000117457 31

hsa04210 Apoptosis 0.234016116 0.555656094 5.64E-05 0.000126666 46

hsa00830 Retinol metabolism 0.169950294 0.563733787 5.83E-05 0.000128713 26

hsa03050 Proteasome 0.23694637 0.543064357 0.000124335 0.000266461 13

hsa04530 Tight junction 0.263604072 0.526600436 0.000124904 0.000266461 44

hsa00514 Other types of O-glycan biosynthesis 0.237020502 0.4881476 0.000418067 0.000877256 18

hsa04012 ErbB signaling pathway 0.275335156 0.460884513 0.00069433 0.001433455 33

hsa00310 Lysine degradation 0.302109833 0.463259641 0.000740288 0.001504077 13

hsa00380 Tryptophan metabolism 0.307501964 0.448949223 0.001038137 0.002076273 13

hsa04971 Gastric acid secretion 0.280461857 0.439202628 0.001193463 0.002350204 26

hsa04110 Cell cycle 0.174672074 0.438409807 0.001290934 0.00250363 67

hsa00053 Ascorbate and aldarate metabolism 0.244022689 0.446737097 0.001436722 0.002744782 12

hsa04520 Adherens junction 0.280661701 0.415702695 0.001965147 0.0036991 31

hsa04916 Melanogenesis 0.301552839 0.362558429 0.00600789 0.011145072 33

hsa00565 Ether lipid metabolism 0.323048551 0.361186826 0.006930128 0.012672234 11

hsa04962 Vasopressin-regulated water reabsorption 0.334798032 0.341723509 0.009086183 0.016380725 19

hsa00040 Pentose and glucuronate interconversions 0.289787808 0.345056996 0.009290416 0.016516295 15

hsa00561 Glycerolipid metabolism 0.335207685 0.332713947 0.010704146 0.018768913 19

hsa04150 mTOR signaling pathway 0.337606934 0.320797255 0.01321614 0.022860349 23

hsa04742 Taste transduction 0.291028874 0.318718039 0.014402291 0.024579911 16

hsa04730 Long-term depression 0.332568569 0.306091725 0.017434524 0.029363409 21

hsa00590 Arachidonic acid metabolism 0.343245314 0.263932337 0.034058501 0.056616729 20

hsa04623 Cytosolic DNA-sensing pathway 0.354542664 0.24402704 0.04715838 0.077388111 15

hsa04330 Notch signaling pathway 0.35982807 0.192659157 0.091907442 0.148913325 20

hsa03440 Homologous recombination 0.374767582 0.189320952 0.098329508 0.157327212 14

hsa04062 Chemokine signaling pathway 0.292295649 0.15879599 0.133791923 0.211424273 89

hsa04960 Aldosterone-regulated sodium reabsorption 0.390562423 0.147898997 0.1562751 0.243941619 13

hsa04973 Carbohydrate digestion and absorption 0.331954464 0.146179308 0.158477206 0.244398583 21

hsa04146 Peroxisome 0.372717362 0.119390199 0.204114911 0.311032246 38

hsa00534 Glycosaminoglycan biosynthesis - heparan sulfate 0.442049424 0.093139652 0.261343285 0.393552241 10

hsa00532 Glycosaminoglycan biosynthesis - chondroitin sulfate 0.437849575 0.066691854 0.322068246 0.479357389 12

hsa04720 Long-term potentiation 0.440235062 0.039588775 0.391737188 0.576348967 24

hsa04972 Pancreatic secretion 0.428445132 0.032656772 0.409181868 0.595173626 33

hsa00071 Fatty acid metabolism 0.413265823 0.022840238 0.437035673 0.628545687 20

hsa04974 Protein digestion and absorption 0.400587787 -0.00438383 0.512075141 0.728284645 19

hsa04310 Wnt signaling pathway 0.437700838 -0.072120803 0.692043725 0.973424141 57

hsa04662 B cell receptor signaling pathway 0.36103007 -0.07710529 0.703307374 0.978514608 39

hsa04970 Salivary secretion 0.512955548 -0.202513164 0.919414566 1 30

hsa00562 Inositol phosphate metabolism 0.545419933 -0.209184316 0.925845588 1 20

hsa00340 Histidine metabolism 0.544360095 -0.232828907 0.943784408 1 11

hsa00350 Tyrosine metabolism 0.592645522 -0.30800538 0.982976014 1 14

hsa00240 Pyrimidine metabolism 0.535418227 -0.304257045 0.983005963 1 45

hsa00640 Propanoate metabolism 0.567389887 -0.32318709 0.9869299 1 15

hsa02010 ABC transporters 0.571371839 -0.33534975 0.98958063 1 19

hsa04120 Ubiquitin mediated proteolysis 0.477484887 -0.33461628 0.990023117 1 60

hsa04350 TGF-beta signaling pathway 0.586336612 -0.347386422 0.99214281 1 38

hsa00620 Pyruvate metabolism 0.543545294 -0.363893801 0.993642594 1 16

hsa00020 Citrate cycle (TCA cycle) 0.548225146 -0.373598925 0.994268151 1 10

hsa00270 Cysteine and methionine metabolism 0.582681441 -0.366283307 0.994290773 1 18

hsa04070 Phosphatidylinositol signaling system 0.621304624 -0.456833384 0.999213644 1 26

hsa04640 Hematopoietic cell lineage 0.560465997 -0.468457781 0.999437017 1 51

hsa00230 Purine metabolism 0.624657536 -0.501315713 0.999763164 1 71

hsa03060 Protein export 0.573804743 -0.547418273 0.999875767 1 12

hsa00563 Glycosylphosphatidylinositol(GPI)-anchor biosynthesis 0.679893069 -0.765756219 0.999999882 1 12

hsa03410 Base excision repair 0.649822419 -0.782178347 0.999999944 1 15

hsa03022 Basal transcription factors 0.694202219 -0.787481501 0.999999947 1 13

hsa03430 Mismatch repair 0.672068333 -0.802861444 0.999999969 1 12

hsa03015 mRNA surveillance pathway 0.738013337 -0.875760598 0.999999999 1 40

hsa00510 N-Glycan biosynthesis 0.735702206 -0.926550133 1 1 20

hsa00280 Valine, leucine and isoleucine degradation 0.684717543 -0.95942875 1 1 20

hsa03420 Nucleotide excision repair 0.637905123 -0.963139829 1 1 22

hsa03030 DNA replication 0.610906963 -0.976498889 1 1 18

hsa00970 Aminoacyl-tRNA biosynthesis 0.708900242 -0.989717566 1 1 13

hsa04514 Cell adhesion molecules (CAMs) 0.664625853 -1.052692299 1 1 59

hsa04660 T cell receptor signaling pathway 0.665854995 -1.108616638 1 1 56

hsa04650 Natural killer cell mediated cytotoxicity 0.654357518 -1.116430068 1 1 49

hsa03008 Ribosome biogenesis in eukaryotes 0.616574079 -1.913281458 1 1 48

hsa03010 Ribosome 0.478525766 -1.58007293 1 1 52

hsa03013 RNA transport 0.628940109 -2.356686231 1 1 78

hsa03018 RNA degradation 0.722605049 -1.260348082 1 1 35

hsa03040 Spliceosome 0.682745561 -1.611086829 1 1 58

hsa04612 Antigen processing and presentation 0.834803569 -2.239990543 1 1 40

hsa04672 Intestinal immune network for IgA production 0.790429155 -1.866945746 1 1 31

hsa00061 Fatty acid biosynthesis NA NA NA NA 2

hsa00072 Synthesis and degradation of ketone bodies NA NA NA NA 3

hsa00100 Steroid biosynthesis NA NA NA NA 6

hsa00120 Primary bile acid biosynthesis NA NA NA NA 3

hsa00130 Ubiquinone and other terpenoid-quinone biosynthesis NA NA NA NA 4

hsa00232 Caffeine metabolism NA NA NA NA 1

hsa00250 Alanine, aspartate and glutamate metabolism NA NA NA NA 9

hsa00260 Glycine, serine and threonine metabolism NA NA NA NA 7

hsa00290 Valine, leucine and isoleucine biosynthesis NA NA NA NA 3

hsa00300 Lysine biosynthesis NA NA NA NA 0

hsa00360 Phenylalanine metabolism NA NA NA NA 5

hsa00400 Phenylalanine, tyrosine and tryptophan biosynthesis NA NA NA NA 1

hsa00410 beta-Alanine metabolism NA NA NA NA 9

hsa00430 Taurine and hypotaurine metabolism NA NA NA NA 4

hsa00450 Selenocompound metabolism NA NA NA NA 9

hsa00460 Cyanoamino acid metabolism NA NA NA NA 2

hsa00471 D-Glutamine and D-glutamate metabolism NA NA NA NA 1

hsa00472 D-Arginine and D-ornithine metabolism NA NA NA NA 0

hsa00511 Other glycan degradation NA NA NA NA 4

hsa00533 Glycosaminoglycan biosynthesis - keratan sulfate NA NA NA NA 6

hsa00591 Linoleic acid metabolism NA NA NA NA 6

hsa00592 alpha-Linolenic acid metabolism NA NA NA NA 5

hsa00601 Glycosphingolipid biosynthesis - lacto and neolacto series NA NA NA NA 9

hsa00603 Glycosphingolipid biosynthesis - globo series NA NA NA NA 6

hsa00604 Glycosphingolipid biosynthesis - ganglio series NA NA NA NA 9

hsa00630 Glyoxylate and dicarboxylate metabolism NA NA NA NA 6

hsa00650 Butanoate metabolism NA NA NA NA 9

hsa00670 One carbon pool by folate NA NA NA NA 8

hsa00730 Thiamine metabolism NA NA NA NA 0

hsa00740 Riboflavin metabolism NA NA NA NA 5

hsa00750 Vitamin B6 metabolism NA NA NA NA 4

hsa00770 Pantothenate and CoA biosynthesis NA NA NA NA 6

hsa00780 Biotin metabolism NA NA NA NA 0

hsa00785 Lipoic acid metabolism NA NA NA NA 2

hsa00790 Folate biosynthesis NA NA NA NA 5

hsa00900 Terpenoid backbone biosynthesis NA NA NA NA 4

hsa00910 Nitrogen metabolism NA NA NA NA 7

hsa00920 Sulfur metabolism NA NA NA NA 5

hsa01040 Biosynthesis of unsaturated fatty acids NA NA NA NA 8

hsa03020 RNA polymerase NA NA NA NA 9

hsa03450 Non-homologous end-joining NA NA NA NA 5

hsa04122 Sulfur relay system NA NA NA NA 3

hsa04140 Regulation of autophagy NA NA NA NA 9

hsa04340 Hedgehog signaling pathway NA NA NA NA 9

hsa04614 Renin-angiotensin system NA NA NA NA 5

hsa04710 Circadian rhythm - mammal NA NA NA NA 7

hsa04744 Phototransduction NA NA NA NA 9

hsa04964 Proximal tubule bicarbonate reclamation NA NA NA NA 6

hsa04977 Vitamin digestion and absorption NA NA NA NA 8

hsa03013 RNA transport 0.00224852 -2.356686231 4.45E-58 5.69E-56 78

hsa04612 Antigen processing and presentation 0.007962322 -2.239990543 2.80E-52 1.79E-50 40

hsa03008 Ribosome biogenesis in eukaryotes 0.010169063 -1.913281458 9.27E-39 3.95E-37 48

hsa04672 Intestinal immune network for IgA production 0.019476949 -1.866945746 8.33E-37 2.67E-35 31

hsa03040 Spliceosome 0.023118923 -1.611086829 1.93E-28 4.93E-27 58

hsa03010 Ribosome 0.014619546 -1.58007293 3.80E-26 8.11E-25 52

hsa03018 RNA degradation 0.06964811 -1.260348082 3.16E-18 5.78E-17 35

hsa04650 Natural killer cell mediated cytotoxicity 0.082410013 -1.116430068 7.03E-15 1.12E-13 49

hsa04660 T cell receptor signaling pathway 0.084384899 -1.108616638 8.87E-15 1.26E-13 56

hsa04514 Cell adhesion molecules (CAMs) 0.09348904 -1.052692299 1.62E-13 2.07E-12 59

hsa00970 Aminoacyl-tRNA biosynthesis 0.133072857 -0.989717566 2.16E-11 2.52E-10 13

hsa03030 DNA replication 0.109155318 -0.976498889 2.36E-11 2.52E-10 18

hsa03420 Nucleotide excision repair 0.115046907 -0.963139829 2.97E-11 2.93E-10 22

hsa00280 Valine, leucine and isoleucine degradation 0.128816843 -0.95942875 3.21E-11 2.93E-10 20

hsa00510 N-Glycan biosynthesis 0.152023099 -0.926550133 1.07E-10 9.17E-10 20

hsa03015 mRNA surveillance pathway 0.165892281 -0.875760598 6.44E-10 5.15E-09 40

hsa03430 Mismatch repair 0.175299738 -0.802861444 3.09E-08 2.32E-07 12

hsa03022 Basal transcription factors 0.188589289 -0.787481501 5.30E-08 3.77E-07 13

hsa03410 Base excision repair 0.173489164 -0.782178347 5.62E-08 3.78E-07 15

hsa00563 Glycosylphosphatidylinositol(GPI)-anchor biosynthesis 0.19078658 -0.765756219 1.18E-07 7.55E-07 12

hsa03060 Protein export 0.232311418 -0.547418273 0.000124233 0.000757228 12

hsa00230 Purine metabolism 0.270034866 -0.501315713 0.000236836 0.001377954 71

hsa04640 Hematopoietic cell lineage 0.254676694 -0.468457781 0.000562983 0.003133122 51

hsa04070 Phosphatidylinositol signaling system 0.291470638 -0.456833384 0.000786356 0.004193896 26

hsa00270 Cysteine and methionine metabolism 0.321081727 -0.366283307 0.005709227 0.028218335 18

hsa00020 Citrate cycle (TCA cycle) 0.298123638 -0.373598925 0.005731849 0.028218335 10

hsa00620 Pyruvate metabolism 0.295943726 -0.363893801 0.006357406 0.030138814 16

hsa04350 TGF-beta signaling pathway 0.330278115 -0.347386422 0.00785719 0.035918582 38

hsa04120 Ubiquitin mediated proteolysis 0.268074308 -0.33461628 0.009976883 0.044035897 60

hsa02010 ABC transporters 0.328365142 -0.33534975 0.01041937 0.044455978 19

hsa00640 Propanoate metabolism 0.335237773 -0.32318709 0.0130701 0.053966863 15

hsa00240 Pyrimidine metabolism 0.326085856 -0.304257045 0.016994037 0.066032429 45

hsa00350 Tyrosine metabolism 0.360644035 -0.30800538 0.017023986 0.066032429 14

hsa00340 Histidine metabolism 0.37328606 -0.232828907 0.056215592 0.211635171 11

hsa00562 Inositol phosphate metabolism 0.387769636 -0.209184316 0.074154412 0.271193278 20

hsa04970 Salivary secretion 0.365046672 -0.202513164 0.080585434 0.286525988 30

hsa04662 B cell receptor signaling pathway 0.316557592 -0.07710529 0.296692626 1 39

hsa04310 Wnt signaling pathway 0.386978715 -0.072120803 0.307956275 1 57

hsa04974 Protein digestion and absorption 0.398761984 -0.00438383 0.487924859 1 19

hsa00071 Fatty acid metabolism 0.428667747 0.022840238 0.562964327 1 20

hsa04972 Pancreatic secretion 0.449222421 0.032656772 0.590818132 1 33

hsa04720 Long-term potentiation 0.469431029 0.039588775 0.608262812 1 24

hsa00532 Glycosaminoglycan biosynthesis - chondroitin sulfate 0.486384279 0.066691854 0.677931754 1 12

hsa00534 Glycosaminoglycan biosynthesis - heparan sulfate 0.512465415 0.093139652 0.738656715 1 10

hsa04146 Peroxisome 0.457986174 0.119390199 0.795885089 1 38

hsa04973 Carbohydrate digestion and absorption 0.430683001 0.146179308 0.841522794 1 21

hsa04960 Aldosterone-regulated sodium reabsorption 0.497436247 0.147898997 0.8437249 1 13

hsa04062 Chemokine signaling pathway 0.380277652 0.15879599 0.866208077 1 89

hsa03440 Homologous recombination 0.513956383 0.189320952 0.901670492 1 14

hsa04330 Notch signaling pathway 0.494678063 0.192659157 0.908092558 1 20

hsa04623 Cytosolic DNA-sensing pathway 0.529832268 0.24402704 0.95284162 1 15

hsa00590 Arachidonic acid metabolism 0.528832108 0.263932337 0.965941499 1 20

hsa04730 Long-term depression 0.550441643 0.306091725 0.982565476 1 21

hsa04742 Taste transduction 0.495743941 0.318718039 0.985597709 1 16

hsa04150 mTOR signaling pathway 0.570644265 0.320797255 0.98678386 1 23

hsa00561 Glycerolipid metabolism 0.576153301 0.332713947 0.989295854 1 19

hsa00040 Pentose and glucuronate interconversions 0.513728483 0.345056996 0.990709584 1 15

hsa04962 Vasopressin-regulated water reabsorption 0.584312457 0.341723509 0.990913817 1 19

hsa00565 Ether lipid metabolism 0.580006712 0.361186826 0.993069872 1 11

hsa04916 Melanogenesis 0.556507152 0.362558429 0.99399211 1 33

hsa04520 Adherens junction 0.559181787 0.415702695 0.998034853 1 31

hsa00053 Ascorbate and aldarate metabolism 0.513443249 0.446737097 0.998563278 1 12

hsa04110 Cell cycle 0.435995418 0.438409807 0.998709066 1 67

hsa04971 Gastric acid secretion 0.58475622 0.439202628 0.998806537 1 26

hsa00380 Tryptophan metabolism 0.635066091 0.448949223 0.998961863 1 13

hsa00310 Lysine degradation 0.639111986 0.463259641 0.999259712 1 13

hsa04012 ErbB signaling pathway 0.592629829 0.460884513 0.99930567 1 33

hsa00514 Other types of O-glycan biosynthesis 0.540895545 0.4881476 0.999581933 1 18

hsa04530 Tight junction 0.633836273 0.526600436 0.999875096 1 44

hsa03050 Proteasome 0.591292167 0.543064357 0.999875665 1 13

hsa00830 Retinol metabolism 0.468670082 0.563733787 0.999941677 1 26

hsa04210 Apoptosis 0.601301566 0.555656094 0.999943594 1 46

hsa04664 Fc epsilon RI signaling pathway 0.604192424 0.560978645 0.999948613 1 31

hsa04975 Fat digestion and absorption 0.652024693 0.578036426 0.99995668 1 10

hsa04260 Cardiac muscle contraction 0.637269495 0.590618725 0.999976426 1 24

hsa04360 Axon guidance 0.586093277 0.591536051 0.999980033 1 52

hsa00520 Amino sugar and nucleotide sugar metabolism 0.646334324 0.622511145 0.999991412 1 29

hsa04630 Jak-STAT signaling pathway 0.616445633 0.648508842 0.999996753 1 59

hsa04976 Bile secretion 0.635464081 0.683424679 0.999998703 1 24

hsa04622 RIG-I-like receptor signaling pathway 0.693807151 0.683065578 0.999998707 1 18

hsa04370 VEGF signaling pathway 0.656028461 0.695719348 0.999999216 1 29

hsa04115 p53 signaling pathway 0.688345993 0.695100274 0.999999276 1 35

hsa04141 Protein processing in endoplasmic reticulum 0.653812985 0.715726473 0.999999677 1 60

hsa00330 Arginine and proline metabolism 0.707152267 0.729924788 0.999999761 1 23

hsa04966 Collecting duct acid secretion 0.697630823 0.740427687 0.999999767 1 12

hsa00760 Nicotinate and nicotinamide metabolism 0.720020429 0.753077767 0.999999832 1 10

hsa00500 Starch and sucrose metabolism 0.548600828 0.76812641 0.999999916 1 29

hsa04722 Neurotrophin signaling pathway 0.663390448 0.795151256 0.999999983 1 49

hsa00512 Mucin type O-Glycan biosynthesis 0.727336763 0.810585824 0.999999985 1 14

hsa04914 Progesterone-mediated oocyte maturation 0.696081054 0.804150142 0.999999987 1 42

hsa04912 GnRH signaling pathway 0.686409867 0.808486958 0.999999989 1 39

hsa04270 Vascular smooth muscle contraction 0.71107199 0.8137631 0.999999992 1 41

hsa04130 SNARE interactions in vesicular transport 0.686226057 0.883380558 0.999999999 1 16

hsa04114 Oocyte meiosis 0.662166189 0.888103923 1 1 50

hsa04540 Gap junction 0.68400657 0.894186191 1 1 36

hsa04320 Dorso-ventral axis formation 0.750619375 0.935747774 1 1 11

hsa00030 Pentose phosphate pathway 0.753237135 0.952746064 1 1 17

hsa00600 Sphingolipid metabolism 0.734002849 0.958750688 1 1 17

hsa00980 Metabolism of xenobiotics by cytochrome P450 0.675621088 0.973182501 1 1 23

hsa00860 Porphyrin and chlorophyll metabolism 0.728729034 0.982959373 1 1 19

hsa00140 Steroid hormone biosynthesis 0.641876744 0.999920789 1 1 18

hsa00564 Glycerophospholipid metabolism 0.70791611 0.976771438 1 1 31

hsa04144 Endocytosis 0.6613336 0.971773613 1 1 81

hsa04670 Leukocyte transendothelial migration 0.697138615 1.039236046 1 1 46

hsa04010 MAPK signaling pathway 0.707186444 1.04808766 1 1 98

hsa00982 Drug metabolism - cytochrome P450 0.610342455 1.087563285 1 1 23

hsa04020 Calcium signaling pathway 0.695878714 1.071985624 1 1 64

hsa00190 Oxidative phosphorylation 0.737152037 1.089125289 1 1 44

hsa04920 Adipocytokine signaling pathway 0.796193616 1.089798802 1 1 33

hsa00480 Glutathione metabolism 0.786642536 1.110178902 1 1 24

hsa00051 Fructose and mannose metabolism 0.823922855 1.172167665 1 1 17

hsa00531 Glycosaminoglycan degradation 0.803065014 1.257329942 1 1 10

hsa00010 Glycolysis / Gluconeogenesis 0.849968102 1.438619389 1 1 31

hsa00052 Galactose metabolism 0.829285731 1.272071517 1 1 16

hsa00983 Drug metabolism - other enzymes 0.749096419 1.452537004 1 1 23

hsa03320 PPAR signaling pathway 0.886542926 1.597233745 1 1 24

hsa04142 Lysosome 0.818320689 2.027149078 1 1 59

hsa04145 Phagosome 0.668260621 1.302658365 1 1 73

hsa04380 Osteoclast differentiation 0.733913776 1.948024403 1 1 65

hsa04510 Focal adhesion 0.74311781 1.478517677 1 1 76

hsa04512 ECM-receptor interaction 0.69877176 1.290258349 1 1 33

hsa04610 Complement and coagulation cascades 0.899216217 2.627115289 1 1 27

hsa04620 Toll-like receptor signaling pathway 0.75078068 1.404108552 1 1 35

hsa04621 NOD-like receptor signaling pathway 0.790571784 1.507315834 1 1 25

hsa04666 Fc gamma R-mediated phagocytosis 0.781807072 1.692698376 1 1 45

hsa04740 Olfactory transduction 0.577770189 1.286221584 1 1 29

hsa04810 Regulation of actin cytoskeleton 0.811449739 1.886943364 1 1 77

hsa04910 Insulin signaling pathway 0.783235592 1.37837677 1 1 61

hsa00061 Fatty acid biosynthesis NA NA NA NA 2

hsa00072 Synthesis and degradation of ketone bodies NA NA NA NA 3

hsa00100 Steroid biosynthesis NA NA NA NA 6

hsa00120 Primary bile acid biosynthesis NA NA NA NA 3

hsa00130 Ubiquinone and other terpenoid-quinone biosynthesis NA NA NA NA 4

hsa00232 Caffeine metabolism NA NA NA NA 1

hsa00250 Alanine, aspartate and glutamate metabolism NA NA NA NA 9

hsa00260 Glycine, serine and threonine metabolism NA NA NA NA 7

hsa00290 Valine, leucine and isoleucine biosynthesis NA NA NA NA 3

hsa00300 Lysine biosynthesis NA NA NA NA 0

hsa00360 Phenylalanine metabolism NA NA NA NA 5

hsa00400 Phenylalanine, tyrosine and tryptophan biosynthesis NA NA NA NA 1

hsa00410 beta-Alanine metabolism NA NA NA NA 9

hsa00430 Taurine and hypotaurine metabolism NA NA NA NA 4

hsa00450 Selenocompound metabolism NA NA NA NA 9

hsa00460 Cyanoamino acid metabolism NA NA NA NA 2

hsa00471 D-Glutamine and D-glutamate metabolism NA NA NA NA 1

hsa00472 D-Arginine and D-ornithine metabolism NA NA NA NA 0

hsa00511 Other glycan degradation NA NA NA NA 4

hsa00533 Glycosaminoglycan biosynthesis - keratan sulfate NA NA NA NA 6

hsa00591 Linoleic acid metabolism NA NA NA NA 6

hsa00592 alpha-Linolenic acid metabolism NA NA NA NA 5

hsa00601 Glycosphingolipid biosynthesis - lacto and neolacto series NA NA NA NA 9

hsa00603 Glycosphingolipid biosynthesis - globo series NA NA NA NA 6

hsa00604 Glycosphingolipid biosynthesis - ganglio series NA NA NA NA 9

hsa00630 Glyoxylate and dicarboxylate metabolism NA NA NA NA 6

hsa00650 Butanoate metabolism NA NA NA NA 9

hsa00670 One carbon pool by folate NA NA NA NA 8

hsa00730 Thiamine metabolism NA NA NA NA 0

hsa00740 Riboflavin metabolism NA NA NA NA 5

hsa00750 Vitamin B6 metabolism NA NA NA NA 4

hsa00770 Pantothenate and CoA biosynthesis NA NA NA NA 6

hsa00780 Biotin metabolism NA NA NA NA 0

hsa00785 Lipoic acid metabolism NA NA NA NA 2

hsa00790 Folate biosynthesis NA NA NA NA 5

hsa00900 Terpenoid backbone biosynthesis NA NA NA NA 4

hsa00910 Nitrogen metabolism NA NA NA NA 7

hsa00920 Sulfur metabolism NA NA NA NA 5

hsa01040 Biosynthesis of unsaturated fatty acids NA NA NA NA 8

hsa03020 RNA polymerase NA NA NA NA 9

hsa03450 Non-homologous end-joining NA NA NA NA 5

hsa04122 Sulfur relay system NA NA NA NA 3

hsa04140 Regulation of autophagy NA NA NA NA 9

hsa04340 Hedgehog signaling pathway NA NA NA NA 9

hsa04614 Renin-angiotensin system NA NA NA NA 5

hsa04710 Circadian rhythm - mammal NA NA NA NA 7

hsa04744 Phototransduction NA NA NA NA 9

hsa04964 Proximal tubule bicarbonate reclamation NA NA NA NA 6

hsa04977 Vitamin digestion and absorption NA NA NA NA 8

GSE72829 p.geomean stat.mean p.val q.val set.size

hsa04670 Leukocyte transendothelial migration 0.05415705 1.300194212 2.01E-20 2.73E-18 51

hsa04810 Regulation of actin cytoskeleton 0.048998803 1.27237948 5.85E-20 3.98E-18 91

hsa04142 Lysosome 0.084005127 1.204835267 3.40E-18 1.54E-16 74

hsa04666 Fc gamma R-mediated phagocytosis 0.061311067 1.166895344 6.26E-17 2.13E-15 51

hsa04380 Osteoclast differentiation 0.061344043 1.083941709 5.54E-15 1.51E-13 76

hsa00030 Pentose phosphate pathway 0.128313338 0.990613986 2.73E-12 6.18E-11 16

hsa04664 Fc epsilon RI signaling pathway 0.135860182 0.913638652 4.95E-11 9.62E-10 32

hsa04145 Phagosome 0.133686986 0.758400951 2.76E-08 4.69E-07 82

hsa00010 Glycolysis / Gluconeogenesis 0.20366167 0.739123308 6.59E-08 9.96E-07 35

hsa00500 Starch and sucrose metabolism 0.230093425 0.705126812 2.63E-07 3.58E-06 20

hsa04966 Collecting duct acid secretion 0.213661309 0.704455122 3.65E-07 4.51E-06 13

hsa04650 Natural killer cell mediated cytotoxicity 0.168056547 0.681314767 5.55E-07 6.29E-06 65

hsa04976 Bile secretion 0.228454012 0.676990106 7.91E-07 8.27E-06 19

hsa00190 Oxidative phosphorylation 0.154362526 0.668569919 9.97E-07 9.69E-06 57

hsa00590 Arachidonic acid metabolism 0.24434838 0.654543094 1.53E-06 1.39E-05 22

hsa04662 B cell receptor signaling pathway 0.154654558 0.651501074 1.75E-06 1.49E-05 48

hsa04510 Focal adhesion 0.188401339 0.641011783 2.16E-06 1.73E-05 77

hsa04722 Neurotrophin signaling pathway 0.21096455 0.636079059 2.56E-06 1.94E-05 62

hsa04520 Adherens junction 0.182161148 0.639337348 2.75E-06 1.97E-05 34

hsa00531 Glycosaminoglycan degradation 0.255402943 0.633352137 3.55E-06 2.42E-05 15

hsa04910 Insulin signaling pathway 0.253927411 0.602859056 7.45E-06 4.83E-05 64

hsa00860 Porphyrin and chlorophyll metabolism 0.232700362 0.62124119 7.93E-06 4.90E-05 12

hsa00520 Amino sugar and nucleotide sugar metabolism 0.269205274 0.598621439 9.98E-06 5.90E-05 24

hsa04620 Toll-like receptor signaling pathway 0.245260934 0.552207624 3.88E-05 0.000219832 46

hsa00480 Glutathione metabolism 0.273746242 0.547138149 5.02E-05 0.000273148 19

hsa04210 Apoptosis 0.253284213 0.523650063 8.82E-05 0.000461596 50

hsa04971 Gastric acid secretion 0.264769995 0.518522828 0.000109677 0.000552449 27

hsa04920 Adipocytokine signaling pathway 0.252463312 0.510432064 0.000131522 0.00063882 37

hsa04270 Vascular smooth muscle contraction 0.286367247 0.505650324 0.000148116 0.000694611 37

hsa04062 Chemokine signaling pathway 0.180036176 0.497523935 0.000179582 0.000814105 93

hsa00770 Pantothenate and CoA biosynthesis 0.29851147 0.489354291 0.000280391 0.001230102 11

hsa00051 Fructose and mannose metabolism 0.311024424 0.465713336 0.000453299 0.001926519 20

hsa04530 Tight junction 0.237591825 0.433256405 0.000986309 0.004064789 49

hsa04730 Long-term depression 0.285485043 0.427601115 0.001197421 0.004789683 19

hsa00052 Galactose metabolism 0.346320092 0.383642286 0.003168516 0.012311946 17

hsa04962 Vasopressin-regulated water reabsorption 0.318861612 0.374360219 0.003867763 0.01461155 20

hsa03320 PPAR signaling pathway 0.344239701 0.370554856 0.0040354 0.014832821 29

hsa04622 RIG-I-like receptor signaling pathway 0.334721092 0.367593145 0.004338377 0.015526822 24

hsa00564 Glycerophospholipid metabolism 0.353347659 0.349951851 0.006080495 0.021203777 40

hsa00980 Metabolism of xenobiotics by cytochrome P450 0.347086045 0.344640139 0.007433866 0.025275145 12

hsa04146 Peroxisome 0.357941218 0.336681878 0.007852141 0.026046126 43

hsa04010 MAPK signaling pathway 0.303773303 0.326024378 0.009486739 0.030718964 116

hsa04540 Gap junction 0.35262357 0.306802879 0.014069983 0.044500412 33

hsa00983 Drug metabolism - other enzymes 0.369557312 0.307722038 0.014480895 0.044759131 16

hsa00330 Arginine and proline metabolism 0.369542922 0.296236364 0.017017578 0.051430903 25

hsa04610 Complement and coagulation cascades 0.332372122 0.293370135 0.018308377 0.054085628 25

hsa00600 Sphingolipid metabolism 0.382277084 0.291911068 0.018691357 0.054085628 23

hsa04144 Endocytosis 0.338752752 0.28779002 0.019277297 0.054619009 83

hsa04260 Cardiac muscle contraction 0.313122423 0.286526928 0.021167507 0.058750631 24

hsa04115 p53 signaling pathway 0.37829683 0.264931243 0.028799736 0.078335282 35

hsa00760 Nicotinate and nicotinamide metabolism 0.380364897 0.252157541 0.037371028 0.099656075 11

hsa00512 Mucin type O-Glycan biosynthesis 0.398577935 0.246671028 0.040446026 0.105781914 11

hsa00982 Drug metabolism - cytochrome P450 0.400001444 0.206047237 0.073351914 0.188223778 10

hsa04370 VEGF signaling pathway 0.386378166 0.192329988 0.083921656 0.20857907 35

hsa04973 Carbohydrate digestion and absorption 0.418162638 0.193062855 0.08435183 0.20857907 18

hsa00511 Other glycan degradation 0.419016152 0.182453464 0.099354195 0.241288759 11

hsa04975 Fat digestion and absorption 0.438299 0.144159942 0.1534701 0.366174274 14

hsa04912 GnRH signaling pathway 0.451066488 0.105164164 0.225725679 0.5292878 34

hsa04360 Axon guidance 0.356112742 0.093695189 0.249742443 0.575677495 58

hsa04630 Jak-STAT signaling pathway 0.369759818 0.077546745 0.287398899 0.638999628 64

hsa04972 Pancreatic secretion 0.460986032 0.078237041 0.28776535 0.638999628 29

hsa04621 NOD-like receptor signaling pathway 0.417932398 0.076897348 0.291308654 0.638999628 33

hsa00071 Fatty acid metabolism 0.462888562 0.068292168 0.312546346 0.674703223 22

hsa02010 ABC transporters 0.472205352 0.053002746 0.353234104 0.750622471 13

hsa00562 Inositol phosphate metabolism 0.464587509 0.044131808 0.376372083 0.773007771 23

hsa04130 SNARE interactions in vesicular transport 0.46712152 0.044084385 0.376543896 0.773007771 22

hsa04974 Protein digestion and absorption 0.476241193 0.042573966 0.380820005 0.773007771 21

hsa04150 mTOR signaling pathway 0.461384546 0.02970994 0.415831959 0.831663918 25

hsa00532 Glycosaminoglycan biosynthesis - chondroitin sulfate 0.475815539 0.022498477 0.436063778 0.859488026 14

hsa04114 Oocyte meiosis 0.452100421 0.014563241 0.458823796 0.891429089 53

hsa04960 Aldosterone-regulated sodium reabsorption 0.4899527 0.006472858 0.481575601 0.915739302 13

hsa00640 Propanoate metabolism 0.470580566 0.001655696 0.495325236 0.915739302 18

hsa01040 Biosynthesis of unsaturated fatty acids 0.49627936 0.000527438 0.498599457 0.915739302 13

hsa00380 Tryptophan metabolism 0.489786822 -0.000565913 0.501391975 0.915739302 17

hsa00450 Selenocompound metabolism 0.497639791 -0.001773829 0.505003291 0.915739302 12

hsa00601 Glycosphingolipid biosynthesis - lacto and neolacto series 0.501935428 -0.010456592 0.529230933 0.938909975 10

hsa04623 Cytosolic DNA-sensing pathway 0.446327511 -0.011807675 0.53436575 0.938909975 22

hsa04310 Wnt signaling pathway 0.431892011 -0.013720991 0.538492486 0.938909975 70

hsa04320 Dorso-ventral axis formation 0.498647009 -0.023521742 0.566589646 0.965592759 11

hsa00565 Ether lipid metabolism 0.504964205 -0.026166305 0.573383612 0.965592759 15

hsa04914 Progesterone-mediated oocyte maturation 0.498039317 -0.028208371 0.580086656 0.965592759 42

hsa04640 Hematopoietic cell lineage 0.457126988 -0.029081356 0.582195634 0.965592759 56

hsa00604 Glycosphingolipid biosynthesis - ganglio series 0.515383567 -0.055600076 0.651622018 1 11

hsa04110 Cell cycle 0.469470205 -0.067187099 0.68558538 1 70

hsa00310 Lysine degradation 0.518944513 -0.071687324 0.695507903 1 23

hsa00240 Pyrimidine metabolism 0.511212106 -0.077453642 0.710719124 1 50

hsa04710 Circadian rhythm - mammal 0.517044399 -0.082909105 0.719708411 1 13

hsa04512 ECM-receptor interaction 0.520742941 -0.081874779 0.720578404 1 27

hsa00260 Glycine, serine and threonine metabolism 0.521665791 -0.08724246 0.73171996 1 16

hsa00561 Glycerolipid metabolism 0.529267397 -0.091483179 0.743411647 1 27

hsa00830 Retinol metabolism 0.531488923 -0.095501208 0.74959857 1 13

hsa04012 ErbB signaling pathway 0.531334814 -0.108659929 0.781558867 1 36

hsa04916 Melanogenesis 0.536788558 -0.116275365 0.79776179 1 43

hsa04720 Long-term potentiation 0.539812588 -0.122464462 0.808898973 1 25

hsa03022 Basal transcription factors 0.546697669 -0.133308704 0.826859015 1 14

hsa04740 Olfactory transduction 0.546439763 -0.134106761 0.831112177 1 31

hsa00534 Glycosaminoglycan biosynthesis - heparan sulfate 0.553232201 -0.143383745 0.844185555 1 13

hsa03050 Proteasome 0.375344809 -0.141609659 0.850481622 1 23

hsa00270 Cysteine and methionine metabolism 0.513154296 -0.151542624 0.859221979 1 18

hsa04070 Phosphatidylinositol signaling system 0.541313547 -0.1573775 0.869816489 1 32

hsa04742 Taste transduction 0.566138028 -0.179815313 0.897661298 1 13

hsa00563 Glycosylphosphatidylinositol(GPI)-anchor biosynthesis 0.565878474 -0.181485207 0.898275033 1 10

hsa00350 Tyrosine metabolism 0.556096966 -0.188749709 0.90935875 1 17

hsa03060 Protein export 0.516360908 -0.198999719 0.920339465 1 14

hsa00650 Butanoate metabolism 0.566745718 -0.207254297 0.927911669 1 13

hsa04141 Protein processing in endoplasmic reticulum 0.531407778 -0.206011985 0.930676734 1 75

hsa04020 Calcium signaling pathway 0.570299867 -0.215597936 0.939172555 1 58

hsa04970 Salivary secretion 0.526780031 -0.225270502 0.946310651 1 23

hsa04340 Hedgehog signaling pathway 0.583940654 -0.228833175 0.947791429 1 17

hsa00280 Valine, leucine and isoleucine degradation 0.579311776 -0.235482498 0.953612291 1 28

hsa00340 Histidine metabolism 0.582973922 -0.243246524 0.95705023 1 13

hsa03020 RNA polymerase 0.580728054 -0.248585184 0.959928142 1 13

hsa03420 Nucleotide excision repair 0.554329955 -0.252428913 0.96352221 1 24

hsa00620 Pyruvate metabolism 0.544684903 -0.263946904 0.970046305 1 22

hsa04120 Ubiquitin mediated proteolysis 0.552684596 -0.272689134 0.974702814 1 62

hsa03410 Base excision repair 0.575019955 -0.275968942 0.974751055 1 19

hsa00020 Citrate cycle (TCA cycle) 0.563113544 -0.283547069 0.977726892 1 16

hsa04330 Notch signaling pathway 0.598547966 -0.285039521 0.978917046 1 23

hsa03440 Homologous recombination 0.609275818 -0.308974306 0.984705069 1 10

hsa00510 N-Glycan biosynthesis 0.604102455 -0.304665233 0.984745201 1 20

hsa03430 Mismatch repair 0.610380943 -0.314208095 0.986416668 1 12

hsa03030 DNA replication 0.595469255 -0.343968193 0.992555544 1 19

hsa03008 Ribosome biogenesis in eukaryotes 0.593444661 -0.388030206 0.99722464 1 38

hsa04350 TGF-beta signaling pathway 0.619756171 -0.392880015 0.997558426 1 41

hsa00970 Aminoacyl-tRNA biosynthesis 0.622543257 -0.422848601 0.998675768 1 24

hsa00230 Purine metabolism 0.632716287 -0.450409799 0.999391342 1 74

hsa00514 Other types of O-glycan biosynthesis 0.665157296 -0.472557863 0.99958388 1 15

hsa03015 mRNA surveillance pathway 0.675444585 -0.545651016 0.999951037 1 33

hsa04660 T cell receptor signaling pathway 0.66128095 -0.605689364 0.999993102 1 68

hsa03018 RNA degradation 0.67967229 -0.646287141 0.999998053 1 35

hsa04514 Cell adhesion molecules (CAMs) 0.639242597 -0.707047442 0.999999778 1 57

hsa03013 RNA transport 0.679393562 -1.044284155 1 1 69

hsa03040 Spliceosome 0.748589683 -1.058781283 1 1 76

hsa03010 Ribosome 0.320420663 -4.234576248 1 1 50

hsa04612 Antigen processing and presentation 0.743070708 -1.247614409 1 1 48

hsa04672 Intestinal immune network for IgA production 0.83971879 -1.555921998 1 1 29

hsa00040 Pentose and glucuronate interconversions NA NA NA NA 8

hsa00053 Ascorbate and aldarate metabolism NA NA NA NA 3

hsa00061 Fatty acid biosynthesis NA NA NA NA 4

hsa00072 Synthesis and degradation of ketone bodies NA NA NA NA 4

hsa00100 Steroid biosynthesis NA NA NA NA 6

hsa00120 Primary bile acid biosynthesis NA NA NA NA 5

hsa00130 Ubiquinone and other terpenoid-quinone biosynthesis NA NA NA NA 4

hsa00140 Steroid hormone biosynthesis NA NA NA NA 7

hsa00232 Caffeine metabolism NA NA NA NA 1

hsa00250 Alanine, aspartate and glutamate metabolism NA NA NA NA 8

hsa00290 Valine, leucine and isoleucine biosynthesis NA NA NA NA 8

hsa00300 Lysine biosynthesis NA NA NA NA 0

hsa00360 Phenylalanine metabolism NA NA NA NA 6

hsa00400 Phenylalanine, tyrosine and tryptophan biosynthesis NA NA NA NA 2

hsa00410 beta-Alanine metabolism NA NA NA NA 9

hsa00430 Taurine and hypotaurine metabolism NA NA NA NA 3

hsa00460 Cyanoamino acid metabolism NA NA NA NA 3

hsa00471 D-Glutamine and D-glutamate metabolism NA NA NA NA 1

hsa00472 D-Arginine and D-ornithine metabolism NA NA NA NA 1

hsa00533 Glycosaminoglycan biosynthesis - keratan sulfate NA NA NA NA 6

hsa00591 Linoleic acid metabolism NA NA NA NA 5

hsa00592 alpha-Linolenic acid metabolism NA NA NA NA 3

hsa00603 Glycosphingolipid biosynthesis - globo series NA NA NA NA 9

hsa00630 Glyoxylate and dicarboxylate metabolism NA NA NA NA 6

hsa00670 One carbon pool by folate NA NA NA NA 8

hsa00730 Thiamine metabolism NA NA NA NA 2

hsa00740 Riboflavin metabolism NA NA NA NA 5

hsa00750 Vitamin B6 metabolism NA NA NA NA 4

hsa00780 Biotin metabolism NA NA NA NA 1

hsa00785 Lipoic acid metabolism NA NA NA NA 2

hsa00790 Folate biosynthesis NA NA NA NA 4

hsa00900 Terpenoid backbone biosynthesis NA NA NA NA 6

hsa00910 Nitrogen metabolism NA NA NA NA 7

hsa00920 Sulfur metabolism NA NA NA NA 8

hsa03450 Non-homologous end-joining NA NA NA NA 5

hsa04122 Sulfur relay system NA NA NA NA 5

hsa04140 Regulation of autophagy NA NA NA NA 8

hsa04614 Renin-angiotensin system NA NA NA NA 7

hsa04744 Phototransduction NA NA NA NA 7

hsa04964 Proximal tubule bicarbonate reclamation NA NA NA NA 4

hsa04977 Vitamin digestion and absorption NA NA NA NA 6

hsa03010 Ribosome 9.17E-07 -4.234576248 5.32E-154 7.24E-152 50

hsa04672 Intestinal immune network for IgA production 0.04881873 -1.555921998 3.72E-28 2.53E-26 29

hsa04612 Antigen processing and presentation 0.073141503 -1.247614409 4.33E-19 1.96E-17 48

hsa03040 Spliceosome 0.118116448 -1.058781283 1.87E-14 6.36E-13 76

hsa03013 RNA transport 0.104722919 -1.044284155 4.33E-14 1.18E-12 69

hsa04514 Cell adhesion molecules (CAMs) 0.181617266 -0.707047442 2.22E-07 5.03E-06 57

hsa03018 RNA degradation 0.232682495 -0.646287141 1.95E-06 3.78E-05 35

hsa04660 T cell receptor signaling pathway 0.239198359 -0.605689364 6.90E-06 0.000117272 68

hsa03015 mRNA surveillance pathway 0.278293382 -0.545651016 4.90E-05 0.000739888 33

hsa00514 Other types of O-glycan biosynthesis 0.313532887 -0.472557863 0.00041612 0.005659234 15

hsa00230 Purine metabolism 0.303521584 -0.450409799 0.000608658 0.007525229 74

hsa00970 Aminoacyl-tRNA biosynthesis 0.315071711 -0.422848601 0.001324232 0.015007958 24

hsa04350 TGF-beta signaling pathway 0.327643526 -0.392880015 0.002441574 0.025542624 41

hsa03008 Ribosome biogenesis in eukaryotes 0.315995839 -0.388030206 0.00277536 0.026960636 38

hsa03030 DNA replication 0.343720855 -0.343968193 0.007444456 0.067496399 19

hsa03430 Mismatch repair 0.372674369 -0.314208095 0.013583332 0.115458325 12

hsa00510 N-Glycan biosynthesis 0.372324425 -0.304665233 0.015254799 0.115561699 20

hsa03440 Homologous recombination 0.376001956 -0.308974306 0.015294931 0.115561699 10

hsa04330 Notch signaling pathway 0.379533174 -0.285039521 0.021082954 0.150909564 23

hsa00020 Citrate cycle (TCA cycle) 0.358399472 -0.283547069 0.022273108 0.151457134 16

hsa03410 Base excision repair 0.370392003 -0.275968942 0.025248945 0.156382607 19

hsa04120 Ubiquitin mediated proteolysis 0.355060165 -0.272689134 0.025297186 0.156382607 62

hsa00620 Pyruvate metabolism 0.35635919 -0.263946904 0.029953695 0.177117501 22

hsa03420 Nucleotide excision repair 0.36979153 -0.252428913 0.03647779 0.206707474 24

hsa03020 RNA polymerase 0.392774662 -0.248585184 0.040071858 0.217990907 13

hsa00340 Histidine metabolism 0.397288327 -0.243246524 0.04294977 0.224660335 13

hsa00280 Valine, leucine and isoleucine degradation 0.398078308 -0.235482498 0.046387709 0.233656608 28

hsa04340 Hedgehog signaling pathway 0.406776113 -0.228833175 0.052208571 0.251784533 17

hsa04970 Salivary secretion 0.366581271 -0.225270502 0.053689349 0.251784533 23

hsa04020 Calcium signaling pathway 0.403703432 -0.215597936 0.060827445 0.275751086 58

hsa04141 Protein processing in endoplasmic reticulum 0.381702908 -0.206011985 0.069323266 0.304127877 75

hsa00650 Butanoate metabolism 0.409377434 -0.207254297 0.072088331 0.306375407 13

hsa03060 Protein export 0.377675947 -0.198999719 0.079660535 0.328297964 14

hsa00350 Tyrosine metabolism 0.412770751 -0.188749709 0.09064125 0.362565001 17

hsa00563 Glycosylphosphatidylinositol(GPI)-anchor biosynthesis 0.426613556 -0.181485207 0.101724967 0.386612873 10

hsa04742 Taste transduction 0.427173797 -0.179815313 0.102338702 0.386612873 13

hsa04070 Phosphatidylinositol signaling system 0.420503382 -0.1573775 0.130183511 0.478512363 32

hsa00270 Cysteine and methionine metabolism 0.403625067 -0.151542624 0.140778021 0.503837127 18

hsa03050 Proteasome 0.30919273 -0.141609659 0.149518378 0.521397421 23

hsa00534 Glycosaminoglycan biosynthesis - heparan sulfate 0.44199671 -0.143383745 0.155814445 0.529769115 13

hsa04740 Olfactory transduction 0.441576533 -0.134106761 0.168887823 0.560213266 31

hsa03022 Basal transcription factors 0.443549634 -0.133308704 0.173140985 0.560646998 14

hsa04720 Long-term potentiation 0.444469694 -0.122464462 0.191101027 0.604412552 25

hsa04916 Melanogenesis 0.445697524 -0.116275365 0.20223821 0.625099923 43

hsa04012 ErbB signaling pathway 0.446963275 -0.108659929 0.218441133 0.660177645 36

hsa00830 Retinol metabolism 0.457717745 -0.095501208 0.25040143 0.740317271 13

hsa00561 Glycerolipid metabolism 0.4576984 -0.091483179 0.256588353 0.742468426 27

hsa00260 Glycine, serine and threonine metabolism 0.455112992 -0.08724246 0.26828004 0.760126779 16

hsa04512 ECM-receptor interaction 0.457427865 -0.081874779 0.279421596 0.762393123 27

hsa04710 Circadian rhythm - mammal 0.45427317 -0.082909105 0.280291589 0.762393123 13

hsa00240 Pyrimidine metabolism 0.451698641 -0.077453642 0.289280876 0.77141567 50

hsa00310 Lysine degradation 0.463147847 -0.071687324 0.304492097 0.796363946 23

hsa04110 Cell cycle 0.424928972 -0.067187099 0.31441462 0.80679978 70

hsa00604 Glycosphingolipid biosynthesis - ganglio series 0.472727794 -0.055600076 0.348377982 0.8773964 11

hsa04640 Hematopoietic cell lineage 0.432301457 -0.029081356 0.417804366 1 56

hsa04914 Progesterone-mediated oocyte maturation 0.476273347 -0.028208371 0.419913344 1 42

hsa00565 Ether lipid metabolism 0.484641547 -0.026166305 0.426616388 1 15

hsa04320 Dorso-ventral axis formation 0.480447046 -0.023521742 0.433410354 1 11

hsa04310 Wnt signaling pathway 0.419394903 -0.013720991 0.461507514 1 70

hsa04623 Cytosolic DNA-sensing pathway 0.438963293 -0.011807675 0.46563425 1 22

hsa00601 Glycosphingolipid biosynthesis - lacto and neolacto series 0.493861413 -0.010456592 0.470769067 1 10

hsa00450 Selenocompound metabolism 0.49627572 -0.001773829 0.494996709 1 12

hsa00380 Tryptophan metabolism 0.489696095 -0.000565913 0.498608025 1 17

hsa01040 Biosynthesis of unsaturated fatty acids 0.496729691 0.000527438 0.501400543 1 13

hsa00640 Propanoate metabolism 0.473172547 0.001655696 0.504674764 1 18

hsa04960 Aldosterone-regulated sodium reabsorption 0.495021092 0.006472858 0.518424399 1 13

hsa04114 Oocyte meiosis 0.466606892 0.014563241 0.541176204 1 53

hsa00532 Glycosaminoglycan biosynthesis - chondroitin sulfate 0.49316074 0.022498477 0.563936222 1 14

hsa04150 mTOR signaling pathway 0.483898981 0.02970994 0.584168041 1 25

hsa04974 Protein digestion and absorption 0.509542926 0.042573966 0.619179995 1 21

hsa04130 SNARE interactions in vesicular transport 0.501382719 0.044084385 0.623456104 1 22

hsa00562 Inositol phosphate metabolism 0.498536373 0.044131808 0.623627917 1 23

hsa02010 ABC transporters 0.513567334 0.053002746 0.646765896 1 13

hsa00071 Fatty acid metabolism 0.516120141 0.068292168 0.687453654 1 22

hsa04621 NOD-like receptor signaling pathway 0.474619639 0.076897348 0.708691346 1 33

hsa04972 Pancreatic secretion 0.522192854 0.078237041 0.71223465 1 29

hsa04630 Jak-STAT signaling pathway 0.41328336 0.077546745 0.712601101 1 64

hsa04360 Axon guidance 0.413139664 0.093695189 0.750257557 1 58

hsa04912 GnRH signaling pathway 0.533390471 0.105164164 0.774274321 1 34

hsa04975 Fat digestion and absorption 0.550249731 0.144159942 0.8465299 1 14

hsa00511 Other glycan degradation 0.55853026 0.182453464 0.900645805 1 11

hsa04973 Carbohydrate digestion and absorption 0.568165781 0.193062855 0.91564817 1 18

hsa04370 VEGF signaling pathway 0.527348675 0.192329988 0.916078344 1 35

hsa00982 Drug metabolism - cytochrome P450 0.55530716 0.206047237 0.926648086 1 10

hsa00512 Mucin type O-Glycan biosynthesis 0.588221709 0.246671028 0.959553974 1 11

hsa00760 Nicotinate and nicotinamide metabolism 0.567379195 0.252157541 0.962628972 1 11

hsa04115 p53 signaling pathway 0.58016617 0.264931243 0.971200264 1 35

hsa04260 Cardiac muscle contraction 0.508816415 0.286526928 0.978832493 1 24

hsa04144 Endocytosis 0.543355757 0.28779002 0.980722703 1 83

hsa00600 Sphingolipid metabolism 0.608181201 0.291911068 0.981308643 1 23

hsa04610 Complement and coagulation cascades 0.541208764 0.293370135 0.981691623 1 25

hsa00330 Arginine and proline metabolism 0.595473579 0.296236364 0.982982422 1 25

hsa00983 Drug metabolism - other enzymes 0.603809476 0.307722038 0.985519105 1 16

hsa04540 Gap junction 0.579858035 0.306802879 0.985930017 1 33

hsa04010 MAPK signaling pathway 0.520784553 0.326024378 0.990513261 1 116

hsa04146 Peroxisome 0.615680193 0.336681878 0.992147859 1 43

hsa00980 Metabolism of xenobiotics by cytochrome P450 0.603305618 0.344640139 0.992566134 1 12

hsa00564 Glycerophospholipid metabolism 0.620620284 0.349951851 0.993919505 1 40

hsa04622 RIG-I-like receptor signaling pathway 0.606541439 0.367593145 0.995661623 1 24

hsa03320 PPAR signaling pathway 0.624806784 0.370554856 0.9959646 1 29

hsa04962 Vasopressin-regulated water reabsorption 0.584585048 0.374360219 0.996132237 1 20

hsa00052 Galactose metabolism 0.63845918 0.383642286 0.996831484 1 17

hsa04730 Long-term depression 0.573727796 0.427601115 0.998802579 1 19

hsa04530 Tight junction 0.498026244 0.433256405 0.999013691 1 49

hsa00051 Fructose and mannose metabolism 0.658391143 0.465713336 0.999546701 1 20

hsa00770 Pantothenate and CoA biosynthesis 0.653033847 0.489354291 0.999719609 1 11

hsa04062 Chemokine signaling pathway 0.43477825 0.497523935 0.999820418 1 93

hsa04270 Vascular smooth muscle contraction 0.653822166 0.505650324 0.999851884 1 37

hsa04920 Adipocytokine signaling pathway 0.592032085 0.510432064 0.999868478 1 37

hsa04971 Gastric acid secretion 0.618065663 0.518522828 0.999890323 1 27

hsa04210 Apoptosis 0.610866122 0.523650063 0.999911754 1 50

hsa00480 Glutathione metabolism 0.669833063 0.547138149 0.999949789 1 19

hsa04620 Toll-like receptor signaling pathway 0.617618464 0.552207624 0.999961206 1 46

hsa00520 Amino sugar and nucleotide sugar metabolism 0.708634695 0.598621439 0.999990023 1 24

hsa00860 Porphyrin and chlorophyll metabolism 0.631675707 0.62124119 0.999992068 1 12

hsa04910 Insulin signaling pathway 0.687013218 0.602859056 0.999992549 1 64

hsa00531 Glycosaminoglycan degradation 0.711055963 0.633352137 0.999996447 1 15

hsa04520 Adherens junction 0.537875253 0.639337348 0.999997254 1 34

hsa04722 Neurotrophin signaling pathway 0.620863687 0.636079059 0.999997438 1 62

hsa04510 Focal adhesion 0.570032152 0.641011783 0.99999784 1 77

hsa04662 B cell receptor signaling pathway 0.482772868 0.651501074 0.999998246 1 48

hsa00590 Arachidonic acid metabolism 0.71040965 0.654543094 0.999998467 1 22

hsa00190 Oxidative phosphorylation 0.54379846 0.668569919 0.999999003 1 57

hsa04976 Bile secretion 0.691975954 0.676990106 0.999999209 1 19

hsa04650 Natural killer cell mediated cytotoxicity 0.539077201 0.681314767 0.999999445 1 65

hsa04966 Collecting duct acid secretion 0.677815621 0.704455122 0.999999635 1 13

hsa00500 Starch and sucrose metabolism 0.728003334 0.705126812 0.999999737 1 20

hsa00010 Glycolysis / Gluconeogenesis 0.708824854 0.739123308 0.999999934 1 35

hsa04145 Phagosome 0.494910222 0.758400951 0.999999972 1 82

hsa04664 Fc epsilon RI signaling pathway 0.654002099 0.913638652 1 1 32

hsa00030 Pentose phosphate pathway 0.689858521 0.990613986 1 1 16

hsa04380 Osteoclast differentiation 0.480740605 1.083941709 1 1 76

hsa04666 Fc gamma R-mediated phagocytosis 0.559549688 1.166895344 1 1 51

hsa04142 Lysosome 0.745016658 1.204835267 1 1 74

hsa04670 Leukocyte transendothelial migration 0.612557078 1.300194212 1 1 51

hsa04810 Regulation of actin cytoskeleton 0.560717439 1.27237948 1 1 91

hsa00040 Pentose and glucuronate interconversions NA NA NA NA 8

hsa00053 Ascorbate and aldarate metabolism NA NA NA NA 3

hsa00061 Fatty acid biosynthesis NA NA NA NA 4

hsa00072 Synthesis and degradation of ketone bodies NA NA NA NA 4

hsa00100 Steroid biosynthesis NA NA NA NA 6

hsa00120 Primary bile acid biosynthesis NA NA NA NA 5

hsa00130 Ubiquinone and other terpenoid-quinone biosynthesis NA NA NA NA 4

hsa00140 Steroid hormone biosynthesis NA NA NA NA 7

hsa00232 Caffeine metabolism NA NA NA NA 1

hsa00250 Alanine, aspartate and glutamate metabolism NA NA NA NA 8

hsa00290 Valine, leucine and isoleucine biosynthesis NA NA NA NA 8

hsa00300 Lysine biosynthesis NA NA NA NA 0

hsa00360 Phenylalanine metabolism NA NA NA NA 6

hsa00400 Phenylalanine, tyrosine and tryptophan biosynthesis NA NA NA NA 2

hsa00410 beta-Alanine metabolism NA NA NA NA 9

hsa00430 Taurine and hypotaurine metabolism NA NA NA NA 3

hsa00460 Cyanoamino acid metabolism NA NA NA NA 3

hsa00471 D-Glutamine and D-glutamate metabolism NA NA NA NA 1

hsa00472 D-Arginine and D-ornithine metabolism NA NA NA NA 1

hsa00533 Glycosaminoglycan biosynthesis - keratan sulfate NA NA NA NA 6

hsa00591 Linoleic acid metabolism NA NA NA NA 5

hsa00592 alpha-Linolenic acid metabolism NA NA NA NA 3

hsa00603 Glycosphingolipid biosynthesis - globo series NA NA NA NA 9

hsa00630 Glyoxylate and dicarboxylate metabolism NA NA NA NA 6

hsa00670 One carbon pool by folate NA NA NA NA 8

hsa00730 Thiamine metabolism NA NA NA NA 2

hsa00740 Riboflavin metabolism NA NA NA NA 5

hsa00750 Vitamin B6 metabolism NA NA NA NA 4

hsa00780 Biotin metabolism NA NA NA NA 1

hsa00785 Lipoic acid metabolism NA NA NA NA 2

hsa00790 Folate biosynthesis NA NA NA NA 4

hsa00900 Terpenoid backbone biosynthesis NA NA NA NA 6

hsa00910 Nitrogen metabolism NA NA NA NA 7

hsa00920 Sulfur metabolism NA NA NA NA 8

hsa03450 Non-homologous end-joining NA NA NA NA 5

hsa04122 Sulfur relay system NA NA NA NA 5

hsa04140 Regulation of autophagy NA NA NA NA 8

hsa04614 Renin-angiotensin system NA NA NA NA 7

hsa04744 Phototransduction NA NA NA NA 7

hsa04964 Proximal tubule bicarbonate reclamation NA NA NA NA 4

hsa04977 Vitamin digestion and absorption NA NA NA NA 6

---

## [Decision Letter · Decision Letter 1]

15 Sep 2020

PONE-D-20-17102R1

Immune gene expression networks in sepsis: A network biology approach

PLOS ONE

Dear Dr. JEKARL,

Thank you for submitting your manuscript to PLOS ONE. After careful consideration, we feel that it has merit but does not fully meet PLOS ONE’s publication criteria as it currently stands. Therefore, we invite you to submit a revised version of the manuscript that addresses the points raised during the review process.

ACADEMIC EDITOR:

Both reviewers still have several major concerns which need to be addressed. The authors should also provide either a marked up file indicating specific changes that have been made, or corresponding excerpts from the revised manuscript collated into a single document. 

We look forward to receiving your revised manuscript.

Kind regards,

Jishnu Das, Ph.D.

Academic Editor

PLOS ONE

Reviewers' comments:

Reviewer's Responses to Questions

**Comments to the Author**

1. If the authors have adequately addressed your comments raised in a previous round of review and you feel that this manuscript is now acceptable for publication, you may indicate that here to bypass the “Comments to the Author” section, enter your conflict of interest statement in the “Confidential to Editor” section, and submit your "Accept" recommendation.

Reviewer #1: (No Response)

Reviewer #2: (No Response)

2. Is the manuscript technically sound, and do the data support the conclusions?

Reviewer #1: Partly

Reviewer #2: Partly

3. Has the statistical analysis been performed appropriately and rigorously? 

Reviewer #1: Yes

Reviewer #2: Yes

4. Have the authors made all data underlying the findings in their manuscript fully available?

Reviewer #1: Yes

Reviewer #2: Yes

5. Is the manuscript presented in an intelligible fashion and written in standard English?

Reviewer #1: Yes

Reviewer #2: No

6. Review Comments to the Author

Reviewer #1: The authors state that they have revised the manuscript extensively, having analyzed the full transcriptome and report those results, and then focus on immune gene expression. However, I cannot find evidence for these major changes in the revision that was provided, other than in the material submitted to the journal that generates the first ~60 pages of the PDF. Specifically, there the core concerns I had are still there. Specifically:

1. Abstract: there is no mention of the analysis of the full transcriptome. Furthermore, in the Results, the authors report the finding that the adaptive immunity gene transcripts are reduced in expression. As written currently in the Abstract (“…the network topological structure revealed that adaptive immunity tended to form a prominent and isolated cluster in sepsis”), it is possible to misinterpret the findings of an adaptive immune cluster in non-survivors as meaning that the expression of these genes is increased, when in fact it is decreased. This needs to be revised.

2. Introduction: the Introduction needs to be revised to mention clearly that the authors looked at the full transcriptome and that they then focused on immune gene expression. As it stands now, the Introduction is essentially unchanged from the original version.

3. Results: the Results seem to be the same as those submitted previously. All of the analysis described in the first ~60 pages of the PDF prior to the title page seems to be missing from the manuscript. This needs to be revised to include those analyses.

4. The authors should indicate all of their changes in yellow highlight to make reviewing possible.

Reviewer #2: Dear Associate Editor,

The manuscript 'Gene expression networks in sepsis: A network biology approach' by Kim et al analyzes various immune system pathways in patients with sepsis compared to healthy controls using network analysis approach and show that adaptive immune cells are isolated and prominent in that network in patients with sepsis. The strength of this manuscript is that they have used diverse datasets for analysis and obtained comparable results across all the datasets. However, there are some minor concerns in this manuscript, which the authors have failed to address.

Major comment:

1. The revised manuscript contains a lot of minor errors and there is no attention to detail. It looks like the authors were in a rush to resubmit.

2. Figure 1 is missing in the revised manuscript.

3. The authors need to seek help from some professional services on the English language for the following reasons:

a. Words are redundant in a sentence in so many places

b. There is no continuity and the logic of presentation is missing and hard to follow at most places.

4. The most interesting part of this analyses are:

a. the relationship of platelets and complement-coagulation in sepsis survivors and

b. the relationship of adaptive immune genes in sepsis non-survivors.

Appreciate the authors for the additional analyses in relation to this point in the previous iteration and providing two references in connection with point a, but they fail to discuss this properly and are not providing sufficient biological evidences to support their claim and inferring it clearly or its relevance to the actual pathogenesis in SIRS and sepsis. Instead they are just stating their findings in the discussion.

5. The discussion is still long and there is too much discussion on the network analysis rather than the actual data and science behind that finding.

Minor comments:

1. The authors should follow standard abbreviations rather than having their own in the new table (table 3) that has been included.

For example: Toll like receptor – TLR; B cell receptor – BCR; Chemokine – CXC and etc. All the abbreviations should be in a standard format.

2. When the authors use a particular term, they should stick to that throughout the manuscript, example: healthy control or normal control.

3. Track changes in the manuscript was very difficult when we compare to version 1 submitted earlier, the authors are not consistent with the striking through (deleted parts) at many places.

4. Supp fig 14 – 19: They are green cells and not gene cells

7. PLOS authors have the option to publish the peer review history of their article (what does this mean?). If published, this will include your full peer review and any attached files.

Reviewer #1: No

---

## [Author Response · Author response to Decision Letter 1]

6 Oct 2020

Summary of revision

As commented by the reviewers, english editing service (Wiley Editing Services) was performed.

PONE-D-20-17102R1

Immune gene expression networks in sepsis: A network biology approach

PLOS ONE

Reviewer #1: The authors state that they have revised the manuscript extensively, having analyzed the full transcriptome and report those results, and then focus on immune gene expression. However, I cannot find evidence for these major changes in the revision that was provided, other than in the material submitted to the journal that generates the first ~60 pages of the PDF. Specifically, there the core concerns I had are still there. Specifically:

1-1. Abstract: there is no mention of the analysis of the full transcriptome. Furthermore, in the Results, the authors report the finding that the adaptive immunity gene transcripts are reduced in expression. 

 Thank you for your kind comment.

Following sentence was added in Abstract.

"Total transcriptome was analyzed for differentially expressed genes (DEG), followed by gene set enrichment analysis."

Following sentence was added in Introduction.

"Total transcriptome was analyzed for differentially expressed genes (DEGs) between healthy control and sepsis groups. We performed gene set enrichment analysis for DEGs, which revealed up- or down-regulated pathways."

Following sentence was added in Method section

" Analysis of differentially expressed genes (DEG) between control and patient groups with equal sample number was performed for total transcriptome."

Following sentence was added in Result section

"Total transcriptome analysis of healthy control and sepsis group revealed that 2180 to 8229 genes showed expression differences in datasets."

1-2. Furthermore, in the Results, the authors report the finding that the adaptive immunity gene transcripts are reduced in expression. As written currently in the Abstract ("…the network topological structure revealed that adaptive immunity tended to form a prominent and isolated cluster in sepsis"), it is possible to misinterpret the findings of an adaptive immune cluster in non-survivors as meaning that the expression of these genes is increased, when in fact it is decreased. This needs to be revised.

 Thank you for your kind comments.

We have revised abstract as follows

"Altogether, network and gene set enrichment analysis showed that adaptive-immunity-related genes along with TCR pathway were down-regulated and isolated from immune the network that seemed to be associated with unfavorable prognosis. Prominence of platelet and complement-coagulation-related genes in the immune network was associated with survival in sepsis. Complement-coagulation pathway was up-regulated in the sepsis group that was associated with favorable prognosis. Network and gene set enrichment analysis supported elucidation of sepsis pathogenesis."

We have revised result section as follows

"Altogether, these findings were interpreted along with the network analysis results. In network analysis, adaptive immunity genes were prominent and isolated from network in sepsis patient group (Fig. 1). T cell receptor signaling (TCR) pathway was down-regulated in 100% (6/6) of the datasets. These data imply that adaptive immunity related with T cells might be impaired or dysregulated in sepsis. In addition, adaptive immunity was associated with unfavorable prognosis in sepsis group. Complement and coagulation cascade was prominent in sepsis survivors of GSE95233 datasets (Fig. 2). Complement and coagulation cascade was increased in sepsis in 83.3% (5/6) of datasets, and these data imply that these pathways may be involved in convalescence process."

2. Introduction: the Introduction needs to be revised to mention clearly that the authors looked at the full transcriptome and that they then focused on immune gene expression. As it stands now, the Introduction is essentially unchanged from the original version.

 Thank you for your kind comments.

Following sentence was added in Introduction.

"Total transcriptome was analyzed for differentially expressed genes (DEGs) between healthy control and sepsis groups. We performed gene set enrichment analysis for DEGs, which revealed up- or down-regulated pathways."

3. Results: the Results seem to be the same as those submitted previously. All of the analysis described in the first ~60 pages of the PDF prior to the title page seems to be missing from the manuscript. This needs to be revised to include those analyses.

 Thank you for your kind comments.

We have revised as commented.

4. The authors should indicate all of their changes in yellow highlight to make reviewing possible.

 Thank you for your kind comments.

We have marked in red highlight

Reviewer #2: Dear Associate Editor,

The manuscript 'Gene expression networks in sepsis: A network biology approach' by Kim et al analyzes various immune system pathways in patients with sepsis compared to healthy controls using network analysis approach and show that adaptive immune cells are isolated and prominent in that network in patients with sepsis. The strength of this manuscript is that they have used diverse datasets for analysis and obtained comparable results across all the datasets. However, there are some minor concerns in this manuscript, which the authors have failed to address.

Major comment:

1. The revised manuscript contains a lot of minor errors and there is no attention to detail. It looks like the authors were in a rush to resubmit.

 Thank you for your kind comment

We had our manuscript checked by English Editing Service

2. Figure 1 is missing in the revised manuscript.

 Sorry for the error and thank you for the comment.

We have uploaded Figure 1.

3. The authors need to seek help from some professional services on the English language for the following reasons:

a. Words are redundant in a sentence in so many places

b. There is no continuity and the logic of presentation is missing and hard to follow at most places.

 Thank you for your kind comment

We had our manuscript checked by English Editing Service

4. The most interesting part of this analyses are:

a. the relationship of platelets and complement-coagulation in sepsis survivors and

b. the relationship of adaptive immune genes in sepsis non-survivors.

Appreciate the authors for the additional analyses in relation to this point in the previous iteration and providing two references in connection with point a, but they fail to discuss this properly and are not providing sufficient biological evidences to support their claim and inferring it clearly or its relevance to the actual pathogenesis in SIRS and sepsis. Instead they are just stating their findings in the discussion.

 Thank you for your kind comment

We have added in Discussion section

"Clustering coefficient is one of the network topologic parameters that reflects the number of links between neighboring nodes, and its increased values denote an increased probability of similar biological function46,47."

Following sentences were added in Discussion section

" On the other hand, network analysis showed that network heterogeneity was higher in adult sepsis, whereas it was lower in pediatric sepsis. Gene set enrichment analysis showed that the adult sepsis group had more down-regulated pathways compared to the pediatric group, which had more up-regulated pathways. High network heterogeneity values imply that the nodes are connected with other nodes of different types, while lower values denote that the nodes are connected with similar other nodes. Altogether, these data suggest that immune senescence or immune hyperactivation are a driving force of immune response in adult and pediatric sepsis, respectively.

 The features of immune dysfunction or immune response in sepsis include defective antigen presentation and adaptive immunity (T and B cell abnormalities), abnormalities in NK cell activity, diminished immunoglobulin level, alteration in neutrophils, abnormal cytokine levels, unbalanced pro- and anti-inflammatory cytokines, complement consumption, and defective bacterial killing4,5. This complexity hampers understanding of immunologic features in sepsis. Immune suppression, immune paralysis and hypoinflammatory response are reported in sepsis that is intermingled with hyper inflammatory state6-8. 

 This study showed that in the adult sepsis group, there were more down-regulated pathways, whereas in the pediatric group, there were more up-regulated pathways. These data suggest that immune suppression might be more dominant in the adult sepsis group, whereas hyper-inflammatory states might be dominant in the pediatric sepsis group. The mechanism of immune response was indeed different between adult and pediatric patients in sepsis. However, down-regulation of TCR and APC pathway and up-regulation of complement-coagulation cascade pathway was found in most of adult and pediatric sepsis patients. These findings are congruent with previous reports that, in sepsis, CD4+ T cell showed cell exhaustion, increased apoptosis and decreased adhesion molecules, and CD28 and TCR diversity. In sepsis, CD8+ T cell also showed cell exhaustion, increased apoptosis, and decreased cytotoxic function, cytokine secretion, and TCR diverstiy8. Antigen presentation by follicular dendritic cell or dendritic cells is decreased in sepsis8. In the present study, isolation and prominent adaptive immune genes from the rest of the immune network were noted, and these data imply that part of the adaptive immune function was not in equilibrium or in a regulated state. As this isolated adaptive immune network was found in nonsurvivors, initial network state along with down-regulated pathways related with adaptive immunity resulted in unfavorable prognosis. 

 Complement and coagulation pathway was up-regulated in sepsis patients. However, network analysis showed that platelet and complement-and-coagulation-related genes were prominent and isolated in survivors, and that this initial state resulted in a favorable prognosis. On the contrary, it is reported that persistently activated complement system, including C3a and C5a, is found in sepsis, which is associated with unfavorable prognosis6-8. Excessive activation of complement system potentiated activated innate immune system in sepsis, and this also resulted in an unfavorable prognosis in the sepsis group6. 57. These findings imply that initial inactivation of complement-coagulation pathway or persistent, uncontrolled activation of these pathways might be associated with unfavorable prognosis.

 Among prominent clusters from survivors, IL1A, IL12, IL21, and IL23 genes were also included. Although statistically insignificant, previous studies showed that the mean cytokines levels (pg/mL) for survivors (n = 63) and nonsurvivors (n = 17) were as follows49: IL1A, 24.8, 7.4; IL12, 15.7, 0; and IL21, 138.9, 10.1, IL23, 75.4, 16.7, respectively. Mean cytokine level of IL1A, IL12, IL21, IL23 was increased in survivors of sepsis. These data suggest that platelet, complement and coagulation related molecules and IL1A, IL12, IL21, IL23 might be closely related with convalescence process in sepsis."

5. The discussion is still long and there is too much discussion on the network analysis rather than the actual data and science behind that finding.

 Thank you for your kind comment

We have deleted following sentences in Discussion section

Following sentences are moved to other part of the discussion section

"Features of immune dysfunction or the immune response in sepsis include defective antigen presentation and adaptive immunity (T and B cell abnormalities), abnormalities in NK cell activity, diminished immunoglobulin level, alteration in neutrophils, abnormal cytokine levels, unbalanced pro- and anti-inflammatory cytokines, complement consumption, and defective bacterial killing4,5. This complexity hampers understanding of immunologic features in sepsis."

Following sentences are deleted.

"Studying gene expression profiles might be an improved way to understand immune dysregulation and the host immune response in sepsis11-17. The interrelations among immune genes were analysed by a network analysis approach in this study that might reflect the comprehensive aspect of the dynamics of gene expression. Analysis of the topologic structure of the network in sepsis revealed that adaptive immunity-related genes formed a cluster in sepsis patients (Supplementary Figs. S1 - S12), which tended to be isolated. In the isolated cluster, most of them were annotated to the T cell receptor signalling pathway, which implies that these functions were dysregulated or uncontrolled with other immune networks compared to those in the healthy normal control group. In addition, gene set enrichment for pathway analysis showed that T cell receptor signalling pathway was down regulated in studies all the datasets. Altogether, adaptive immunity, especially T cell associated gene seemed to be suppressed or dysregulated7."

Following sentences are deleted

"In the isolated cluster, most of them were annotated to the T cell receptor signalling pathway, which implies that these functions were dysregulated or uncontrolled with other immune networks compared to those in the healthy normal control group. In addition, gene set enrichment for pathway analysis showed that T cell receptor signalling pathway was down regulated in studies all the datasets. Altogether, adaptive immunity, especially T cell associated gene seemed to be suppressed or dysregulated7." 

Following sentences were deleted

"Network topologic parameters describe the architecture and structure of the network. Degree, distance, diameter, clustering coefficient, modularity and shortest path can be calculated. In this study, the topologic structure of isolated or prominent clusters was in line with some of the network topologic parameters. The clustering coefficient and modularity values in the sepsis group were increased in 5 out of 6 datasets compared to those in the healthy control group, with statistical significance. The clustering coefficient reflects the number of links between neighbouring nodes, and increased values denote an increased probability of similar biological function46,47. The clustering coefficient is known to be correlated with modularity and scale-free networks, which implies a functional aspect of the network46,47."

Following sentences was deleted.

"Although modularity implies the biological process of a network, the isolation and disconnection of a module with a component of a similar functional node might result in adverse reactions in the whole network."

Following sentences was deleted

"Shortest path length was decreased in 66% (2/3) of adult and paediatric datasets, respectively, which was in line with previous literature20. This parameter, along with network diameter, is related to the functional and biological processes of a network and increases the probability of the presence of a sub-network or module with biological functions46,47 It has been reported that as a network evolves, complex biological functions are executed through modules, which increases the path length or network diameter52. The inconsistency of the decreased shortest path and increased clustering coefficient might be an indication of immune dysfunction, which requires further study.

 Nodes with the highest degree that might be a hub were searched and expected to play an important role in the biological network18,19. In this study, the gene with the highest degree varied among the different tested datasets. The hub node has a great influence on the network, and removal of the hub would disconnect the network and could lead to dissociation of the network. The hub could be a therapeutic target for treatment. However, as the gene with the highest degree varied in this study, immune function might be exerted not by a few genes or a hub but rather by a module formed by genes."

Minor comments:

1. The authors should follow standard abbreviations rather than having their own in the new table (table 3) that has been included.

For example: Toll like receptor - TLR; B cell receptor - BCR; Chemokine - CXC and etc. All the abbreviations should be in a standard format.

Thank you for your kind comment. Abbreviation was revised to standard abbreviations

Table 3. Differentially expressed gene analysis between control and sepsis followed by gene set pathway analysis 

 GSE

54514a GSE

57065 GSE

95233 GSE

64456 GSE

66099 GSE

72829

Patient type Adult Adult Adult Pediatric Pediatric Pediatric

Control cases (n) 18 25 22 19 49 52

Sequential selection 

of sepsis cases (n) 18 25 22 19 49 52

Differentially expressed 

Genes (n) 2180 7885 6784 3123 8229 7463

Up regulated pathway (n) 4 60 44 13 78 51

Down regulated pathway (n) 1 33 33 16 33 15

Up regulated immune

pathway C, Fc, RIG C, Fc C, Fc, NK, K, TLR APC, C, CXC & etc, Fc, IgA, BCR, C, Fc, CXC & etc, LK, NK, TLR 

Down regulated immune

pathway TCR APC, TCR, 

H, K APC, BCR, 

CXC & etc, 

IgA, TCR APC, IgA, TCR H, TCR APC, IgA, TCR

aDifferentially expressed gene of GSE54514 was selected based on global P-value (0.05) and other datasets are based on FDR (<0.01).

APC, Antigen processing and presentation; BCR, B cell receptor signaling; C, Complement and coagulation; CXC & etc, chemokine signaling; Fc, Fc gamma R mediated phagocytosis or Fc epsilon RI signaling; H, hematopoietic cell lineage; IgA, intestinal immune network for IgA production; LM, leukocyte endothelial migration; NK, NK cell mediated cytotoxicity; RIG, RIG-I like receptor signaling; TLR, Toll like receptor signaling; TCR, T cell receptor signaling.

2. When the authors use a particular term, they should stick to that throughout the manuscript, example: healthy control or normal control.

Thank you for your kind comment.

We have corrected the errors.

3. Track changes in the manuscript was very difficult when we compare to version 1 submitted earlier, the authors are not consistent with the striking through (deleted parts) at many places.

Thank you for your kind comment.

Track changes are noted as commented.

4. Supp fig 14 - 19: They are green cells and not gene cells

Thank you for your kind comment.

We have corrected the errors.

---

## [Decision Letter · Decision Letter 2]

23 Oct 2020

PONE-D-20-17102R2

Immune gene expression networks in sepsis: A network biology approach

PLOS ONE

Dear Dr. JEKARL,

Thank you for submitting your manuscript to PLOS ONE. After careful consideration, we feel that it has merit but does not fully meet PLOS ONE’s publication criteria as it currently stands. Therefore, we invite you to submit a revised version of the manuscript that addresses the points raised during the review process.

ACADEMIC EDITOR:

Reviewer 1 still has major concerns with the structure of the manuscript. As recommended by Reviewer 1, the global transcriptomic analysis should be presented first, and the other findings should then be contextualized with regard to these systems analyses. I concur with the reviewer's suggestions, and recommend a complete re-ordering of the manuscript rather than minor changes to a few sentences.

We look forward to receiving your revised manuscript.

Kind regards,

Jishnu Das, Ph.D.

Academic Editor

PLOS ONE

Reviewers' comments:

Reviewer's Responses to Questions

**Comments to the Author**

1. If the authors have adequately addressed your comments raised in a previous round of review and you feel that this manuscript is now acceptable for publication, you may indicate that here to bypass the “Comments to the Author” section, enter your conflict of interest statement in the “Confidential to Editor” section, and submit your "Accept" recommendation.

Reviewer #1: (No Response)

Reviewer #2: All comments have been addressed

2. Is the manuscript technically sound, and do the data support the conclusions?

Reviewer #1: No

Reviewer #2: Yes

3. Has the statistical analysis been performed appropriately and rigorously? 

Reviewer #1: Yes

Reviewer #2: Yes

4. Have the authors made all data underlying the findings in their manuscript fully available?

Reviewer #1: Yes

Reviewer #2: Yes

5. Is the manuscript presented in an intelligible fashion and written in standard English?

Reviewer #1: Yes

Reviewer #2: Yes

6. Review Comments to the Author

Reviewer #1: The authors have not addressed my underlying critique, which is that the paper needs to be rewritten to reflect the complete data analysis initially, and then discuss why they focused on only the immune-related genes.

Reviewer #2: Dear Associate Editor,

The manuscript 'Gene expression networks in sepsis: A network biology approach' by Kim et al analyzes various immune system pathways in patients with sepsis compared to healthy controls using network analysis approach and show that adaptive immune cells are isolated and prominent in that network in patients with sepsis. The strength of this manuscript is that they have used diverse datasets for analysis and obtained comparable results across all the datasets. The authors have addressed all the concerns raised earlier and they are satisfactory.

7. PLOS authors have the option to publish the peer review history of their article (what does this mean?). If published, this will include your full peer review and any attached files.

Reviewer #1: No

---

## [Author Response · Author response to Decision Letter 2]

29 Oct 2020

ACADEMIC EDITOR #

Reviewer 1 still has major concerns with the structure of the manuscript. As recommended by Reviewer 1, the global transcriptomic analysis should be presented first, and the other findings should then be contextualized with regard to these systems analyses. I concur with the reviewer's suggestions, and recommend a complete re-ordering of the manuscript rather than minor changes to a few sentences.

Reviewer #1: The authors have not addressed my underlying critique, which is that the paper needs to be rewritten to reflect the complete data analysis initially, and then discuss why they focused on only the immune-related genes.

 Thank you for your kind comments.

We have re-orderded the manuscript as commented by academic editor and Reviewer. The global transcriptomic analysis was presented first, followed by other analysis.

Correction 1. Following sentence was revised.

In this study, gene expression signatures from a public database, Gene Expression Omnibus (GEO), were analyzed. We selected 998 immune-related genes included in immune response pathways using gene expression profiles to construct a network for the healthy control and sepsis groups. Total transcriptome was analyzed for differentially expressed genes (DEGs) between healthy control and sepsis groups. We performed gene set enrichment analysis for DEGs, which revealed up- or down-regulated pathways.

 Revised sentence

In this study, gene expression signatures from a public database, Gene Expression Omnibus (GEO), were analyzed. Total transcriptome was analyzed for differentially expressed genes (DEGs) between healthy control and sepsis groups, followed by gene set enrichment analysis for DEGs, which revealed up- or down-regulated pathways. We selected 998 immune-related genes included in immune response pathways using gene expression profiles to construct a network for the healthy control and sepsis groups...

 Correction 2. Following sentence was revised.

For network construction, several appropriate R software packages were used24, including Bioconductor25, affy26, GEOquery27, lumi28, and limma29.

 Revised sentence

For differentially expressed genes (DEG) analysis between control and patient groups based on total transcriptome, gene set enrichment for pathway analysis and network construction, several appropriate R software packages were used24, including Bioconductor25, affy26, GEOquery27, lumi28, and limma29

Correction 3. Following sentence was placed head of network analysis phrases in method section

 Gene set enrichment for pathway analysis

 Analysis of DEG was performed for total transcriptome . After a t-test using basic functions in R software, genes were selected based on false discovery rate of less than 0.01 (FDR < 0.01)32 except GSE54514, which was based on P-value of less than 0.05. Based on DEG analysis, gene set enrichment for pathway analysis using GAGE package, based on R language, was performed33. GAGE package applies input data to the Kyoto Encylopedia of Genes and Genomes (KEGG) pathways. Significant pathways were plotted for heatmap that was selected based on global P-value of less than 0.05.

Correction 4. Following subtitle was added.

Bioinformatics phrase in method section was divided into 2 parts, and following subtitle was added.

 Immune response related genes

Correction 5. In results section, gene set enrichment analysis section was presented ahead of network analysis.

Correction 6. Table 3 was revised to Table 2 and vice versa.

Correction 7. In Discussion section, following paragraph was presented ahead. 

 The features of immune dysfunction or immune response in sepsis include defective antigen presentation and adaptive immunity (T and B cell abnormalities), abnormalities in NK cell activity, diminished immunoglobulin level, alteration in neutrophils, abnormal cytokine levels, unbalanced pro- and anti-inflammatory cytokines, complement consumption, and defective bacterial killing4,5. This complexity hampers understanding of immunologic features in sepsis. Immune suppression, immune paralysis and hypoinflammatory response are reported in sepsis that is intermingled with hyper inflammatory state6-8. 

 This study showed that in the adult sepsis group, there were more down-regulated pathways, whereas in the pediatric group, there were more up-regulated pathways. These data suggest that immune suppression might be more dominant in the adult sepsis group, whereas hyper-inflammatory states might be dominant in the pediatric sepsis group. The mechanism of immune response was indeed different between adult and pediatric patients in sepsis. However, down-regulation of TCR and APC pathway and up-regulation of complement-coagulation cascade pathway was found in most of adult and pediatric sepsis patients. These findings are congruent with previous reports that, in sepsis, CD4+ T cell showed cell exhaustion, increased apoptosis and decreased adhesion molecules, and CD28 and TCR diversity. In sepsis, CD8+ T cell also showed cell exhaustion, increased apoptosis, and decreased cytotoxic function, cytokine secretion, and TCR diverstiy8. Antigen presentation by follicular dendritic cell or dendritic cells is decreased in sepsis8. In the present study, isolation and prominent adaptive immune genes from the rest of the immune network were noted, and these data imply that part of the adaptive immune function was not in equilibrium or in a regulated state. As this isolated adaptive immune network was found in nonsurvivors, initial network state along with down-regulated pathways related with adaptive immunity resulted in unfavorable prognosis. 

 Complement and coagulation pathway was up-regulated in sepsis patients. However, network analysis showed that platelet and complement-and-coagulation-related genes were prominent and isolated in survivors, and that this initial state resulted in a favorable prognosis. On the contrary, it is reported that persistently activated complement system, including C3a and C5a, is found in sepsis, which is associated with unfavorable prognosis6-8. Excessive activation of complement system potentiated activated innate immune system in sepsis, and this also resulted in an unfavorable prognosis in the sepsis group6. These findings imply that initial inactivation of complement-coagulation pathway or persistent, uncontrolled activation of these pathways might be associated with unfavorable prognosis

Correction 8. Order of references, Supplementary Tables and Supplementary Figures were revised.

Reviewer #2: Dear Associate Editor,

The manuscript 'Gene expression networks in sepsis: A network biology approach' by Kim et al analyzes various immune system pathways in patients with sepsis compared to healthy controls using network analysis approach and show that adaptive immune cells are isolated and prominent in that network in patients with sepsis. The strength of this manuscript is that they have used diverse datasets for analysis and obtained comparable results across all the datasets. The authors have addressed all the concerns raised earlier and they are satisfactory.

 Thank you for taking time reading, and correcting this article. The article was improved better than the previous ones. We unexpectedly found comparable results across the datasets. We also thank other researchers for sharing and uploading the precious data. We hope this article might help or support others to treat or elucidate sepsis.

---

## [Decision Letter · Decision Letter 3]

4 Jan 2021

PONE-D-20-17102R3

Immune gene expression networks in sepsis: A network biology approach

PLOS ONE

Dear Dr. JEKARL,

Thank you for submitting your manuscript to PLOS ONE. After careful consideration, we feel that it has merit but does not fully meet PLOS ONE’s publication criteria as it currently stands. Therefore, we invite you to submit a revised version of the manuscript that addresses the points raised during the review process.

ACADEMIC EDITOR:

As one referee continued to raise significant concerns with the manuscript, we reached out to an arbitrating reviewer. The comments raised by this additional referee should be addressed in a revised manuscript.

We look forward to receiving your revised manuscript.

Kind regards,

Jishnu Das, Ph.D.

Academic Editor

PLOS ONE

Reviewers' comments:

Reviewer's Responses to Questions

**Comments to the Author**

1. If the authors have adequately addressed your comments raised in a previous round of review and you feel that this manuscript is now acceptable for publication, you may indicate that here to bypass the “Comments to the Author” section, enter your conflict of interest statement in the “Confidential to Editor” section, and submit your "Accept" recommendation.

Reviewer #3: (No Response)

2. Is the manuscript technically sound, and do the data support the conclusions?

Reviewer #3: Partly

3. Has the statistical analysis been performed appropriately and rigorously? 

Reviewer #3: Yes

4. Have the authors made all data underlying the findings in their manuscript fully available?

Reviewer #3: Yes

5. Is the manuscript presented in an intelligible fashion and written in standard English?

Reviewer #3: Yes

6. Review Comments to the Author

Reviewer #3: Major comments:

1. The Results section subtitles are written in a method-centric manner. Changing these subtitles to informative insights resulting from the analysis would help bring out the message/insight that the authors are conveying. At this point, the manuscript reads like a report of a set of analyses run on the data. A similar undue emphasis on the step-by-step approach rather than results is reflected in the Abstract as well.

A side effect of this problem are the many instances of redundancy in the Results section, where content similar to the Methods section was restated. For example, the “Network construction” section can be removed altogether. There are only a few non-redundant sentences toward the end of that section, which can be merged with the next section titled “Network topology analysis”.

Overall, the results section does not flow and the analyses do not flow well into biological insights.

2. In the section titled “Network parameter analysis”, the increase or decrease in the network topological parameters that were highlighted seemed minimal. For example, the sentence “Network heterogeneity was increased in 100% (3/3) of adult datasets, but were decreased in 100% (3/3) of pediatric datasets.”  the difference of Network heterogeneity for GSE57065 between the control (1.151) and sepsis group (1.172) is 1.8%. Additionally, for dataset GSE95233-1, non-survivors have decreased network heterogeneity compared to survivors. The reader might not be convinced that the minute differences in these topological parameters matter. Similarly, for the clustering coefficient the differences between control and sepsis groups were low. Reporting the differences in terms of percentages might help clarify the more important differences.

3. Inconsistencies in the results text/table/abstract:

(a) “Average degree and shortest path were increased in the sepsis group among 66% (2/3) of adult and pediatric datasets.” This sentence from the Results section simply seems to be in contrast with the shortest paths reported in Table. In fact, shortest path decreased in 4 out of 6 datasets total.

(b) This sentence in Abstract is inaccurate: The clustering coefficient and modularity parameters were increased in 5/6 datasets in the sepsis  According to Table 3, the modularity coefficient is increased in 4/6 datasets. This is also mentioned in the Results section: “Modularity value was decreased in 66% (2/3) of adult datasets, whereas modularity value was increased in 100% (3/3) of pediatric datasets in the sepsis group.” A similar mistake was made when referring to this in the Conclusions.

Also, was there a reason why only these two network topology parameters were highlighted (inaccurately for the clustering coefficient) in the abstract and conclusions?

Mistakes such as (a) and (b) plus the ones listed below among “minor typos and edits” are unacceptable, especially in the third revision. Other minor inconsistencies might exist that the reviewers might have not caught. The authors should take initiative in giving the manuscript a thorough last read for accuracy, consistency and correctness.

Minor comments:

1. GSE95233 is repeated twice in the abstract, please remove the redundant one. There should be a total of 6 datasets studied (as listed in Table 1), not 7.

2. Where the randomized selection without replacement (to match the number of controls) for all datasets done such that the male-female percentage of control and sepsis groups were similar per dataset? Were they selected such that the mean age was comparable between the control and sepsis groups? If no, the conclusions from analysis that followed might fall apart. And if yes, please mention in the methods section that the control and sepsis groups were selected such that they were comparable with respect to age and male/female percentage. This can be added to the section in the methods where the “Cases in the cohort” were described and Table 1 was explained.

3. Table 1: For the sake of completeness, please add description for APACHEII and SOFA abbreviations in the Table 1 legend.

4. Methods: Gene set enrichment for pathway analysis: The authors say “After a t-test using basic functions in R software, genes were selected based on false discovery rate of less than 0.01 (FDR < 0.01) except GSE54514, which was based on P-value of less than 0.05.” Why was a different metric used for this dataset? Changing the threshold for one dataset out of 6 seems arbitrary and should be justified.

7. PLOS authors have the option to publish the peer review history of their article (what does this mean?). If published, this will include your full peer review and any attached files.

Reviewer #3: No

---

## [Author Response · Author response to Decision Letter 3]

4 Feb 2021

Reviewer #3: Major comments:

1. The Results section subtitles are written in a method-centric manner. Changing these subtitles to informative insights resulting from the analysis would help bring out the message/insight that the authors are conveying. At this point, the manuscript reads like a report of a set of analyses run on the data. A similar undue emphasis on the step-by-step approach rather than results is reflected in the Abstract as well. 

 Thank you for your insightful and accurate comments.

Result of gene set enrichment analysis and network analysis could be regarded as a different analysis method. We analyzed the data using different method and obtained results, respectively. Then, the result by each method was interpreted altogether. Finally and unexpectedly, we obtained some common factors from each result. Therefore, we tried to show that the results by different method had common factors. As description of result was corresponded to the method, description in manuscript might have read like an analysis run on the data.

We corrected the subtitle as follows.

Network topologic analysis was revised to "Network structure and immune function"

1-1. A side effect of this problem are the many instances of redundancy in the Results section, where content similar to the Methods section was restated. For example, the "Network construction" section can be removed altogether. There are only a few non-redundant sentences toward the end of that section, which can be merged with the next section titled "Network topology analysis". Overall, the results section does not flow and the analyses do not flow well into biological insights.

 Thank you for your kind comment.

We deleted following paragraph in result section as the paragraph contains redundant contents.

 Network Construction

Gene expression data from sepsis patients and normal controls were selected from a GEO database that was eligible for this study (Table 1, Methods section). Networks were constructed with immune-related genes from 21 pathways from the KEGG30 database, which are listed in S1–S2 Table. Genes were annotated with the most frequent function from the KEGG30, GO31 and Reactome32 databases, which included, in order of importance, adaptive immunity, innate immunity, NK cell activity, leukocyte migration, antigen presentation, complement-coagulation, platelet activity, cytokines-chemokines, signaling, and hematopoiesis (S3 Table).

2. In the section titled "Network parameter analysis", the increase or decrease in the network topological parameters that were highlighted seemed minimal. For example, the sentence "Network heterogeneity was increased in 100% (3/3) of adult datasets, but were decreased in 100% (3/3) of pediatric datasets."  the difference of Network heterogeneity for GSE57065 between the control (1.151) and sepsis group (1.172) is 1.8%. 

 Thank you for your comments.

We admit that the numerical presentation of network parameters values seems to be minimal. We think that the tendency of network parameter provided more information compared to that of numerical value of parameters. Therefore, we reported the results whether the parameter was increased or decreased along with comparison of control and sepsis group. In addition, we thought that comparing the numerical values of parameters and using statistical method might not reflect exact information implied in the values of parameter due to the small datasets (3 adult and 3 children), which was available in public database. 

As the description in the manuscript might exaggerate the result of parameters and might lead to misinterpretation of network parameters. Therefore, we deleted all the percentage (%) in the result section. 

We deleted following sentences

Other parameters did not show significant differences between the healthy control group and the sepsis group.

2-1. Additionally, for dataset GSE95233-1, non-survivors have decreased network heterogeneity compared to survivors. The reader might not be convinced that the minute differences in these topological parameters matter. Similarly, for the clustering coefficient the differences between control and sepsis groups were low. Reporting the differences in terms of percentages might help clarify the more important differences.

 Thank you for your comments.

As commented by reviewer, we added percentages that might clarify the result in regard to GSE95233-1.

 Following sentence was added in result section

For GSE95233-1 analysis, difference between network parameters was calculated as follows: (case - control / control) x 100. The difference of clustering coefficient between survivor (0.390) and nonsurvivor (0.337) was -13.5%. The difference of modularity between survivor (0.581) and nonsurvivor (0.694) was 19.4%. 

3. Inconsistencies in the results text/table/abstract:

3(a) "Average degree and shortest path were increased in the sepsis group among 66% (2/3) of adult and pediatric datasets." This sentence from the Results section simply seems to be in contrast with the shortest paths reported in Table. In fact, shortest path decreased in 4 out of 6 datasets total.

 We are sorry for the careless errors.

We have corrected the sentence as follows.

 Shortest path was decreased in 2/3 of adult and pediatric datasets, whereas average degree was increased in 2/3 of adult and pediatric datasets.

3(b) This sentence in Abstract is inaccurate: The clustering coefficient and modularity parameters were increased in 5/6 datasets in the sepsis  According to Table 3, the modularity coefficient is increased in 4/6 datasets. This is also mentioned in the Results section: "Modularity value was decreased in 66% (2/3) of adult datasets, whereas modularity value was increased in 100% (3/3) of pediatric datasets in the sepsis group." A similar mistake was made when referring to this in the Conclusions.

 We are sorry for the careless errors.

We have corrected the error in the abstract as follows

 The clustering coefficient and modularity parameters were increased in 5/6 and 4/6 datasets in the sepsis group that seemed to be associated with increased functional aspect of the network.

We have corrected the error in Result section.

 Modularity value was increased in 1/3 of adult datasets, whereas it was increased in 3/3 of pediatric datasets in the sepsis group.

3(c) Also, was there a reason why only these two network topology parameters were highlighted (inaccurately for the clustering coefficient) in the abstract and conclusions?

 Thank you for insightful comment.

Among many network parameter, clustering coefficient are one of most studied parameters. In addition, from previous studies, we found out that clustering coefficient seemed to be associated with CRP in network analysis from hepatocellular carcinoma patients. These data seemed to imply that clustering coefficient was associated with increased inflammatory function of cytokine network (Jekarl DW et al, 2019;14(11):PLoS ONE e0224318). In this study, clustering coefficient was increased in 5/6 dataset of sepsis group which seemed to be associated with increased functional aspect of the network.

Modularity is a parameter that are associated with evolvability, multifunctionality and robustness of a network (Jekarl DW et al, 2019;14(11):PLos ONE e0224318). High modularity seemed to be associated with functional unit or function of a network. In this study, pediatric patient showed increased modularity, which implies that pediatric patient might have increased inflammatory function. These data are in line with gene set enrichment analysis that more of pathways are involved in pediatric patients compared adult.

In addition, these two parameters showed tendencies and clustering coefficient and modularity was increased in 5/6 and 4/6 datasets.

4(d) Mistakes such as (a) and (b) plus the ones listed below among "minor typos and edits" are unacceptable, especially in the third revision. Other minor inconsistencies might exist that the reviewers might have not caught. The authors should take initiative in giving the manuscript a thorough last read for accuracy, consistency and correctness.

 Thank you for your precise comment. Although this manuscript was english language edited by specialist (Wiley Language Editing Serivce), authors are responsible for these errors. We are sorry for these error and we re-read the manuscript to find potential errors. 

Minor comments:

1. GSE95233 is repeated twice in the abstract, please remove the redundant one. There should be a total of 6 datasets studied (as listed in Table 1), not 7.

 Thank you for your kind comment

 We deleted GSE95233.

2. Where the randomized selection without replacement (to match the number of controls) for all datasets done such that the male-female percentage of control and sepsis groups were similar per dataset? Were they selected such that the mean age was comparable between the control and sepsis groups? If no, the conclusions from analysis that followed might fall apart. And if yes, please mention in the methods section that the control and sepsis groups were selected such that they were comparable with respect to age and male/female percentage. This can be added to the section in the methods where the "Cases in the cohort" were described and Table 1 was explained.

 Thank you for your kind comment.

As selection was peformed randomly, sex and age is not matched. The information of sex and age of the datasets in Table 1 are derived from published paper based on these datasets. As commented by other reviewers, we divided adult patient group from pediatric patient group.

 Following sentences are added in the discussion section

As this study was a retrospective study using public datasets, sex and age was not matched for control and sepsis cases. The results might have been affected by these parameters.

3. Table 1: For the sake of completeness, please add description for APACHEII and SOFA abbreviations in the Table 1 legend.

 Thank you for your kind comment 

We have added followings in the Table 1 legend.

 ED, emergency department; ICU, intensive care unit; NA, data not available; hr, hour; d, day; APACHEII, acute physiology and chronic health evaluation II; SOFA, sequential (sepsis related) organ failure assessment

4. Methods: Gene set enrichment for pathway analysis: The authors say "After a t-test using basic functions in R software, genes were selected based on false discovery rate of less than 0.01 (FDR < 0.01) except GSE54514, which was based on P-value of less than 0.05." Why was a different metric used for this dataset? Changing the threshold for one dataset out of 6 seems arbitrary and should be justified.

Thank you for your accurate comment.

We are not sure why GSE54514 showed smallest number of differentially expressed gene (2180 genes by P-value of less than 0.05) and after adapting FDR (P-0.01), the differentially expressed gene was 176 gene. It might be explained by smallest size of control (n=18) and patient group (n=18) compared to other datasets. Small size of data might have caused dispersion of the gene expression level, which might have led to insufficient number of genes for gene set enrichment analysis selected by FDR (P-0.01) values.

 We re-analyzed the GSE54154 using same threshold and DEG was 176, which was written in the Table 2. For GAGE analysis, none of the pathway was adapted due to small size of the genes. We deleted supplementary Figure 2.

 We changed the Table1 as follows. 

Table 2. Differentially expressed gene analysis between control and sepsis followed by gene set pathway analysis 

 GSE54514a GSE57065 GSE95233 GSE64456 GSE66099 GSE72829

Patient type Adult Adult Adult Pediatric Pediatric Pediatric

Control cases (n) 18 25 22 19 49 52

Sequential selection 

of sepsis cases (n) 18 25 22 19 49 52

Differentially expressed 

Genes (n) 176 7885 6784 3123 8229 7463

Up regulated pathway (n) 0 60 44 13 78 51

Down regulated pathway (n) 0 33 33 16 33 15

Up regulated immune

pathway C, Fc, RIG C, Fc C, Fc, NK, K, TLR APC, C, CXC & etc, Fc, IgA, BCR, C, Fc, CXC & etc, LK, NK, TLR 

Down regulated immune

pathway APC, TCR, 

H, K APC, BCR, 

CXC & etc, 

IgA, TCR APC, IgA, TCR H, TCR APC, IgA, TCR

aDifferentially expressed gene was selected based on FDR (P<0.01).

APC, Antigen processing and presentation; BCR, B cell receptor signaling; C, Complement and coagulation; CXC & etc, chemokine signaling; Fc, Fc gamma R mediated phagocytosis or Fc epsilon RI signaling; H, hematopoietic cell lineage; IgA, intestinal immune network for IgA production; LM, leukocyte endothelial migration; NK, NK cell mediated cytotoxicity; RIG, RIG-I like receptor signaling; TLR, Toll like receptor signaling; TCR, T cell receptor signaling.

Following sentence was deleted.

 except GSE54514, which was based on P value of less than 0.05

Following Supplementary Figure was deleted and following supplementary figure number was changes.

 S2 Fig. The gene set analysis result of GSE54514. Red cells indicate up regulated gene sets, whereas green cells indicate down regulated gene sets.

---

## [Editor Report · Decision Letter 4]

11 Feb 2021

Immune gene expression networks in sepsis: A network biology approach

PONE-D-20-17102R4

Dear Dr. JEKARL,

We’re pleased to inform you that your manuscript has been judged scientifically suitable for publication and will be formally accepted for publication once it meets all outstanding technical requirements.

Kind regards,

Jishnu Das, Ph.D.

Academic Editor

PLOS ONE
---

## [Editor Report · Acceptance letter]

23 Feb 2021

PONE-D-20-17102R4 

Immune gene expression networks in sepsis: A network biology approach 

Dear Dr. Jekarl:

I'm pleased to inform you that your manuscript has been deemed suitable for publication in PLOS ONE. Congratulations! Your manuscript is now with our production department. 

Kind regards, 

on behalf of

Dr. Jishnu Das 

Academic Editor

PLOS ONE